# A Unified Survey on Anomaly, Novelty, Open-Set, and Out-of-Distribution Detection: Solutions and Future Challenges

**Mohammadreza Salehi**                                          *s.salehidehnavi@uva.nl*
*University of Amsterdam*

**Hossein Mirzaei**                                          *hmirzayees@ce.sharif.edu*
*Sharif University of Technology*

**Dan Hendrycks**                                          *hendrycks@berkeley.edu*
*UC Berkeley*

**Yixuan Li**                                          *sharonli@cs.wisc.edu*
*University of Wisconsin-Madison*

**Mohammad Hossein Rohban**                                          *rohban@sharif.edu*
*Sharif University of Technology*

**Mohammad Sabokrou**                                          *sabokro@ipm.ir*
*Institute For Research In Fundamental Sciences (IPM)*

**Reviewed on OpenReview:** *https://openreview.net/forum?id=aRtjVZvbpK*

## Abstract

Machine learning models often encounter samples that are diverged from the training distribution. Failure to recognize an out-of-distribution (OOD) sample, and consequently assign that sample to an in-class label, significantly compromises the reliability of a model. The problem has gained significant attention due to its importance for safety deploying models in open-world settings. Detecting OOD samples is challenging due to the intractability of modeling all possible unknown distributions. To date, several research domains tackle the problem of detecting unfamiliar samples, including anomaly detection, novelty detection, one-class learning, open set recognition, and out-of-distribution detection. Despite having similar and shared concepts, out-of-distribution, open-set, and anomaly detection have been investigated independently. Accordingly, these research avenues have not cross-pollinated, creating research barriers. While some surveys intend to provide an overview of these approaches, they seem to only focus on a specific domain without examining the relationship between different domains. This survey aims to provide a cross-domain and comprehensive review of numerous eminent works in respective areas while identifying their commonalities. Researchers can benefit from the overview of research advances in different fields and develop future methodology synergistically. Furthermore, to the best of our knowledge, while there are surveys in anomaly detection or one-class learning, there is no comprehensive or up-to-date survey on out-of-distribution detection, which this survey covers extensively. Finally, having a unified cross-domain perspective, this study discusses and sheds light on future lines of research, intending to bring these fields closer together. All the implementations and benchmarks reported in the paper can be found at : `https://github.com/taslimisina/osr-ood-ad-methods`

# 1 Introduction

Machine learning models commonly make the closed-set assumption, where the test data is drawn *i.i.d.* from the same distribution as the training data. Yet in practice, all types of test input data—even those on which the classifier has not been trained—can be encountered. Unfortunately, models can assign misleading confidence values for unseen test samples (162; 104; 117; 183; 185). This leads to concerns about the reliability of classifiers, particularly for safety-critical applications (63; 57). In the literature, several fields attempt to address the issue of identifying the unknowns/anomalies/out-of-distribution data in the open-world setting. In particular, the problems of anomaly detection (AD), Novelty Detection (ND), One-Class Classification (OCC), Out-of-Distribution (OOD) detection, and Open-Set Recognition (OSR) have gained significant attention owing to their fundamental importance and practical relevance. They have been used for similar tasks, although the differences and connections are often overlooked.

Specifically, OSR trains a model on $K$ classes of a $N$ class training dataset; then, at the test time, the model is faced with $N$ different classes of which $N - K$ are not seen during training. OSR aims to assign correct labels to seen test-time samples while detecting unseen samples. Novelty detection or one-class classification is an extreme case of open-set recognition, in which $K$ is 1. In the multi-class classification setting, the problem of OOD detection is canonical to OSR: accurately classify in-distribution (ID) samples into the known categories and detect OOD data that is semantically different and therefore should not be predicted by the model. However, OOD detection encompasses a broader spectrum of learning tasks and solution space, which we comprehensively reviewed in this paper.

While all the aforementioned domains hold the assumption of accessing an entirely normal training dataset, anomaly detection assumes the training dataset is captured in a fully unsupervised manner without applying any filtration; therefore, it might contain some abnormal samples too. However, as abnormal events barely occur, AD methods have used this fact and proposed filtering during the training process to reach a final semantic space that fully grasps normal features. Despite previous approaches that are mostly used in object detection and image classification domains, this setting is more common in the industrial defect detection tasks, in which abnormal events are rare and normal samples have a shared concept of normality. Fig. 1 depicts an overview of the mentioned domains, in which the differences are shown visually. Note that even if there are differences in the formulation of these domains, they have so much in common, and are used interchangeably.

As an important research area, there have been several surveys in the literature (17; 124; 138; 118; 21), focusing on each domain independently or providing a very general notion of anomaly detection to cover all different types of datasets. Instead, this paper in-depth explanations for methodologies in respective areas. We make cross-domain bridges by which ideas can be easily propagated and inspire future research. For instance, the idea of using some outlier samples from different datasets to improve task-specific features is called Outlier Exposure in (60) or background modeling in (33), and is very similar to semi-supervised anomaly detection in (137). Despite the shared idea, all are considered to be novel ideas in their respective domains.

In this survey, we identify the commonalities that address different but related fields. Although some of the mentioned tasks have very close methodological setups, they differ in their testing protocols. Therefore, a comprehensive review of all the methods can better reveal their limitations in practical applications. As a key part of this survey, methods are described both mathematically and visually to give better insights to both newcomers and experienced researchers. Our survey also complements existing surveys (138; 180) which focus on higher-level categorizations. Finally, comprehensive future lines of research are provided both practically and fundamentally to not only address the issues of current methods but also shed light on critical applications of these methods in different fields.

In summary, the main contributions are as follows :

1. Identification of the relationship between different research areas that, despite being highly correlated with each other, have been examined separately.

2. Comprehensive methodological analysis of recent eminent research, and providing a clear theoretical and visual explanation for methods reviewed.

3. Performing comprehensive tests on existing baselines in order to provide a solid ground for current and future lines of research.

4. Providing plausible future lines of research and specifying some fundamental necessities of the methods that will be presented in the future such as fairness, adversarial robustness, privacy, data efficiency, and explainability.

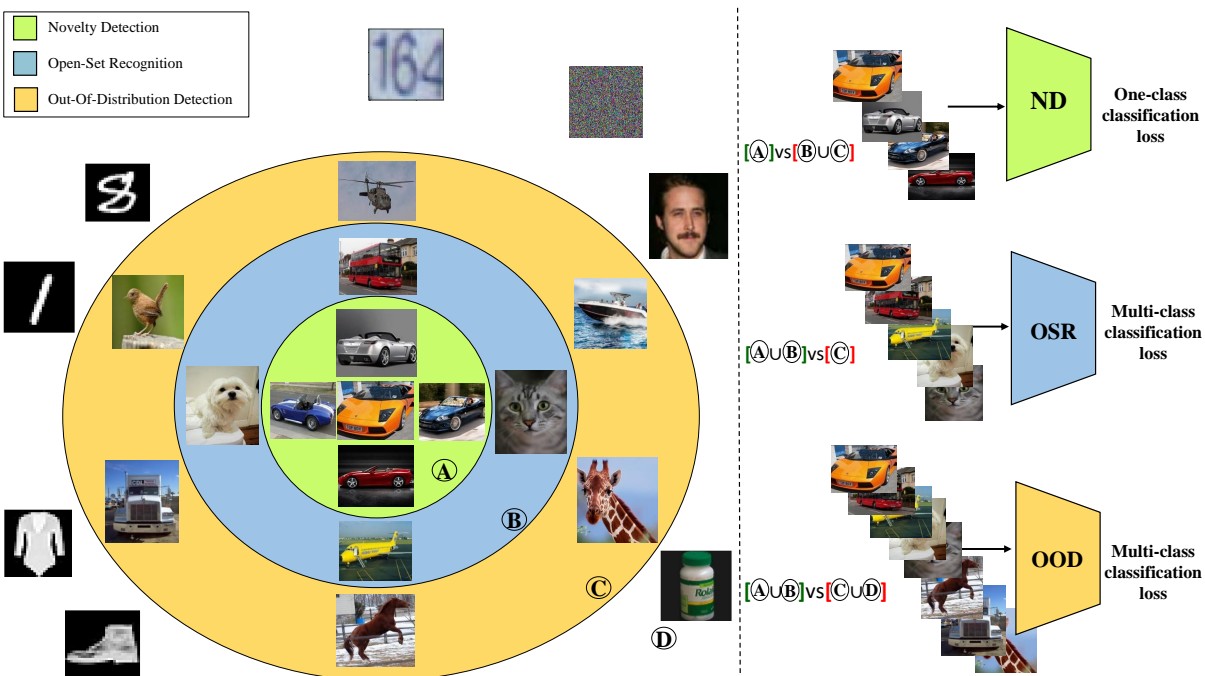

Figure 1: Problem setup for ND, OSR, and ODD from a unified perspective based on the common routine followed in the respective fields. (A), (B) and, (C) are sampled from the same training dataset, while (D) is sampled from different datasets. Typically, in ND, all training samples are deemed normal, and share a common semantic (green region), while samples diverged from such distribution are considered as anomalous. Although samples in area (D) can be regarded as potential outliers, only areas (B) and (C) are used as anomalies for the evaluation phase. In OSR, more supervision is available by accessing the label of normal samples. For example "car", "dog", "cat", "airplane" and, "bus" classes i.e, the union of (A) and (B), are considered as normal while (C) is open-set distribution(see right Figure). Same as ND, (D) is not usually considered an open-set distribution in the field, while there is no specific constraint on the type of open-set distribution in the definition of OSR domain. In OOD detection, multiple classes are considered as normal, which is quite similar to OSR. For example, (A), (B), and (C) comprise the normal training distributions, and another distribution that shows a degree of change with respect to the training distribution and is said to be out-of-distribution, which can be (D) in this case.

## 2   A Unified Categorization

Consider a dataset with training samples $(x_1, y_1), (x_2, y_2), ...$ from the joint distribution $P_{X,Y}$, where $X$ and $Y$ are random variables on an input space $\mathcal{X} = \mathbb{R}^d$ and a label (output) space $\mathcal{Y}$ respectively. In-class (also called "seen" or "normal") domain refers to the training data. In AD or ND, the label space $\mathcal{Y}$ is a binary set, indicating normal vs. abnormal. During testing, provided with an input sample $x$, the model needs to estimate $P(Y = \text{normal/seen/in-class} \mid X = x)$ in the cases of one-class setting. In OOD detection and OSR

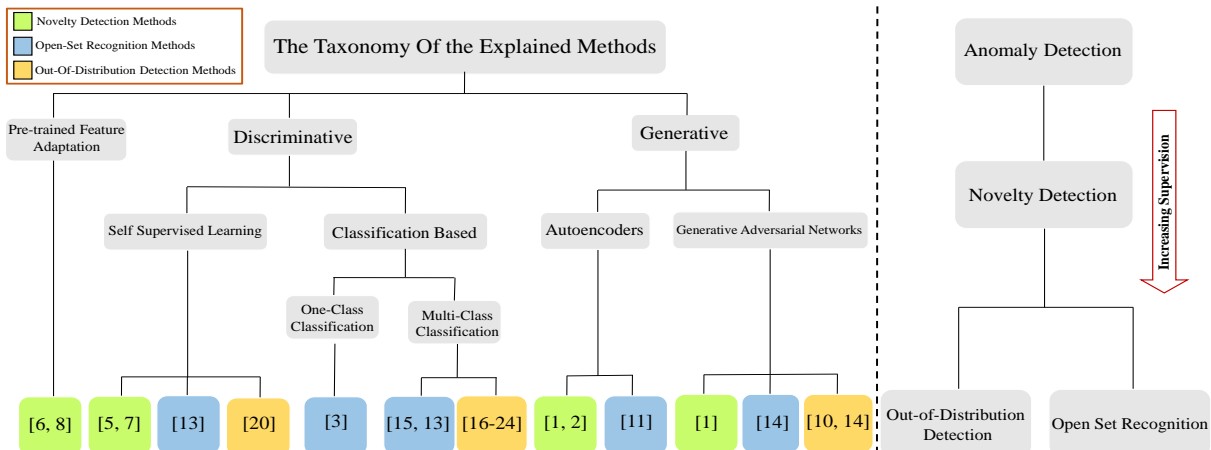

Figure 2: As discussed in section 2, all explained approaches can be unitedly classified in the shown hierarchical structure. The tree on the right points out that although some approaches have been labeled as ND, OSR, or OOD detection in the field, however can be classified in a more grounded and general form such that their knowledge could be shared synergistically. For instance, self-supervised ND methods can be added to multi-class classification approaches without harming their classification assumptions. Unfamiliar sample detection can be done by employing different levels of supervision, which is shown on the left.

for multi-class classification, the label space can contain multiple semantic categories, and the model needs to additionally perform classification on normal samples based on the posterior probability $p(Y = y \mid x)$. It is worth mentioning that in AD, input samples could contain some noises (anomalies) combined with normals; thus, the problem is converted into a noisy-label one-class classification problem; however, the overall formulation of the detection task still remains the same.

To model the conditional probabilities, the two most common and known perspectives are called **generative modeling** and **discriminative modeling**. While discriminative modeling might be easy for OOD detection or OSR settings since there is access to the labels of training samples; nevertheless, the lack of labels makes AD, ND (OCC) challenging. This is because one-class classification problems have a trivial solution to map each of their inputs regardless of being normal or anomaly to the given label $Y$ and, consequently, minimize their objective function as much as possible. This issue can be seen in some of the recent approaches such as DSVDD(136), which maps every input to a single point regardless of being normal or abnormal when it is trained for a large number of training epochs.

Some approaches (48; 12), however, have made some changes in the formulation of $P(Y \mid X)$ to solve this problem. They apply a set of affine transformations on the distribution of $X$ such that the normalized distribution does not change. Then the summation $\sum_{i=1}^{|T|} P(T_i \mid T_i(X))$ is estimated, calculating the aggregated probability of each transformation $T_i$ being applied on the input $X$ given the transformed input $T_i(X)$, which is equal to $|T|P(Y \mid X)$. This is similar to estimating $P(Y \mid X)$ directly; however, it would not collapse and can be used instead of estimating one-class conditional probability. This simple approach circumvents the problem of collapsing; however, it makes the problem dependent on the transformations since transformed inputs must not intersect with each other as much as possible to satisfy the constraint of normalized distribution consistency. Therefore, as elaborated later in the survey, OSR methods can employ AD approaches with classification models to overcome their issues. A similar situation holds for the OOD domain.

In generative modeling, AE (Autoencoder)-based, GAN (Generative Adversarial Network)-based, and explicit density estimation-based methods such as auto-regressive and flow-based models are used to model the data distribution. For AEs, there are two important assumptions.

If the auto-encoder is trained solely on normal training samples:

Table 1: The table summarizes the most notable recent deep learning-based works in different fields, specified by the informative methodological keywords and decision score used in each work. For the methods that have common parts, a shared set of keywords are used to place them on a unitary ground as much as possible. The pros and cons of each methodology are described in detail in the method explanation.

| Taxonomy | Index | Methods | Publication Venue | Decision Score | Keywords |
|---|---|---|---|---|---|
| ND | 1 | ALOCC | CVPR (2018) | reconstruction error for input | discriminative-generative, reconstruction-based, adversarial learning, denoising AutoEncoder, Refinement and Detection |
| | 3 | DeepSvdd | ICML (2018) | distance of the input to the center of the hypersphere | discriminative, extension of SVDD, compressed features, minimum volume hyper-sphere, Autoencoder |
| | 4 | GT | NeurIPS (2018) | sum of Dirichlet probabilities for different transformations of input | self-supervised learning, auxiliary task, self-labeled dataset, geometric transformations, softmax response vector |
| | 2 | Mem-AE | ICCV (2019) | reconstruction error for input | generative, reconstruction-based, Memory-based Representation, sparse addressing technique, hard shrinkage operator |
| | 5 | CSI | NeurIPS (2020) | cosine similarity to the nearest training sample multiplied by the norm of representation | self-supervised learning, contrastive learning, negative samples, shifted distribution, hard augmentations |
| | 6 | Uninformed Students | CVPR (2020) | an ensemble of student networks' variance and regression error | knowledge distillation, teacher-student based, transfer learning, self supervised learning, Descriptor Compactness |
| | 7 | CutPaste | CVPR (2021) | log-density of trained Gaussian density estimator | self-supervised Learning, generative classifiers, auxiliary task, extracting patch-level representations, cutout and scar |
| | 8 | Multi-KD | CVPR (2021) | discrepancy between the intermediate teacher and student networks' activation values | knowledge distillation, teacher-student, transfer the intermediate knowledge, pretrained network, mimicking different layers |
| OSR | 9 | OpenMax | CVPR (2016) | after computing OpenMax probability per channel maximum probability is then selected | Extreme Value Theorem, activation vector, overconfident scores, weibull distribution, Meta-Recognition |
| | 10 | OSRCI | ECCV (2018) | probability assigned to class with label $k+1$, minus maximum probability assigned to first $k$ class | generative, unknown class generation, classification based, Counterfactual Images, learning decision boundary |
| | 11 | C2AE | CVPR (2019) | minimum reconstruction error under different match vectors, compared with the predetermined threshold | generative, Reconstruction based, class conditional Autoencoder, Extreme Value Theory, feature-wise linear modulations |
| | 12 | CROSR | CVPR (2019) | after computing OpenMax probability per channel maximum probability is then selected | discriminative-generative, classification-reconstruction based , Extreme Value Theorem, hierarchical reconstruction nets |
| | 13 | GDFR | CVPR (2020) | by passing the augmented input to the classifier network, and finding maximum activation | discriminative-generative, self-supervised learning, reconstruction based, auxiliary task, geometric transformation |
| | 14 | OpenGan | ICCV (2021) | the trained discriminator utilized as an open-set likelihood function | discriminative-generative, unknown class generation, adversarially synthesized fake data, crucial to use a validation set, model selection |
| | 15 | MLS | ICLR (2021) | negation of maximum logit score | discriminative, classification based, between the closed-set and open-set, Fine-grained datasets, Semantic Shift Benchmark(SSB) |
| OOD | 16 | MSP | ICLR (2016) | Negation of maximum softmax probability | outlier detection metric, softmax distribution, maximum class probability, classification based, various data domains |
| | 17 | ODIN | ICLR (2017) | Negation of maximum softmax probability with temperature scaling | outlier detection metric, softmax distribution, temperature scaling, adding small perturbations, enhancing separability |
| | 18 | A Simple Unified etc. | NeurIPS (2018) | Mahalanobis distance to the closest class-conditional Gaussian | distance based, simple outlier detection metric, small controlled noise, class-incremental learning, Feature ensemble |
| | 19 | OE | ICLR (2018) | negation of maximum softmax probability | outlier exposure , auxiliary dataset, classification based, uniform distribution over $k$ classes, various data domains |
| | 20 | Using SSL etc. | NeurIPS (2019) | negation of maximum softmax probability | self-supervised learning, auxilary task, robustness to adversarial perturbations, MSP, geometric transformation |
| | 21 | G-ODIN | ICCV (2020) | softmax output's categorical distribution | outlier detection metric, softmax distribution, learning temperature scaling, Decomposed Confidence |
| | 22 | Energy-based OOD | NeurIPS (2020) | the energy function (a scalar is derived from the logit outputs) | outlier detection metric, hyperparameter-free, The *Helmholtz free energy*, energy-bounded learning, the Gibbs distribution |
| | 23 | MOS | CVPR (2021) | lowest `others` score among all groups | softmax distribution, group-based Learning, large-scale images, category `others`, pre-trained backbone |
| | 24 | ReAct | NeurIPS (2021) | after applying ReAct different OOD detection methods could be used. they default to using the energy score. | activation truncation, reducing model overconfidence, compatible with different OOD scoring functions |

- They **would** be able to reconstruct unseen normal test-time samples as precisely as training-time ones.

- They **would not** be able to reconstruct unseen abnormal test-time samples as precisely as normal inputs.

Nevertheless, recently proposed AE-based methods show that the above-mentioned assumptions are not always true (143; 49; 189). For instance, even if AE can reconstruct normal samples perfectly, nonetheless with only a one-pixel shift, reconstruction loss will be high.

Likewise, GANs as another famous model family, are widely used for AD, ND, OCC, OSR, and OOD detection. If a GAN is trained on fully normal training samples, it operates on the following assumptions:

- If the input is **normal** then there **is** a latent vector that, if generated, has a low discrepancy with the input.

- If the input is **abnormal** then there **is not** a latent vector that, if generated, has a low discrepancy with the input.

Here, the discrepancy can be defined based on the pixel-level MSE loss of the generated image and test-time input or more complex functions such as layer-wise distance between the discriminator's features when fed a generated image and test-time input. Although GANs have proved their ability to capture semantic abstractions of a given training dataset, they suffer from mode-collapse, unstable training process, and irreproducible result problems (3).

Finally, auto-regressive and flow-based models can be used to explicitly approximate the data density and detect abnormal samples based on their assigned likelihood. Intuitively, normal samples must have higher likelihoods compared to abnormal ones; however, as will be discussed later, auto-regressive models assign an even higher likelihood to abnormal samples despite them not being seen in the training phase, which results in a weak performance in AD, ND, OSR, and OOD detection. Solutions to this problem will be reviewed in subsequent sections. To address this issue, several remedies have been proposed in the OOD detection domain, which can be used in OSR, AD, and ND; however, considering the fact that OOD detection's prevalent testing protocols might be quite different from other domains such as AD or ND, more evaluations on their reliability are needed. As lots of works with similar ideas or intentions have been proposed in each domain, we mainly focus on providing a detailed yet concise summary of each and highlighting the similarities in the explanations. In Table 1 and Fig. 2, a summary of the most notable papers reviewed in this work is presented along with specifications on their contributions with respect to the methodological categorization provided in this paper.

## 3 Anomaly and Novelty Detection

Anomaly Detection (AD) and Novelty Detection (ND) have been used interchangeably in the literature, with few works addressing the differences (122; 176; 172). There are specific inherent challenges with anomaly detection that contradict the premise that the training data includes entirely normal samples. In physical investigations, for example, measurement noise is unavoidable; as a result, algorithms in an unsupervised training process must automatically detect and focus on the normal samples. However, this is not the case for novelty detection problems. There are many applications in which providing a clean dataset with minimum supervision is an easy task. While these domains have been separated over time, their names are still not appropriately used in literature. These are reviewed here in a shared section.

Interests in anomaly detection go back to 1969 (52), which defines anomaly/outlier as **"samples that appear to deviate markedly from other members of the sample in which it occurs"**, explicitly assuming the existence of an underlying shared pattern that a large fraction of training samples follow. This definition has some ambiguity. For example, one should define a criterion for the concept of deviation or make the term "markedly" more quantitative.

To this end, there has been a great effort both before and after the advent of deep learning methods to make the mentioned concepts more clear. To find a sample that deviates from the trend, adopting an appropriate distance metric is necessary. For instance, deviation could be computed in a raw pixel-level input or in a semantic space that is learned through a deep neural network. Some samples might have a low deviation from others in the raw pixel space but exhibit large deviations in representation space. Therefore, choosing the right distance measure for a hypothetical space is another challenge. Finally, the last challenge is choosing the threshold to determine whether the deviation from normal samples is significant.

### 3.1 Non-Deep Learning-Based Approaches

In this section, some of the most notable traditional methods used to tackle the problem of anomaly detection are explained. All other sections are based on the new architectures that are mainly based on deep learning.

### 3.1.1 Isolation Forests (94):

Like Random Forests, Isolation Forests (IF) are constructed using decision trees. It is also an unsupervised model because there are no predefined labels. Isolation Forests were created with the notion that anomalies are "few and distinct" data points. An Isolation Forest analyzes randomly sub-sampled data in a tree structure using attributes chosen at random. As further-reaching samples required more cuts to separate, they are less

likely to be outliers. Similarly, samples that end up in shorter branches exhibit anomalies because the tree could distinguish them from other data more easily. Fig. 3 depicts the overall procedure of this work.

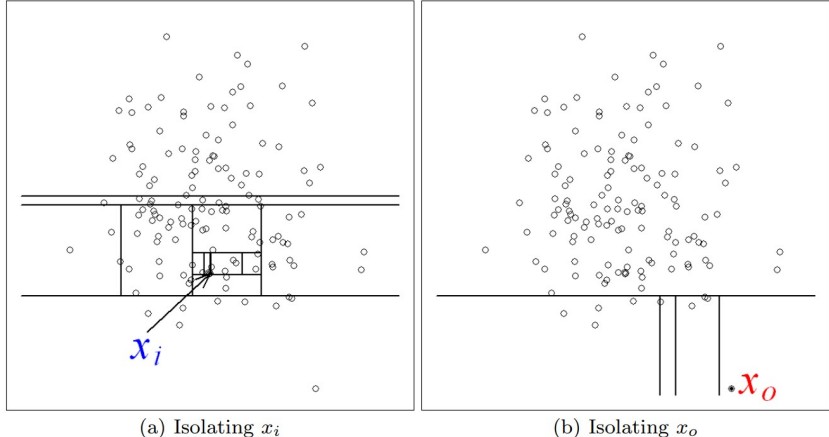

(a) Isolating $x_i$                           (b) Isolating $x_o$

Figure 3: An overview of the isolation forest method is shown. Anomalies are more susceptible to isolation and hence have short path lengths. A normal point $x_i$ requires twelve random partitions to be isolated compared to an anomaly $x_o$ that requires only four partitions to be isolated. The figure is taken from (94).

### 3.1.2 DBSCAN (40):

Given a set of points in a space, DBSCAN combines together points that are densely packed together (points with numerous nearby neighbors), identifying as outliers those that are isolated in low-density regions. The steps of the DBSCAN algorithm are as follows: (1) Identify the core points with more than the minimum number of neighbors required to produce a dense zone. (2) Find related components of core points on the neighbor graph, ignoring all non-core points.(3) If the cluster is a neighbor, assign each non-core point to it, otherwise assign it to noise. Fig. 4 depicts the overall procedure of this work.

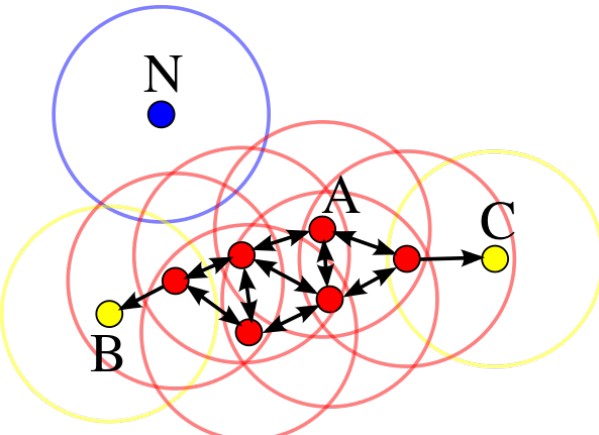

Figure 4: The minimal number of points required to generate a dense zone in this figure is four. Since the area surrounding these points in a radius contains at least four points, Point A and the other red points are core points (including the point itself). They constitute a single cluster because they are all reachable from one another. B and C are not core points, but they may be reached from A (through other core points) and hence are included in the cluster. Point N is a noise point that is neither a core nor reachable directly. The figure is taken from (40).

### 3.1.3 LOF: Identifying Density-Based Local Outliers (16):

The local outlier factor is based on the concept of a local density, in which locality is determined by the distance between K nearest neighbors. One can detect regions of similar density and points that have a significantly lower density than their neighbors by comparing the local density of an object to the local densities of its neighbors. These are classified as outliers. The typical distance at which a point can be "reached" from its neighbors is used to estimate the local density. The LOF notion of "reachability distance" is an extra criterion for producing more stable cluster outcomes. The notions of "core distance" and "reachability distance", which are utilized for local density estimation, are shared by LOF and DBSCAN.

### 3.1.4 OC-SVM (148):

Primary AD methods would use statistical approaches such as comparing each sample with the mean of the training dataset to detect abnormal inputs, which imposes an ungeneralizable and implicit Gaussian distribution assumption on the training dataset. In order to reduce the number of presumptions or relieve the mentioned deficiencies of traditional statistical methods, OC-SVM (148) was proposed. As the name implies, OC-SVM is a one-class SVM maximizing the distance of training samples from the origin using a hyper-plane, including samples on one side and the origin on its other side. Eq. 1 shows the primal form of OC-SVM that attempts to find a space in which training samples lie on just one side, and the more distance of origin to the line, the better the solution to the optimization problem. Each input data is specified by $x_i$, $\Phi$ is a feature extractor, $\xi_i$ is the slack variable for each sample, $w$ and $\rho$ characterize the hyperplane. The parameter $v$ sets an upper bound on the fraction of outliers and also is a lower bound on the number of training examples used as the support vector.

$$
\begin{aligned}
\min_{w,\rho,\xi_i} \frac{1}{2} \|w\|^2 + \frac{1}{vn} \sum_1^n \xi_i - \rho \\
\text{subject to} \quad (w \cdot \Phi(x_i)) \geqslant \rho - \xi_i, i \in 1, ..., n \\
\xi_i \geqslant 0, i \in 1, \ldots, n
\end{aligned}
\tag{1}
$$

Finding a hyper-plane appears to be a brilliant approach for imposing a shared normal restriction on the training samples. But unfortunately, because half-space is not compact enough to grasp unique shared normal features, it generates a large number of false negatives. Therefore, similarly, Support Vector Data Description(SVDD) tries to find the most compact hyper-sphere that includes normal training samples. This is much tighter than a hyper-plan, thus finds richer normal features, and is more robust against unwanted noises as Eq. 2 shows. Both of the mentioned methods offer soft and hard margin settings, and in the former, despite the latter, some samples can cross the border and remain outside of it even if they are normal. OC-SVM has some implicit assumptions too; for example, it assumes training samples obey a shared concept, which is conceivable due to the one-class setting. Also, it works pretty well on the AD setting in which the number of outliers is significantly lower than normal ones however fails on high dimensional datasets. The resulting hyper-sphere is characterized by $a$ and $R$ as the center and radius, respectively. Also, $C$ controls the trade-off between the sphere compactness and average slack values.

$$
\begin{aligned}
\min_{R,a} C \sum_{i=1}^n \xi_i + R^2 \\
\text{subject to} \quad \|\phi(x_i) - a\|^2 \leqslant R^2 + \xi_i, i \in 1, \ldots, n \\
\xi_i \geqslant 0, i \in 1, \ldots, n
\end{aligned}
\tag{2}
$$

### 3.2 Anomaly Detection With Robust Deep Auto-encoders (195):

This work trains an AutoEncoder (AE) on a dataset containing both inliers and outliers. The outliers are detected and filtered during training, under the assumption that inliers are significantly more frequent and

have a shared normal concept. This way, the AE is trained only on normal training samples and consequently poorly reconstructs abnormal test time inputs. The following objective function is used:

$$\min_{\theta} ||L_D - D_\theta(E_\theta(L_D))||_2 + \lambda||S||_1$$
$$\text{s.t.} \quad X - L_D - S = 0, \tag{3}$$

where E and D are encoder and decoder networks, respectively. $L_D$ is the inlier part, $S$ is the outlier part of the training data $X$ and $\lambda$ is a parameter that tunes the level of sparsity in $S$. However, the above optimization is not easily solvable since $S$ and $\theta$ need to be optimized jointly. To address this issue, the Alternating Direction Method of Multipliers (ADMM) is used, which divides the objective into two (or more) pieces. At the first step, by fixing $S$, an optimization problem on the parameters $\theta$ is solved such that $L_D = X - S$, and the objective is $||L_D - D_\theta(E_\theta(L_D))||_2$. Then by setting $L_D$ to be the reconstruction of the trained AE, the optimization problem on its norm is solved when $S$ is set to be $X - L_D$. Since the $L_1$ norm is not differentiable, a proximal operator is employed as an approximation of each of the optimization steps as follows:

$$\text{prox}_{\lambda, L_1}(x_i) = \begin{cases} x_i - \lambda & x_i > \lambda \\ x_i + \lambda & x_i < -\lambda \\ 0 & -\lambda \leq x_i \leq \lambda \end{cases} \tag{4}$$

Such a function is known as a *shrinkage operator* and is quite common in $L_1$ optimization problems. The mentioned objective function with $||S||_1$ separates only unstructured noises, for instance, Gaussian noise on training samples, from the normal content in the training dataset. To separate structured noises such as samples that convey completely different meaning compared to the majority of training samples, $L_{2,1}$ optimization norm can be applied as:

$$\sum_{j=1}^{n} ||x_j||_2 = \sum_{j=1}^{n} (\sum_{i=1}^{m} |x_{ij}|^2)^{1/2}, \tag{5}$$

with a proximal operator that is called block-wise soft-thresholding function (108). At the test time, the reconstruction error is employed to reject abnormal inputs.

### 3.3 Adversarially Learned One-Class Classifier for Novelty Detection (ALOCC) (139):

In this work, it is assumed that a fully normal training sample is given, and the goal is to train a novelty detection model using them. At first, $(\mathcal{R})$ as a Denoising Auto Encoder (DAE) is trained to (1) decrease the reconstruction loss and (2) fool a discriminator in a GAN-based setting. This helps the DAE to produce high-quality images instead of blurred outputs (84). This happens because, on the one hand, the AE model loss explicitly assumes independent Gaussian distributions for each of the pixels. And on the other hand, true distributions of pixels are usually multi-modal, which forces the means of Gaussians to settle in between different modes. Therefore, they produce blurry images for complex datasets. To address this issue, AEs can be trained in a GAN-based framework to force the mean of each of Gaussians to capture just one mode of its corresponding true distribution. Moreover, by using the discriminator's output $(\mathcal{D})$ instead of the pixel-level loss, normal samples that are not properly reconstructed can be detected as normal. This loss reduces the

False Positive Rate (FPR) of the vanilla DAE significantly. The objective function is as follows:

$$\mathcal{L}_{\mathcal{R}+\mathcal{D}} = \min_{\mathcal{R}} \max_{\mathcal{D}} \left( \mathbb{E}_{X \sim p_t}[\log(\mathcal{D}(X))] + \right.$$

$$\left. \mathbb{E}_{\widetilde{X} \sim p_t + \mathcal{N}_\sigma}([\log(1 - \mathcal{D}(\mathcal{R}(\widetilde{X})))]) \right) \tag{6}$$

$$\mathcal{L}_{\mathcal{R}} = ||X - X'||^2$$

$$\mathcal{L} = \mathcal{L}_{\mathcal{R}+\mathcal{D}} + \mathcal{L}_{\mathcal{R}},$$

where $X$ is the input, $X'$ is the reconstructed output of the decoder, and $p_t$ is the distribution of the target class (i.e., normal class). This helps the model to not only have the functionality of AEs for anomaly detection but also produces higher quality outputs. Furthermore, detection can be done based on $\mathcal{D}(\mathcal{R}(X))$ as mentioned above. Fig. 5 depicts the overall architecture of this work.

An extended version of ALOCC, where the $\mathcal{R}(X)$ network has been formed as a Variational AE is presented in (141). Besides, the ALOCC can not process the entire input data (images or video frames) at one step and needs the test samples to divide into several patches. Processing the patches makes the method computationally expensive. To address this problem, AVID (140) was proposed to exploit a fully convolutional network as a discriminator (i.e., $\mathcal{D}$) to score (and hence detect) all abnormal regions in the input frame/image all at once.

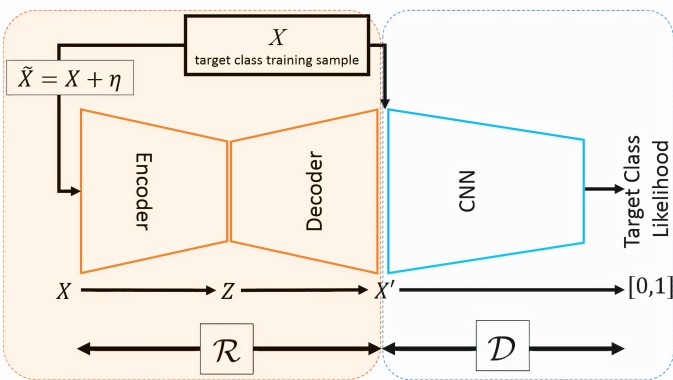

Figure 5: An overview of the ALOCC method is shown. This work trains an autoencoder that fools a discriminator in a GAN-based setting. This helps the AE to make high-quality images instead of blurred outputs. Besides, by using the discriminator's output, a more semantic similarity loss is employed instead of the pixel-level $L_2$ reconstruction loss. The figure is taken from (139).

### 3.4 One-Class Novelty Detection Using GANs With Constrained Latent Representations (OC-GAN) (122):

As one of the challenges, AE trained on entirely normal training samples could reconstruct unseen abnormal inputs with even lower errors. To solve this issue, this work attempts to make the latent distribution of the encoder $(\mathrm{EN}(\cdot))$ similar to the uniform distribution in an adversarial manner:

$$l_{\text{latent}} = -(\mathbb{E}_{s \sim \mathbb{U}(-1,1)}[\log(D_l(s))] + \mathbb{E}_{x \sim p_x}[\log(1 - D_l(\mathrm{EN}(x+n)))]), \tag{7}$$

where $n \sim N(0, 0.2)$, $x$ is an input image, $p_x$ is the distribution of in-class examples, and $D_l$ is the latent discriminator. The discriminator forces the encoder to produce uniform distribution on the latent space. Similarly, the decoder $(\mathrm{De}(\cdot))$ is forced to reconstruct in-class outputs for any latent value sampled from the uniform distribution as follows :

$$l_{\text{visual}} \quad = -(\mathbb{E}_{s \sim \mathbb{U}(-1,1)}[\log(\mathcal{D}_v(\text{De}(s)))]+ \\ \mathbb{E}_{x \sim p_l}[\log(1 - D_v(x))]),$$

$$(8)$$

where $D_v$ is called visual discriminator.

Intuitively, the learning objective distributes normal features in the latent space such that the reconstructed outputs entirely or at least roughly resemble the normal class for both normal and abnormal inputs. Another technique called *informative negative sample mining* is also employed on the latent space to actively seek regions that produce images of poor quality. To do so, a classifier is trained to discriminate between the reconstructed outputs of the decoder and fake images, which are generated from randomly sampled latent vectors. To find informative negative samples, the algorithm (see Fig. 6) starts with a random latent-space sample and uses the classifier to assess the quality of the generated image. After that, the algorithm solves an optimization in the latent space to make the generated image such that the discriminator detects it as fake. Finally, the negative sample is used to boost the training process.

As in previous AE-based methods, reconstruction loss is employed in combination with the objective above. Reconstruction error is used as the test time anomaly score.

**Input** : Set of training data $x$, iteration size $N$,
           parameter $\lambda$
**Output:** Models: En, De, C, $D_l$ , $D_v$
**for** *iteration 1* **to** $\to N$ **do**
   *Classifier update: keep $D_l$, $D_v$, En, De fixed.*
   $n \longleftarrow \mathcal{N}(0, I)$
   $l_1 \longleftarrow En(x + n)$
   $l_2 \longleftarrow \mathbb{U}(-1, 1)$
   $l_{classifier} \longleftarrow C(De(l_2), 0) + C(De(l_1), 1)$
   *Back-propagate$l_{classifier}$ to change C*

   *Discriminator update:*
   $l_{latent} \longleftarrow D_l(l_1, 0) + D_l(l_2, 1)$
   $l_{visual} \longleftarrow D_v(De(l_2), 0) + D_v(x, 1)$
   *Back-propagate$l_{latent}$ +*
     *$l_{visual}$ and change $D_l, D_v$*

   *Informative-negative mining : Keep all*
     *networks fixed.*
   **for** *sub-iteration 1* **to** $\to 5$ **do**
      $l_{classifier} \longleftarrow C(De(l_2), 1)$
      *Back-propagate $l_{classifier}$ to change $l_2$*
   **end**

   *Generator update: keep $D_l, D_v, C$ fixed.*
   $l_{latent} \longleftarrow D_l(l_1, 1) + D_l(l_2, 0)$
   $l_{visual} \longleftarrow D_v(De(l_2), 1) + D_v(x, 0)$
   $l_{mse} \longleftarrow ||x - De(l_1)||^2$
   *Back-propagate*
     *$l_{latent} + l_{visual} + \lambda l_{mse}$ to change En, De*
**end**

Figure 6: The training process of OC-GAN method (122).

## 3.5 Latent Space Autoregression for Novelty Detection (LSA) (1):

This work proposes a concept called "surprise" for novelty detection, which specifies the uniqueness of input samples in the latent space. The more unique a sample is, the less likelihood it has in the latent space, and subsequently, the more likely it is to be an abnormal sample. This is beneficial, especially when there are many similar normal training samples in the training dataset. To minimize the MSE error for visually identical training samples, AEs often learn to reconstruct their average as the outputs. This can result in fuzzy results and large reconstruction errors for such inputs. Instead, by using the surprise loss in combination with the reconstruction error, the issue is alleviated. Besides, abnormal samples are usually more surprising, and this increases their novelty score. Surprise score is learned using an auto-regressive model on the latent space, as Fig. 7 shows. The auto-regressive model ($h$) can be instantiated from different architectures, such as LSTM and RNN networks, to more complex ones. Also, similar to other AE-based methods, the reconstruction error is optimized. The overall objective function is as follows:

$$\mathcal{L} = \mathcal{L}_{\text{Rec}}(\theta_E, \theta_D) + \lambda \cdot \mathcal{L}_{\text{LLK}}(\theta_E, \theta_h) \\ = \mathbb{E}_X \left[ ||x - \hat{x}||^2 - \lambda \cdot \log(h(z; \theta_h)) \right], \ z = f(x, \theta_E),$$

$$(9)$$

where $\mathcal{L}_{\text{Rec}}$ and $\mathcal{L}_{\text{LLK}}$ are the reconstruction and surprise terms. The parameters of the encoder, decoder, and probabilistic model are specified by $\theta_E$, $\theta_D$ and $\theta_h$, respectively. $x$ is the input, $f$ is the encoder, and $\lambda$ controls the weight of $\mathcal{L}_{\text{LLK}}$.

## 3.6 Memory-Augmented Deep Autoencoder for Unsupervised Anomaly Detection (Mem-AE) (49):

This work challenges the second assumption behind using AEs. It shows that some abnormal samples could be perfectly reconstructed even when there are not any of them in the training dataset. Intuitively, AEs

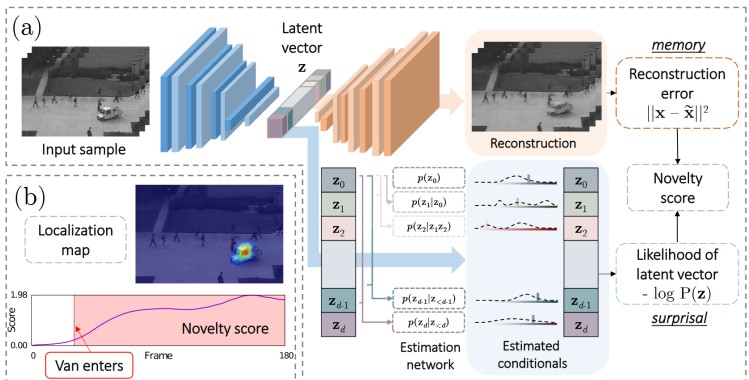

Figure 7: An overview of the LSA method is shown. A surprise score is defined based on the probability distribution of embeddings. The probability distribution is learned using an auto-regressive model. Also, reconstruction error is simultaneously optimized on normal training samples. The figure is taken from (1).

may not learn to extract uniquely describing features of normal samples; as a result, they may extract some abnormal features from abnormal inputs and reconstruct them perfectly. This motivates the need for learning features that allow only normal samples to be reconstructed accurately. To do so, Mem-AE employs a memory that stores unique and sufficient features of normal training samples. During training, the encoder *implicitly* plays the role of a *memory address generator*. The encoder produces an embedding, and memory features that are similar to the generated embedding are combined. The combined embedding is then passed to a decoder to make the corresponding reconstructed output. Also, Mem-AE uses a sparse addressing technique that uses only a small number of memory items. Accordingly, the decoder in Mem-AE is restricted to performing the reconstruction using a small number of addressed memory items, rendering the requirement for efficient utilization of the memory items. Furthermore, the reconstruction error forces the memory to record prototypical patterns that are representative of the normal inputs. To summarize the training process, Mem-AE (1) finds address $\text{Enc}(x) = z$ from the encoder's output; (2) measures the cosine similarity $d(z, m_i)$ of the encoder output $z$ with each of the memory elements $m_i$; (3) computes attention weights $\mathbf{w}$, where each of its elements is computed as follows:

$$w_i = \frac{\exp(d(z, m_i))}{\sum_{j=1}^{N} \exp(d(z, m_j))}; \tag{10}$$

(4) applies address shrinkage techniques to ensure sparsity:

$$\hat{w}_i = \frac{\max(w_i - \lambda, 0).w_i}{|w_i - \lambda| + \epsilon} \tag{11}$$

$$E(\hat{\mathbf{w}}^t) = \sum_{i=1}^{T} -\hat{w}_i . \log(\hat{w}_i), \tag{12}$$

and finally, the loss function is defined as (13), where $R$ is the reconstruction error and is used as the test time anomaly score. Fig. 8 shows the overview of architecture.

$$L(\theta_e, \theta_d, M) = \frac{1}{T} \sum_{t=1}^{T} \left( R(\mathbf{x}^t, \hat{\mathbf{x}}^t) + \alpha E(\hat{\mathbf{w}}^t) \right) \tag{13}$$

where $\theta_e$ and $\theta_d$ denote the parameters of the encoder and decoder. $\mathbf{x}^t$ denotes training samples and $\hat{\mathbf{x}}^t$ denote the reconstructed sample corresponding the each train-

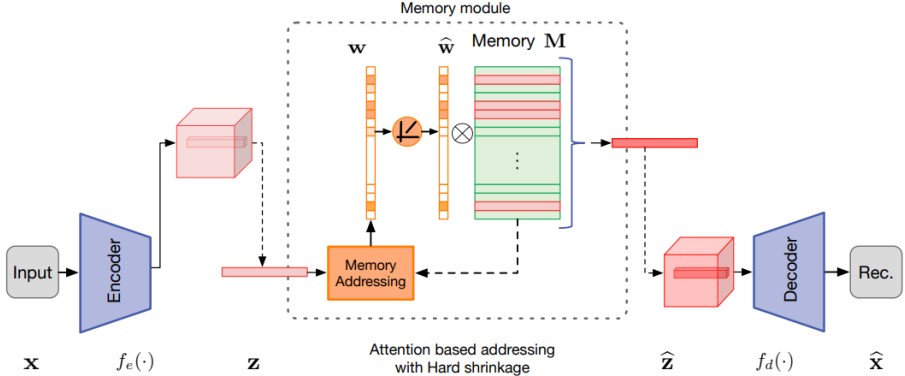

Figure 8: An overview of the Mem-AE method is shown. Each sample is passed through the encoder, and a latent embedding $z$ is extracted. Then using the cosine similarity, some nearest learned normal features are selected from the memory, and the embedding $\hat{z}$ is made as their weighted average. At last, the reconstruction error of the decoded $\hat{z}$ and input is considered as the novelty score. The figure is taken from (49).

### 3.7 Redefining the Adversarially Learned One-Class Classifier Training Paradigm (Old-is-Gold) (189):

This work extends the idea of ALOCC (139). As ALOCC is trained in a GAN-based setting, it suffers from stability and convergence issues. On one hand, the over-training of ALOCC can confuse the discriminator $D$ because of the realistically generated fake data. On the other hand, under-training detriments the usability of discriminator features. To address this issue, a two-phase training process is proposed. In the first phase, a similar training process as ALOCC is followed :

$$\mathcal{L} = \mathcal{L}_{\mathcal{R}+\mathcal{D}} + \mathcal{L}_{\mathcal{R}} \tag{14}$$

As phase one progresses, a low-epoch generator model $\mathcal{G}^{\text{old}}$ for later use in phase two of the training is saved. The sensitivity of the training process to the variations of the selected epoch is discussed in the paper.

During the second phase, samples $\hat{X} = \mathcal{G}$ are considered high-quality reconstructed data. Samples $\hat{X}^{\text{low}} = \mathcal{G}^{\text{old}}(X)$ are considered as low-quality samples. Then, pseudo anomaly samples are created as follows:

$$\hat{\bar{X}} = \frac{\mathcal{G}^{\text{old}}(X_i) + \mathcal{G}^{\text{old}}(X_j)}{2}$$
$$\hat{X}^{\text{pseudo}} = \mathcal{G}(\hat{\bar{X}}) \tag{15}$$

After that, the discriminator is trained again to strengthen its features by distinguishing between good quality samples such as $\{X, \hat{X}\}$ and low-quality or pseudo anomaly ones such as $\{\hat{X}^{\text{low}}, \hat{X}^{\text{pseudo}}\}$ as follows:

$$\begin{aligned}
\max_{\mathcal{D}} \Big( & \alpha \cdot \mathbb{E}_{\text{X}}[\log(1 - D(X))] + (1 - \alpha) \cdot \mathbb{E}_{\hat{X}}[\log(1 - D(\hat{X}))] \\
& + (\beta \cdot \mathbb{E}_{\hat{X}^{\text{low}}}[\log(1 - D(\hat{X}^{\text{low}}))] \\
& + (1 - \beta) \cdot \mathbb{E}_{\hat{X}^{\text{pseudo}}}[\log(1 - D(\hat{X}^{\text{pseudo}}))] \Big)
\end{aligned} \tag{16}$$

In this way, $D$ does not collapse as in ALOCC, and $\mathcal{D}(\mathcal{G}(X))$ is used as the test time criterion. Fig. 9 shows the overall architecture of this method.

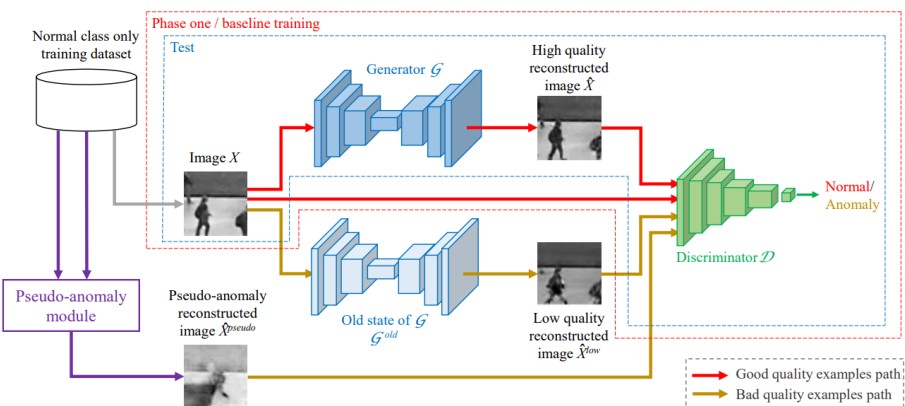

Figure 9: An overview of the Old-Is-Gold method is shown. The architecture is similar to ALOCC; however, it saves the weights of the network $G$ during the training process, which are called $G^{old}$. Then, $G^{old}$ is employed to make some low-quality samples that the discriminator is expected to distinguish as normal. Also, some pseudo abnormal samples are generated by averaging each pair of normal low-quality samples, which the discriminator is supposed to distinguish as fake inputs. This makes the discriminator's features rich and stabilizes the training process. The figure is taken from (189).

## 3.8   Adversarial Mirrored Autoencoder (AMA) (159):

The overall architecture of AMA is similar to ALOCC. However, it challenges the first assumption of AEs. It has been shown that $l_p$ norms are not suitable for training AEs in the anomaly detection domain since they cause blurry reconstructions and subsequently increase the error of normal samples. To address this problem, AMA proposes to minimize the Wasserstein distance between joint distributions $\mathbb{P}_{X,X}$ and $\mathbb{P}_{X,\hat{X}}$. The objective function is as follows:

$$W(\mathbb{P}_{X,X}, \mathbb{P}_{X,\hat{X}}) = \max_{\mathcal{D} \in Lip-1} \mathbb{E}_{x \sim \mathbb{P}_X}[\mathcal{D}(X,X) - \mathcal{D}(X,\hat{X})] \tag{17}$$

where $\hat{X}$ is the reconstructed image.

(15) showed that by forcing the linear combination of latent codes of a pair of data points to look realistic after decoding, the encoder learns a better representation of data. Inspired by this, AMA makes use of $\hat{X}_{\text{inter}}$—obtained by decoding the linear combination of some randomly sampled inputs:

$$\min_G \max_{D \in Lip-1} \mathbb{E}_{x \sim \mathbb{P}_X}[\mathcal{D}(X,X) - \mathcal{D}(X,\hat{X}_{\text{inter}})] \tag{18}$$

To further boost discriminative abilities of $D$, inspired by (27), normal samples are supposed as residing in the typical set (28) while anomalies reside outside of it, a Gaussian regularization equal to $L_2$ norm is imposed on the latent representation. Then using the Gaussian Annulus Theorem (170) stating that in a d-dimensional space, the typical set resides with high probability at a distance of $\sqrt{d}$ from the origin, synthetic negatives (anomalies), are obtained by sampling the latent space outside and closer to the typical set boundaries. Therefore, the final objective function is defined as follows:

$$\min_{G} \max_{D \in Lip-1} \mathcal{L}_{\text{normal}} - \lambda_{\text{neg}} \cdot \mathcal{L}_{\text{neg}}$$

$$\mathcal{L}_{\text{normal}} = \mathbb{E}_{x \sim \mathbb{P}_X} \Big[ D(X, X) - D(X, \hat{X})$$

$$+ \lambda_{\text{inter}} \cdot (D(X, X) - D(X, \hat{X}_{\text{inter}})) + \lambda_{\text{reg}} \cdot \|E(X)\| \Big]$$

$$\mathcal{L}_{\text{neg}} = \mathbb{E}_{x \sim \widetilde{\mathbb{Q}}_X} \Big[ D(X, X) - D(X, \hat{X}_{\text{neg}}) \Big]$$

$$(19)$$

Fig. 10 shows an overview of the method and the test time criterion is $||f(X, X) - f(X, \text{G}(\text{E}(X)))||_1$ where $f$ is the penultimate layer of $D$.

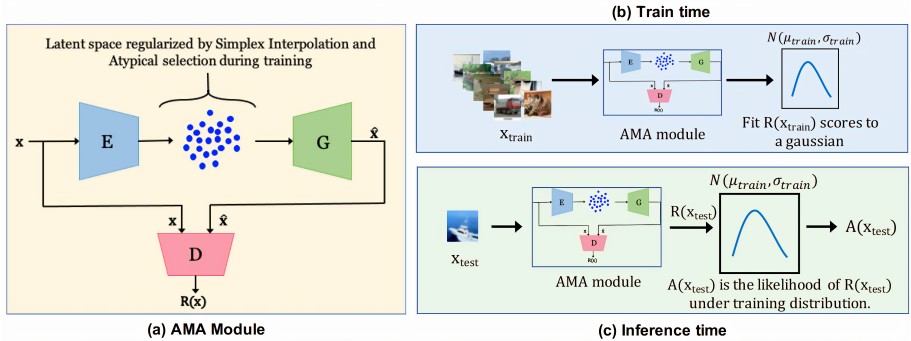

Figure 10: An overview of AMA method is shown. The architecture is similar to ALOCC; however, it does not try to minimize reconstruction error between $x$ and $\hat{x}$. Instead, a discriminator is trained with Wasserstein Loss to minimize the distance between the distribution $(x, x)$ and $(x, \hat{x})$. In this way, it forces $\hat{x}$ to be similar to $x$ without using any $l_p$ norm that causes blurred reconstructed outputs. Moreover, some negative latent vectors are sampled from low probability areas, which the discriminator is supposed to distinguish as fake ones. This makes the latent space consistent compared to the previous similar approaches. The figure is taken from (159).

### 3.9 Unsupervised Anomaly Detection with Generative Adversarial Networks to Guide Marker Discovery (AnoGAN) (147):

This work trains a GAN on normal training samples, then at the test time, solves an optimization problem that attempts to find the best latent space $z$ by minimizing a discrepancy. The discrepancy is found by using a pixel-level loss of the generated image and input in combination with the loss of discriminator's features at different layers when the generated and input images are fed. Intuitively, for normal test time samples, a desired latent vector can be found despite abnormal ones. Fig. 11 shows the structure of the method.

As inferring the desired latent vector through solving an optimization problem is time-consuming, some extensions of AnoGAN have been proposed. For instance, Efficient-GAN (191) tries to substitute the optimization problem by training an encoder $E$ such that the latent vectors $z'$ approximate the distribution of $z$. In this way, $E$ is used to produce the desired latent vector, significantly improving test time speed. Fig. 12 shows the differences. The following optimization problem (20) is solved at the test time to find $z$. The anomaly score is assigned based on how well the found latent vector can minimize the objective function.

$$\min_{z} \quad (1 - \lambda) \cdot \sum |x - G(z)| + \lambda \cdot \sum |D(x) - D(G(z))| \tag{20}$$

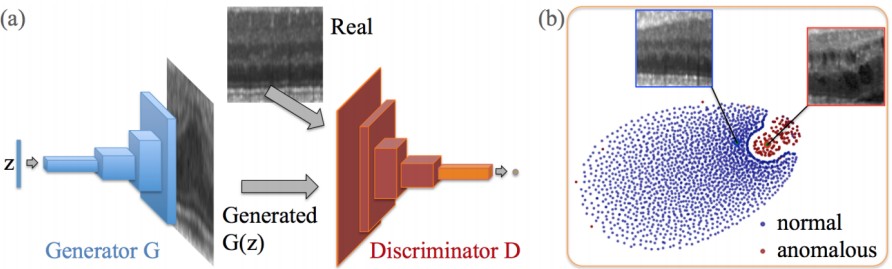

Figure 11: An overview of AnoGAN method. At first, a generator network $G$ and a discriminator $D$ are trained jointly on normal training samples using the standard training loss, which yields a semantic latent representation space. Then at the test time, an optimization problem that seeks to find an optimal latent embedding $z$ that mimics the pixel-level and semantic-level information of the input is solved. Intuitively, for normal samples, a good approximation of inputs can be found in the latent space; however, it is not approximated well for abnormal inputs. The figure is taken from (147).

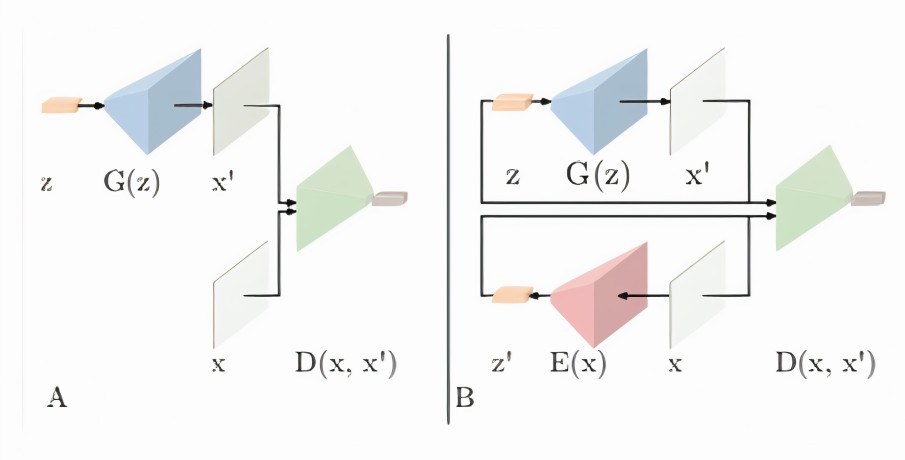

Figure 12: Figure A shows the architecture of AnoGAN. Figure B shows the architecture of Efficient-GAN. The encoder $E$ mimics the distribution of the latent variable $z$. The discriminator $D$ learns to distinguish between joint distribution of $(x, z)$ instead of $x$. The figure is taken from (147).

### 3.10 Deep One-Class Classification (DeepSVDD) (136):

This method can be seen as an extension of SVDD using a deep network. It assumes the existence of shared features between training samples, and tries to find a latent space in which training samples can be compressed into a minimum volume hyper-sphere surrounding them. Fig. 13 shows the overall architecture. The difference $w.r.t$ traditional methods is the automatic learning of kernel function $\phi$ by optimizing the parameters $W$. To find the center $c$ of the hyper-sphere, an AE is first trained on the normal training samples, then the average of normal sample latent embeddings is considered as $c$. After that, by discarding the decoder part, the encoder is trained using the following objective function:

$$\min_{W} \frac{1}{n} \sum_{i=1}^{n} ||\phi(x_i; W) - c||_2^2 + \frac{\lambda}{2} \sum_{l=1}^{L} ||W^l||_F^2, \tag{21}$$

where $W^l$ shows the weights of encoder's $l$th layer. At the test time, anomaly score is computed based on $||\phi(x; W^*) - c||^2$ where $W^*$ denotes trained parameters.

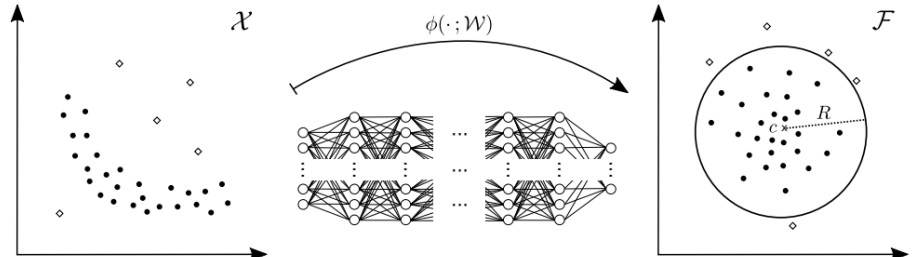

Figure 13: An overview of the DSVDD method is shown. It finds a minimum volume hyper-sphere that contains all training samples. As the minimum radius is found, more distance to the center is expected for abnormal samples compared to normal ones. The figure is taken from (136).

### 3.11 Deep Semi-Supervised Anomaly Detection (137):

This is the semi-supervised version of DSVDD which assumes a limited number of labeled abnormal samples. The loss function is defined to minimize the distance of normal samples from a pre-defined center of a hyper-sphere while maximizing abnormal sample distances. The objective function is defined as follows:

$$
\min_W \frac{1}{n+m} \sum_{i=1}^n ||\phi(x_i; W) - c||^2 +
$$
$$
\frac{\eta}{n+m} \sum_{j=1}^m (||\phi(\hat{x}_i; W) - c||^2)^{\hat{y}_j} + \frac{\lambda}{2} \sum_{l=1}^L ||W^l||_F^2,
\tag{22}
$$

where $\phi(x_i; W)$ is a deep feature extractor with parameter $W$.

Note that, as mentioned above, there is access to $(\hat{x_1}, \hat{y_1}), ..., (\hat{x_m}, \hat{y_m}) \in X \times Y$ with $Y = \{-1, +1\}$ where $\hat{y} = 1$ denotes known normal samples and $\hat{y} = -1$ otherwise. $c$ is specified completely similar to DSVDD by averaging on the latent embeddings of an AE trained on normal training samples. From a somewhat theoretical point of view, AE's objective loss function helps the encoder maximize $I(X; Z)$ in which $X$ denotes input variables and $Z$ denotes latent variables. Then, it can be shown that (22) minimizes the entropy of normal sample latent embeddings while maximizing it for abnormal ones as follows:

$$
H(Z) = E[-\log(P(Z)] = - \int_Z p(z) \log p(z)\, dz
$$
$$
\leq \frac{1}{2} \log((2\pi e)^d \det(\Sigma)) \propto \log \sigma^2
\tag{23}
$$

As the method makes normal samples compact, it forces them to have low variance, and consequently, their entropy is minimized. The final theoretical formulation is approximated as follows:

$$
\max_{p(z|x)} I(X; Z) + \beta(H(Z^-) - H(Z^+)).
\tag{24}
$$

Here, $I$ specifies the mutual information and $\beta$ is a regularization term to obtain statistical properties that is desired for the downstream tasks.

### 3.12 Deep Anomaly Detection Using Geometric Transformations (GT) (48):

GT attempts to reformulate the one-class classification problem into a multi-class classification. GT defines a set of transformations that do not change the data distribution, then trains a classifier to distinguish between

them. Essentially, the classifier is trained in a self-supervised manner. Finally, a Dirichlet distribution is trained on the confidence layer of the classifier to model non-linear boundaries. Abnormal samples are expected to be in low-density areas since the network can not confidently estimate the correct transformation. At the test time, different transformations are applied to the input, and the sum of their corresponding Dirichlet probabilities is assigned as the novelty score. An overview of the method is shown in Fig. 14.

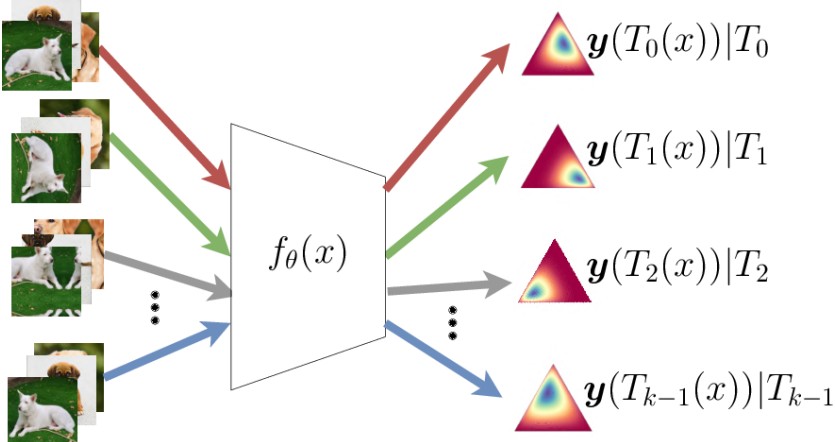

Figure 14: An overview of the GT method is shown. A set of transformations is defined, then a classifier must distinguish between them. Having trained the classifier in a self-supervised manner, a Dirichlet distribution is trained on the confidence layer to model their boundaries. The figure is taken from (48).

### 3.13 Effective End-To-End Unsupervised Outlier Detection via Inlier Priority of Discriminative Network (172):

In this work, similar to GT, a self-supervised learning (SSL) task is employed to train an anomaly detector except in the presence of a minority of outliers or abnormal samples in the training dataset. Suppose the following training loss is used as the objective function:

$$L_{\text{SS}}(x_i \mid \theta) = -\frac{1}{K} \sum_{y=1}^{K} \log(P^{(y)}(x_i^{(y)} \mid \theta)), \tag{25}$$

where $x_i^{(y)}$ is obtained by applying a set of transformations $O(. \mid y)$, and $y$ indicates each transformation. The anomaly detection is obtained based on the objective function score. Take for example the rotation prediction task. During training, the classifier learns to predict the amount of rotation for normal samples. During test time, different rotations are applied on inputs, the objective function scores for normal samples would be lower than abnormal ones.

However, due to the presence of abnormal samples in the training dataset, the objective score for abnormal samples may not always be higher. To address this issue, it is shown that the magnitude and direction of gradient in each step have a significant tendency toward minimizing inlier samples' loss function. Thus the network produces lower scores compared to abnormal ones. (172) exploits the magnitude of transformed inliers and outliers' aggregated gradient to update $w_c$, i.e. $||\nabla_{w_c}^{(\text{in})} L||$ and $||\nabla_{w_c}^{(\text{out})} L||$, which are shown to follow this approximation:

$$\frac{E(||\nabla_{w_c}^{(\text{in})} \cdot L||^2)}{E(||\nabla_{w_c}^{(\text{out})} \cdot L||^2)} \approx \frac{N_{\text{in}}^2}{N_{\text{out}}^2} \tag{26}$$

where $E(\cdot)$ denotes the probability expectation. As $N_{\text{in}} \gg N_{\text{out}}$, normal samples have more effect on the training procedure. Also, by projecting and averaging the gradient in the direction of each of the training sample's gradient $(-\nabla_\theta L(x) \cdot \frac{-\nabla_\theta L(x_i)}{||-\nabla_\theta L(x_i)||})$, the stronger effect of inlier distribution vs outlier is observed again. Fig. 15 shows the effect empirically.

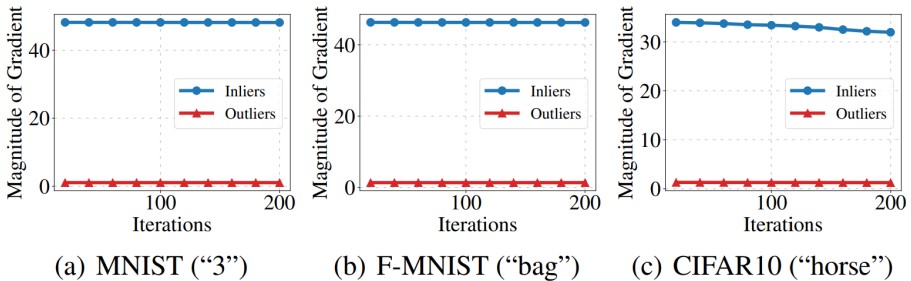

(a) MNIST ("3")   (b) F-MNIST ("bag")   (c) CIFAR10 ("horse")

Figure 15: The average magnitude of the gradient for inliers and outliers with respect to the number of iterations. The class used as inliers is in brackets. The figure is taken from (172).

### 3.14   Classification-Based Anomaly Detection for General Data (GOAD) (12):

This work is very similar to GT. It trains a network to classify between different transformations ($T$) ; however, instead of using cross-entropy loss or training a Dirichlet distribution on the final confidences, it finds a center for each transformation and minimizes the distance of each transformed data with its corresponding center as follows :

$$P(m' \mid T(x,m)) = \frac{e^{-||f(T(x,m))-c_{m'}||^2}}{\sum_{\hat{m}} e^{-||f(T(x,m))-c_{\hat{m}}||^2}} \tag{27}$$

where the centers $c_m$ are given by the average feature over the training set for every transformation i.e. $c_m = \frac{1}{N} \sum_{x \in X} f(T(x,m))$. For training $f$, two options are used. The first one is using a simple cross entropy on $P(m' \mid T(x,m))$ values, and the second one is using the center triplet loss (53) as follows:

$$\sum_i \max(0, ||f(T(x_i,m)) - c_m||^2 + s \\ - \min_{m' \neq m} ||f(T(x_i,m)) - c_{m'}||^2) \tag{28}$$

where $s$ is a margin regularizing the distance between clusters.

The idea can be seen as the combination of DSVDD and GT in which GT's transformations are used, and different compressed hyperspheres are learned to separate them. $M$ different transformations transform each sample at the test time, and the average of correct label probabilities is assigned as the anomaly score.

### 3.15   CSI: Novelty Detection via Contrastive Learning on Distributionally Shifted Instances (165):

This work attempts to formulate the problem of novelty detection into a contrastive framework similar to SimCLR (25). The idea of contrastive learning is to train an encoder $f_\theta$ to extract the necessary information to distinguish similar samples from others. Let $x$ be a query, $x_+$, and $x_-$ be a set of positive and negative samples respectively, $z$ be the encoder's output feature or the output of an additional projection layer $g_\phi(f_\theta(x))$ for each input, and suppose $\text{sim}(z, z')$ is cosine similarity. Then, the primitive form of the contrastive loss is defined as follows:

$$L_{\text{con}}(x, x_+, x_-) := -\frac{1}{|x_+|} \log \frac{\sum_{x' \in x_+} e^{\text{sim}(z(x), z(x'))/\tau}}{\sum_{x' \in x_+ \cup x_-} e^{\text{sim}(z(x), z(x'))/\tau}} \tag{29}$$

Specifically for SimCLR, the contrastive loss above is converted into the formulation below:

$$
\begin{aligned}
L_{\text{SimCLR}}(B; T) := \frac{1}{2B} \sum_{i=1}^{B} & L_{\text{con}}(\hat{x}_i^{(1)}, \hat{x}_i^{(2)}, \hat{B}_{-i}) \\
& + L_{\text{con}}(\hat{x}_i^{(2)}, \hat{x}_i^{(1)}, \hat{B}_{-i})
\end{aligned}
\tag{30}
$$

where $(\hat{x}_i^{(1)}, \hat{x}_i^{(2)}) = (T_1(x_i), T_2(x_i))$ for the transformations $T_1$ and $T_2$ from a set of transformations $T$, $B := \{x_i\}_{i=1}^{B}$, and $\hat{B}_{-i} := \{\hat{x}_j^{(1)}\}_{j \neq i} \cup \{\hat{x}_j^{(2)}\}_{j \neq i}$.

However, contrastive learning requires defining a set of negative samples. To this end, a collection of transformations that shift the distribution of training samples ($S$) is specified to make the desired negative set when applied on each input. For instance, rotation or patch permutation completely shifts the original input samples' distribution; and can therefore be used as negative samples. Another set of transformations called align transformations ($T$) is defined as not changing the distribution of training images as much as ($S$). Then the *Contrasting shifted instances* (CSI) loss can be defined as follows:

$$L_{con-SI} := L_{\text{SimCLR}}\big( \cup_{s \in S} B_s; T \big) \tag{31}$$

where $B_s := \{s(x_i)\}_{i=1}^{B}$. Here, CSI considers each distributionally-shifted sample as an OOD with respect to the original sample. The goal is to discriminate an in-distribution sample from other OOD i.e., $(s \in S)$ samples.

Further, to facilitate $f_\theta$ to discriminate each shifted instance, a classification loss for classifying shifted instances is defined in combination with $L_{\text{con-SI}}$. To do so, a linear layer for modeling an auxiliary softmax classifier $p_{\text{cls-SI}}(y^S \mid x)$ is added to $f_\theta$ as in GT or GOAD:

$$L_{\text{cls-SI}} = \frac{1}{2B} \frac{1}{K} \sum_{s \in S} \sum_{\hat{x}_s \in B_s} -\log p_{\text{cls-SI}}(y^s = s \mid \hat{x}_s), \tag{32}$$

where $\hat{B}_s$ is the batch augmented from $B_s$. In testing, the cosine similarity to the nearest training sample in $\{x_m\}$ multiplied by the norm of representation $||z(x)||$ is used. The contrastive loss increases the norm of in-distribution samples, as it is an easy way to minimize the cosine similarity of identical samples by increasing the denominator of (29). To further improve the scoring criterion, ensembling over random augmentations or utilizing shifting transformations similar to GOAD can be employed.

### 3.16 DA-Contrastive: Learning and Evaluating Representations for Deep One-class Classification (158):

This research tries to put the problem of novelty detection into a contrastive framework, similar to the other methods mentioned previously in (165) and (48). (165) reports that all of the augmentations that are utilized in the contrastive loss presented at 29 are not necessarily effective for the one-class learning. As a result, rotated versions of the input are treated as separate distributions in this study, and are accounted as negative samples in contrastive loss. Positive samples are also created by applying augmentations to each distribution, such as color jittering. Finally, the one-class classifier is trained on the positive and negative samples in a contrastive learning manner. After, a KDE or OC-SVM (148) is trained on the learned representation of normal training samples, which is used at the test time. An overview of the method can be seen in Fig. 16.

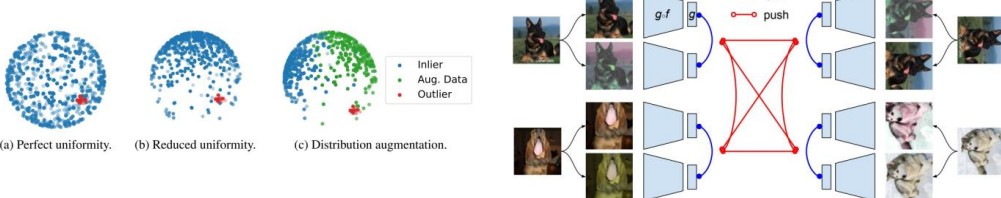

Figure 16: An overview of the DA-Contrastive method is represented. This work trains a one-class classifier employing a novel variant of contrastive learning with distribution augmentations such that the class collision between examples from the same class is reduced. This implies discriminating different distributions generated by augmentations of different rotations from the original input (see the right side), which results in less uniformity and more compactness in the feature space (see the left side). The figure is taken from (158).

### 3.17  Uninformed Students: Student-Teacher Anomaly Detection With Discriminative Latent Embeddings (14):

This work trains a teacher network using metric learning and knowledge distillation techniques to provide a semantic and discriminative feature space. The teacher $T$ is obtained by first training a network $\hat{T}$ that embeds patch-sized images $p$ into a metric space. Then, fast dense local feature extraction for an entire input image can be achieved by a deterministic network transformation from $\hat{T}$ to $T$, as described in (8). To train $\hat{T}$, a large number of training patches $p$ are obtained by randomly cropping an image database, for instance, ImageNet. Then using the following knowledge distillation loss, the knowledge of a pre-trained network $P$ is distilled into $\hat{T}$ as follows:

$$L_k(\hat{T}) = ||D(\hat{T}(p)) - P(p)||^2, \tag{33}$$

where $D$ is used to align the size of output spaces. This helps the use of computationally efficient network $\hat{T}$ instead of $P$ at the test time while the required knowledge is distilled.

To further enrich the feature space of $\hat{T}$, an SSL method is employed such that the feature space of patches that are obtained by applying small translations, small changes in the image luminance, and the addition of Gaussian noise to $p$ be similar. This set is called $p^+$ as opposed to $p^-$, which is obtained using random cropping regardless of the neighborhood to the patch $p$. The SSL loss is defined as follows:

$$\delta^+ = ||\hat{T}(p) - \hat{T}(p^+)||^2$$
$$\delta^- = \min\{||\hat{T}(p) - \hat{T}(p^-)||^2, ||\hat{T}(p^+) - \hat{T}(p^-)||^2\} \tag{34}$$
$$L_m(\hat{T}) = \max\{0, \delta + \delta^+ - \delta^-\}$$

where $\delta > 0$ denotes the margin parameter. Finally, to reduce the redundancy of feature maps of $\hat{T}$, a compactness loss which minimizes their cross-correlation is used as follows:

$$L_c(\hat{T}) = \sum_{i \neq j} c_{ij} \tag{35}$$

therefore, the final loss is $L_c(\hat{T}) + L_m(\hat{T}) + L_k(\hat{T})$.

Having a teacher $T$ trained comprehensively to produce $d$ dimensional feature maps for each pixel of an input image, an ensemble of student networks is forced to approximate the feature maps of $T$ for each pixel located at row $r$ and column $c$ as follows:

$$L(S_i) = \frac{1}{wh} \sum_{(r,c)} ||\mu_{(r,c)}^{S_i} - (y_{(r,c)}^T - \mu)\text{diag}(\sigma)^{-1}||_2^2 \tag{36}$$

where $\mu$ and $\sigma$ are used for data normalization, note that the receptive field of each student is limited to a local image region $p_{(r,c)}$, this helps the students obtain dense predictions for each image pixel with only a single forward pass without having to actually crop the patches $p_{(r,c)}$. At the test time, as students have only learned to follow their teacher on normal training samples, their average ensemble error can be used to detect abnormal samples. Intuitively, they wouldn't follow their teacher on abnormal samples and produce a high average error.

### 3.18 Self-Supervised Learning for Anomaly Detection and Localization (CutPaste) (90):

This work designs a simple SSL task to capture local, pixel-level regularities instead of global, semantic-level ones. While GT or GOAD utilize transformations such as rotation, translation, or jittering, CutPaste cuts a patch of its training input and copies it in another location. The network is trained to distinguish between defected samples and intact ones. Extra auxiliary tasks such as cutout and Scar can be used in combination with the cut-paste operation. After training, a KDE or Gaussian density estimator is trained on the confidence scores of normal training samples, which is used at the test time. Due to the simplicity of the method, it may overfit easily on the classification task. However; several experiments in the paper show contrary to this assumption. An overview of the method can be seen in Fig. 17.

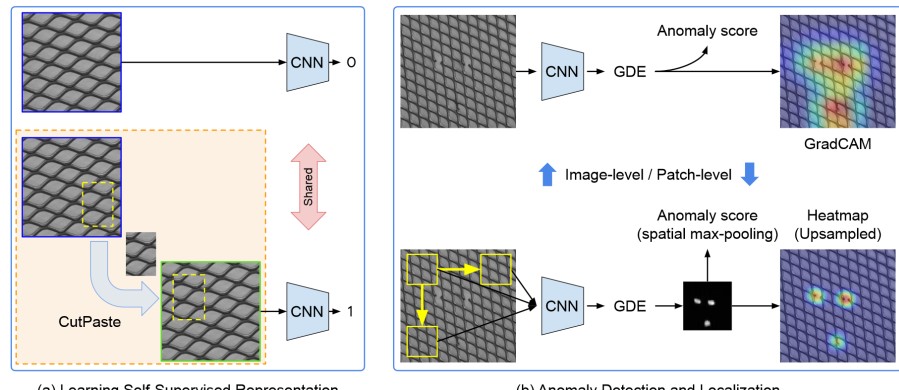

Figure 17: An overview of CutPaste. A network is trained to classify artificially made defected samples from the original ones. Despite the simplicity of the method, it does not overfit on MVTecAD dataset. The figure is taken from (90).

### 3.19 Multiresolution Knowledge Distillation for Anomaly Detection (Multi-KD) (144):

Generative models are suited for detecting pixel-level anomalies; however, they may fail on complex, semantic-level ones. In contrast, discriminative models are good at capturing semantics. Designing an SSL task that captures both semantic and syntax is not easy. To address this issue, Multi-KD attempts to mimic the intermediate layers of a VGG pre-trained network—called intermediate knowledge—into a simpler network using knowledge distillation. This way, multi-resolution modeling of the normal training distribution is obtained and can be used to detect both pixel-level and semantic-level anomalies at the test time. Here, the concept of knowledge is defined to be the length and direction of a pre-trained network on ImageNet (31). A cloner network has a simple yet similar overall architecture compared to the source, making its knowledge similar to the source on normal training samples. At test time, the cloner can follow the source on the normal test time samples; however, it fails for the abnormal ones. This results in a high discrepancy that can be used at test time. Fig. 18 shows the overall architecture.

Similar methods exist, which model the distribution of different layers of a pre-trained network using multivariate Gaussian descriptors such as PaDiM (30) or (134). An overview of the architecture of (134) is shown in the Fig. 19.

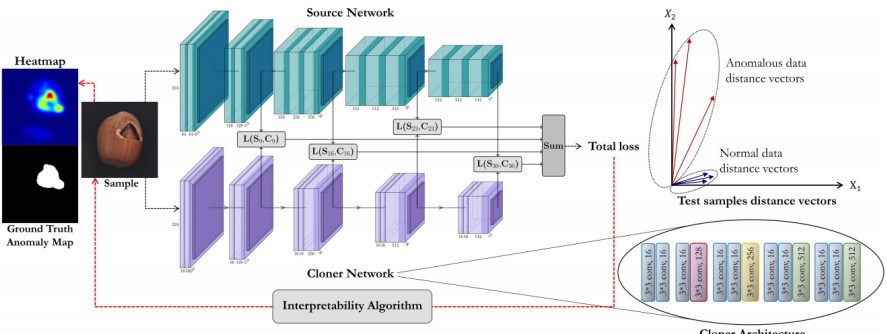

Figure 18: An overview of the Multi-KD method is shown. It distills multi-resolution knowledge of a pre-trained source network on the normal training distribution into a simpler cloner one. The structure of source and cloner is similar to each other. Therefore, the cloner distills the source's knowledge in the corresponding layers. At the test time, their discrepancy would be low for normal inputs as opposed to abnormal ones. The figure is taken from (144).

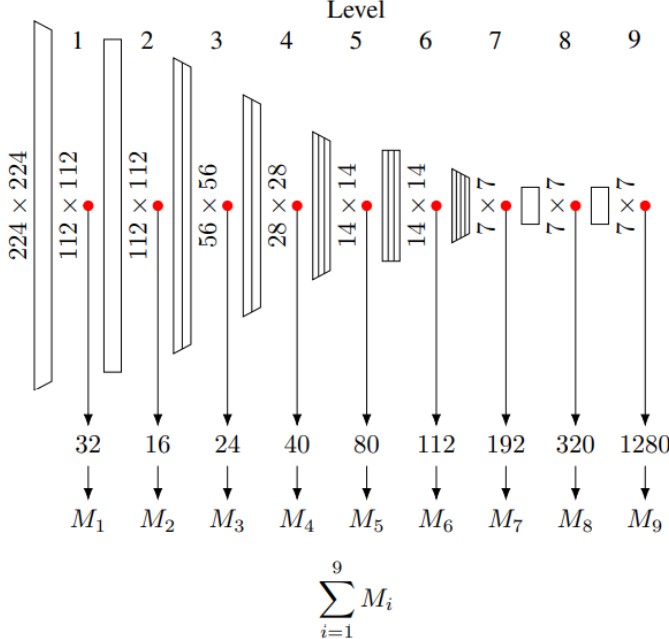

Figure 19: An overview of (134). It models the different layers' distribution of a pre-trained network using Gaussian descriptors. Then, at the test time, the probability of being normal is computed using the training time means and variances. The figure is taken from (134).

## 4 Open-set Recognition

Open-set recognition (OSR) receives more supervision than AD or ND. In this setting, $K$ normal classes are given at the training time. At testing, $N$ classes with $N - K$ unknown and $K$ known classes exist. The objective is to identify unknown classes while classifying the known ones. This has a lot of applications in

which labeling normal datasets is more feasible, or gathering a cleaned dataset without having any abnormal sample is possible. As there is more supervision, training data can be categorized into four classes:

- *known known classes (KKC):* Training samples that we know they are known. They are already given and labeled.

- *known unknown classes (KUC):* Training samples that we know they are not known. This means, they do not belong to the known categories. For example, background images, or any image that we know is not categorized into the known classes are in this group. They are already given and labeled.

- *unknown known classes (UKC):* Training samples that we do not know they are known. For example, known test time samples are in this group. These are not given at the training phase.

- *unknown unknown classes (UUC):* Training samples that we do not know they are not known. For example, unknown test time samples are in this group. These are not given at the training phase.

The space that is far from known classes, including KKC and KUC, is called **open space** $O$(145). Labeling any sample in this space has a risk value denoted by $R_O$. The risk factor is usually represented as Eq. 37 in which $S_o$ is the overall measuring space, and $f$ is the measurable recognition (classification) function (145). The value of $f$ is 1 if the input belongs to KKC otherwise 0.

$$R_O(f) = \frac{\int_O f(x)\,dx}{\int_{S_o} f(x)\,dx} \tag{37}$$

As discussed in (47), there are some difficulties in defining the practical formulation of openness; therefore, this study uses the definition of (47) as in Eq. 38, where $C_{TR}$ is the set of classes used in training, and $C_{TE}$ is the set of classes used in the testing.

$$O = 1 - \sqrt{\frac{2 \times |C_{TR}|}{|C_{TR}| + |C_{TE}|}} \tag{38}$$

Larger values of $O$ correspond to more open problems, and 0 is assigned to a completely closed one. OSR problem can be defined as finding recognition functions $f$ in such a way that minimizes the risk factor $R_O$.

### 4.1 Towards Open Set Deep Networks (OpenMax) (11):

This work addresses the problem of overconfident scores of classification models for unseen test time samples. Due to the normalization in softmax computation, two samples with completely different logit scores may have the same confidence score distribution. Instead of using the confidence score, OpenMax resorts to the logit scores that are shown by *Activation Vector* (AV). AV of each sample captures the distribution over classes. The mean AV (MAV) is defined to be the average of AV values across all samples. As for each input sample, the value in AV corresponding to the ground truth is supposed to be high; its distance to the corresponding value of MAV would be high too. Considering the distance of each element in AVs from the corresponding element in MAV as a random variable, correctly classified inputs would have the highest distances for ground truth elements. This might happen for a few classes that are not ground truth but have a strong relationship with ground truth. For instance, the class `leopard` as the ground truth and `cheetah` as a near class. To model the distribution of these highest values, the Extreme Value Theorem (EVT) can be used as follows:

EVT : Let $(s_1, s_2, ..., s_n)$ be a sequence of i.i.d. samples. Let $M_n = \max(s_1, ..., s_n)$. If a sequence of pairs of real numbers $(a_n, b_n)$ exists such that each $0 \leq a_n$ and

$$\lim_{n \to \infty} P\left(\frac{M_n - b_n}{a_n} \leq x\right) = F(x) \tag{39}$$

Then if $F$ is a non-degenerate distribution function, it belongs to one of three extreme value distributions (126). For this work, the Weibull distribution is used. By modeling the distribution of extremes for each class, one can easily compute the probability of each test input being an extreme and discard remaining ones.

In summary, during training, the CDF of extreme values for each class or element of AVs are approximated using EVT and correctly classified training samples. For each test-time input sample, the top $\alpha$ elements in AV are selected, representing ground truth and near classes; then, their CDFs are assigned to variables $w_{s(i)}(x)$ as Fig. 20 shows. These probabilities represent the chance of each AV's element $v_j(x)$ to be the maximum. After that, an arbitrary class is added to the existing ones as the representative of unknown unknown samples, assumed as class 0, and all activation values are normalized as follows:

$$\hat{v}(x) = \mathbf{v}(x) \circ w(x)$$
$$\hat{v}_0(x) = \sum_i v_i(x)(1 - w_i(x)) \tag{40}$$

The normalization is performed so that the activation vector of UUC would be high if $w_{s(i)}(x)$s are small. Finally, softmax is computed based on the normalized activation vector, and unknown or uncertain inputs are rejected. All hyper-parameters such as $\epsilon, \alpha$ are tuned using a validation set including both seen and unseen samples.

---

**Algorithm**   OpenMax probability estimation with rejection of unknown or uncertain inputs.

---

**Require:** Activation vector for $\mathbf{v}(\mathbf{x}) = v_1(x), \ldots, v_N(x)$
**Require:** means $\mu_j$ and libMR models $\rho_j = (\tau_i, \lambda_i, \kappa_i)$
**Require:** $\alpha$, the numer of "top" classes to revise
1: Let $s(i) = \text{argsort}(v_j(x))$; Let $\omega_j = 1$
2: **for** $i = 1, \ldots, \alpha$ **do**
3:     $\omega_{s(i)}(x) = 1 - \frac{\alpha - i}{\alpha} e^{-\left(\frac{\|x - \tau_{s(i)}\|}{\lambda_{s(i)}}\right)^{\kappa_{s(i)}}}$
4: **end for**
5: Revise activation vector $\hat{v}(x) = \mathbf{v}(\mathbf{x}) \circ \omega(\mathbf{x})$
6: Define $\hat{v}_0(x) = \sum_i v_i(x)(1 - \omega_i(x))$.
7:

$$\hat{P}(y = j | \mathbf{x}) = \frac{e^{\hat{v}_j(\mathbf{x})}}{\sum_{i=0}^{N} e^{\hat{v}_i(\mathbf{x})}}$$

8: Let $y^* = \text{argmax}_j P(y = j | \mathbf{x})$
9: Reject input if $y^* == 0$ or $P(y = y^* | \mathbf{x}) < \epsilon$

---

Figure 20: The inference algorithm of open-max. The class 0 is added in addition to other classes and considered as the unknown class; then, its probability is computed for each of the input samples, and the probabilities of top k values are normalized with respect to the unknown class's probability. At last, if the most significant value is for class 0 or the confidence of top $k$ values is less than a specific amount, the sample is discarded. The figure is taken from (11).

## 4.2   Generative OpenMax for Multi-Class Open Set Classification (G-OpenMax) (46):

This work is similar to OpenMax with exceptions of artificially generating UUC samples with GANs and fine-tuning OpenMax. This removes the need to have a validation dataset. To generate UUCs, a conditional

GAN is trained on the KKCs as follows:

$$\min_{\phi} \max_{\theta} = \mathbb{E}_{x,c \sim P_{\text{data}}}[\log(D_\theta(x,c))]$$
$$+ \mathbb{E}_{z \sim P_z, c \sim P_c}[\log(1 - D_\theta(G_\phi(z,c),c))], \tag{41}$$

where $D$ is the discriminator with parameter $\theta$, and $G$ is the generator with parameter $\phi$.

New samples are generated by interpolating the KKCs' latent space. If the generated sample is not classified as one of the mixing labels, the image is considered UUC. Finally, another classifier called $Net^G$ (see Fig. 21) is trained using both UUC and KKC samples such that UUCs can be assigned to the class 0 of the classifier. The inference steps are similar to OpenMax except that the normalization process for the class 0 and other classes is the same.

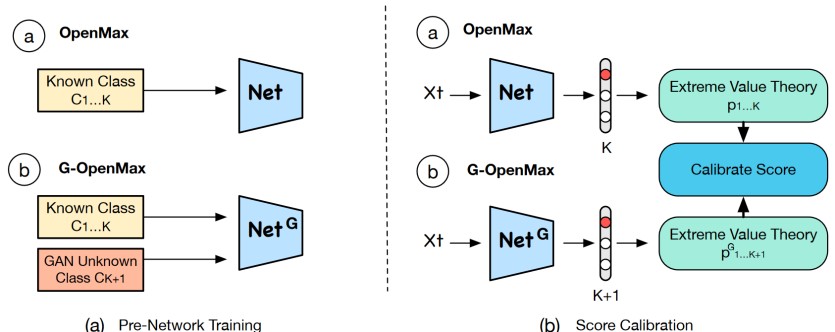

Figure 21: The overall architecture of G-OpenMax compared to the base OpenMax. As it is shown, everything is completely the same except the unknown unknown sample generation, which is done using a GAN. The figure is taken from (46).

## 4.3 Open Set Learning with Counterfactual Images (OSRCI) (111):

This work follows the idea of generating UUC samples as in G-OpenMax. A generated input is made similar to a KKC yet not to be assigned to the same class. Such generated inputs are called counter-factual examples. As these samples are near the boundary of UUCs, they can better help approximate the real UUC distribution.

Specifically, a classifier is trained on $k$ classes. A GAN-based method such as ALOCC is trained. The discriminator is trained using a Wasserstein critic with gradient penalty as follows:

$$L_D = \sum_{x \in X} D(G(E(x))) - D(x) + P(D)$$
$$L_G = \sum_{x \in X} ||x - G(E(x))||_1 - D(G(E(x))), \tag{42}$$

where $P(D)$ is the interpolated gradient penalty term, $E$ encodes the input, $D$ is the discriminator, and $G$ is the generator.

The counter-factual samples are generated by optimizing a latent vector $z$, which has a small distance to the latent vector of an arbitrary training sample $x$ while its decoded image produces low confidence scores for each of the known classes:

$$z^* = \min_z ||z - E(x)||_2^2 + \log\left(1 + \sum_{i=1}^{K} e^{C_K(G(z))_i}\right), \tag{43}$$

where $C_K$ is the classifier and $C_K(G(z))_i$ is the logit of the counterfactual image $G(z)$ for class $i$. The generated UUCs are used in combination with KKCs to train a new classifier $C_{K+1}$, which is used at the test time.

### 4.4 Reducing Network Agnostophobia (33):

In applications such as object detection, there is usually a class called background. On the internet, a large number of samples can be crawled, which can be used as "background" for a specific task. This work employs background samples as an auxiliary KUC distribution to train a classifier. The training facilitates KUC to have small feature magnitudes while KKCs have larger magnitudes with a defined margin. Also, the entropy of the confidence layer is maximized for the background samples, which is equivalent to increasing the classifier's uncertainty for such inputs. The training employs a simple entropic open-set loss that maximizes the entropy of confidence scores, along with the objectosphere loss minimizing the $L_2$ norm of final features. Fig. 22 shows the effect of each loss on the geometric location of each class's samples at the final layer. At the test time, the thresholding is based on $S_c(x).||F(x)||$, where $F(x)$ is the representation from penultimate layer.

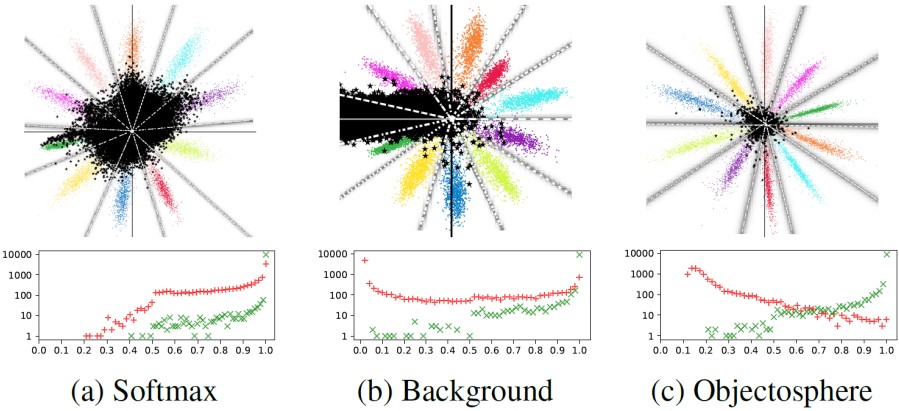

(a) Softmax      (b) Background      (c) Objectosphere

Figure 22: The effect of different loss functions on the last layer of a classifier. The classifier is trained on the MNIST dataset. The background class is considered to be NIST letters (51). Black dots show samples from Devanagari (119) as UUCs, and the gray lines show the boundaries of different classes. (a) shows the last layer when the network is trained only with softmax loss. (b) shows similar setting to (a); however, background samples are used as a separate class. (c) shows the last layer when the network is trained with the objectosphere loss. The figures in the bottom are histograms of softmax probability values for samples of KKCs with green color and UUCs with the red one. The figure is taken from (33).

### 4.5 Class Conditioned Auto-Encoder for Open-Set Recognition (C2AE) (117):

The second premise behind employing AEs is used in this study, which states that abnormal test time samples should not be reconstructed as well as normal ones. Nevertheless, despite AD and ND, in OSR, training labels are used to increase AE abilities. Here, an AE is used as the meta-recognition function while its encoder plays the role of a classifier for the recognition task. Intuitively, this work wants the encoder to classify each passed sample correctly and provide embeddings by which the reconstruction of the original input is possible. Furthermore, it imposes other constraints to force encoder embeddings not to be easily converted to each other, e.g., by applying linear transformations, which prevents the AE from utilizing the learned features to reconstruct abnormal/unseen inputs.

To this end, at first, the encoder that is a classifier is trained and fixed. Then for a given input $X_i$ a match vector $l_m = l_{y_i^m} \in \{-1, 1\}^K$ where $K$ is the number of classes is defined such that $l$ is equal to 1 for the $y^i$th element and $-1$ otherwise. Similarly, some none-match vectors $l_{nm} = l_{y_j^{nm}}$ for any random $y_j^{nm} \neq y_i$ sampled randomly from labels are considered. After that, two neural networks $H_\gamma$ and $H_\beta$ with parameters $\Theta_\gamma$ and $\Theta_\beta$ are defined to receive match and none-match vectors and produce some linear transformations by which the encoder embeddings are transformed. Finally, match transformations must be constructed perfectly; however, none-match ones are forced to have high reconstruction loss as follows:

$$
\begin{aligned}
z_i &= F(X_i), \\
\gamma_{y_i^m} &= H_\gamma(l_{y_i^m}), & \gamma_{y_i^{nm}} &= H_\gamma(l_{y_i^{nm}}) \\
\beta_{y_i^m} &= H_\beta(l_{y_i^m}), & \beta_{y_i^{nm}} &= H_\beta(l_{y_i^{nm}}) \\
z_{il_m} &= \gamma_{y_i^m} \circ z_i + \beta_{y_i^m}, & z_{il_{nm}} &= \gamma_{y_i^{nm}} \circ z_i + \beta_{y_i^{nm}} \\
\hat{X}_i^m &= \mathcal{G}(z_{l_m}). & \hat{X}_i^{nm} &= \mathcal{G}(z_{l_{nm}}). \\
\mathcal{L}_r^m &= \frac{1}{N} \sum_{i=1}^N ||X_i - \hat{X}_i^m||_1 \\
\mathcal{L}_r^{nm} &= \frac{1}{N} \sum_{i=1}^N ||X_i^{nm} - \hat{X}_i^{nm}||_1
\end{aligned}
\tag{44}
$$

where $\hat{X}_i^{nm}$s are sampled randomly based on randomly sampled none-match vectors, and the objective loss is:

$$
\min_{\Theta_\gamma, \Theta_\beta, \Theta_\mathcal{G}} \alpha \cdot \mathcal{L}_r^m + (1-\alpha) \cdot \mathcal{L}_r^{nm}
\tag{45}
$$

The embedding transformation technique is called FiLM (125). This means that a given input would be reconstructed well only when the corresponding match vector is used. Therefore, for each test-time input, the reconstruction error under different match vectors is computed, and the rejection decision is made based on their minimum value. If the minimum value is greater than a threshold $\tau$ it is discarded; else, the encoder's output is assigned. To obtain $\tau$ in a principled way, the optimum threshold is found using EVT on the distribution of match and none-match reconstruction error values i.e $||X_i - \hat{X}_i^m||_1$ and $||X_i - \hat{X}_i^{nm}||_1$ for each $i$ and randomly sampled none-match vectors. Assuming the prior probability of observing unknown samples is $p_u$, the probability of error as a function of threshold $\tau$ is shown in Eq. 46 where $G_m$ and $G_{mn}$ are the extreme value distributions of the match and none-match errors.

$$
\begin{aligned}
\tau^* &= \min_\tau P_{\text{error}}(\tau) \\
&= \min_\tau [(1-p_u) * P_m(r \geq \tau) + p_u * P_{nm}(-r \leq -\tau)] \\
&= \min_\tau [(1-p_u) * (1 - G_m(\tau)) + p_u * (1 - G_{nm}(\tau))]
\end{aligned}
\tag{46}
$$

### 4.6 Deep Transfer Learning For Multiple Class Novelty Detection (DTL) (121):

This work also follows the idea of using a background dataset (called reference dataset). Similar to (33), DTL addresses the deficiencies of using softmax loss in OSR. A new loss function called *membership loss* ($L_M$) is proposed. Specifically, each activation score value $f_i$ of the penultimate layer is normalized into $[0, 1]$ using the sigmoid function. The normalized scores can be interpreted as the probability of the input image belonging to each individual class. Ideally, given the label $y$, $f(x)_i$ should be 1 when $y = i$ and 0 otherwise. The loss function is defined as Eq. 47.

$$
L_M(x, y) = [1 - \sigma(f(x)_y)]^2 + \lambda \cdot \frac{1}{c-1} \sum_{i=1, i \neq y}^c [\sigma(f(x)_i)]^2
\tag{47}
$$

$\lambda$ is a regularization parameter to control the relative weight given to each risk source, $\sigma$ denotes the Sigmoid function, and $c$ is the number of classes.

Another technique for improving the detection performance is based on the *"globally negative filters"*. Filters that provide evidence for a particular class are considered as positive filters and vice versa. For pre-trained

neural networks, it has been shown that only a small fraction of final feature maps are activated positively. Furthermore, some filters are always activated negatively, indicating irrelevance for all known classes. By discarding inputs that activate globally negative filters, a novel sample is less likely to produce high activation scores. To learn such filters for domain-specific tasks, DTL trains two parallel networks with shared weights up to the last layer—the first one solves a classification task on the reference dataset, and the second one solves the domain-specific classification tasks in combination with membership loss. If the reference and domain-specific datasets do not share much information, they provide negative filters for each other. Also, since the reference dataset consists of diverse classes, those learned filters can be considered globally negative filters. Finally, filters of the parallel network in combination with the confidence scores of the domain-specific classifier are used for novelty detection. Fig. 47 shows the overall network architecture.

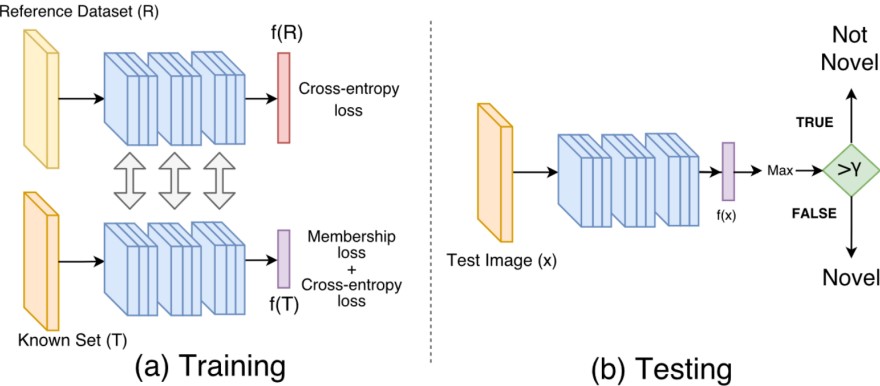

Figure 23: An overview of DTL method. Two parallel networks are trained to produce confidence scores and globally negative filters. The network above solves a simple classification task on the reference dataset, and the one below solves membership loss in combination with the domain-specific classification task. Test time detection is done by putting a threshold on the confidence of maximum class. The figure is taken from (121).

## 4.7 Classification-Reconstruction Learning for Open-Set Recognition (CROSR) (183):

This work follows the similar idea as C2AE. In particular, CROSR employs an encoder network for classification and producing the latent vectors for reconstruction task. Importantly, the latent vector $z$ used for the reconstruction task and penultimate layer $y$ used for the classification task are not shared. The reason is that there is an excessive amount of information loss in the penultimate layer, which makes distinguishing between unknown and known samples hard. The overall procedure can be described as follows:

$$
\begin{aligned}
(y, z) &= f(x), \\
p(C_i \mid x) &= \text{Softmax}_i(y), \\
\hat{x} &= g(z)
\end{aligned}
\tag{48}
$$

Moreover, to preserve information at different levels of abstraction, each layer of the encoder is compressed into a latent vector $z_i$. The latent vector is then decoded to minimize the reconstruction error of the corresponding layer as follows:

$$
\begin{aligned}
x_{l+1} &= f_l(x_l), \\
z_l &= h_l(x_l), \\
\hat{x}_l &= g_l(\hat{x}_{l+1} + \hat{h}_l(z_l))
\end{aligned}
\tag{49}
$$

where $f_l$ and $g_l$ are layers of the encoder and decoder, respectively. $\hat{h}_l$ is a reprojection to the original space of $x_l$. The autoencoding structure is based on a ladder network (130).

The final latent vector $\mathbf{z}$ is the concatenation of each $z_i$. EVT is employed as in OpenMax (11). However, the distance is not only computed on the penultimate layer $y$ but also on the latent vector $\mathbf{z}$. Specifically, EVT is applied on the joint $[y, \mathbf{z}]$. Test-time detection is performed similar to OpenMax. Fig. 24 shows an overview of the method.

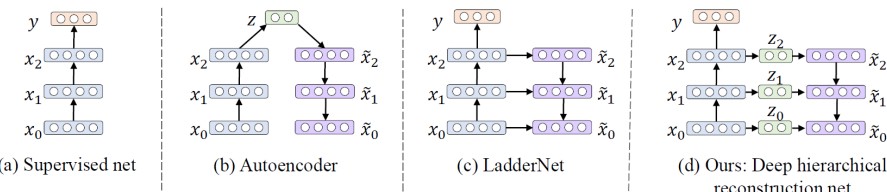

Figure 24: An overview of the proposed method compared to similar methods in the literature. As it is obvious, an encoder is used for the classification task, and the reconstruction task is done in a ladder network. The figure is taken from (183).

## 4.8 Generative-Discriminative Feature Representations For Open-Set Recognition (GDFR) (123):

Similar to CROSR, this proposed work trains a discriminative model in combination with a generative one. Discriminative approaches may lose important features that are utilitarian for distinguishing between seen and unseen samples. Generative modeling can provide complementary information. Similar to GT, GDFR employs SSL to improve the features of the discriminator. A shared network performs both the classification and SSL tasks, predicting the geometric transformations applied to the input.

Moreover, a generative model such as AE is used, producing reconstructed outputs $\hat{x}$ for a given input $x$. Then the collection of input-reconstruction pairs $(x, \hat{x})$ are passed to the discriminator network for classification and SSL tasks. The disparity between $\hat{x}$ and $x$ for unseen samples improves the discriminator network's detection power. Fig. 25 shows the method.

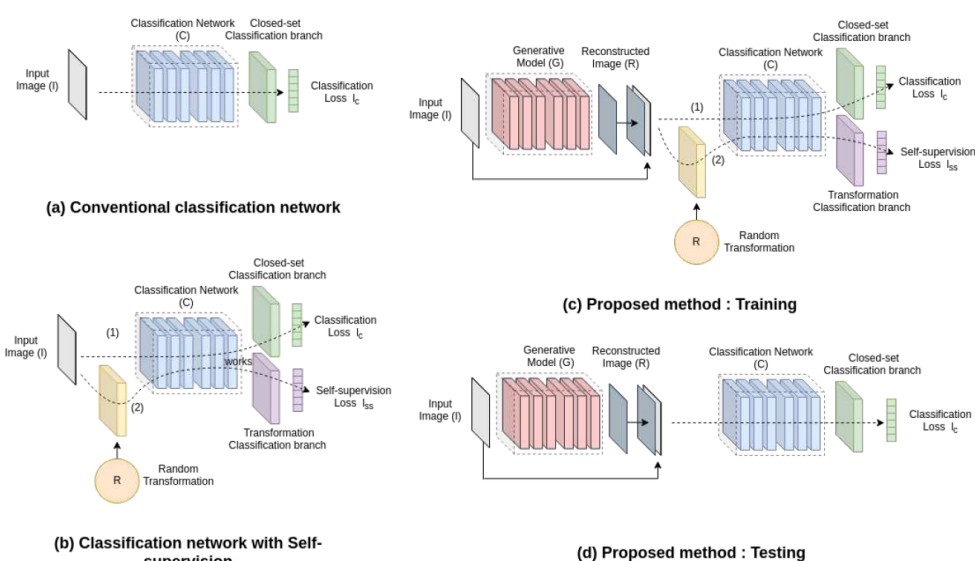

Figure 25: An overview of GDFR. As it can be seen, a classification network is trained on both SSL and classification tasks to make the learned features richer. Also, to benefit from generative modeling, each input is passed to an AE, and further tasks are applied on the joined reconstructed and input images. The figure is taken from (123).

### 4.9 Conditional Gaussian Distribution Learning for Open Set Recognition (CGDL) (162):

The main idea of this research is very similar to CROSR. However, CGDL uses the probabilistic ladder network based on variational encoding and decoding (160). The overall encoding process for the $l$th layer is as follows:

$$
\begin{aligned}
x_l &= \text{Conv}(x_{l-1}) \\
h_l &= \text{Flatten}(x_l) \\
\mu_l &= \text{Linear}(h_l) \\
\sigma_l^2 &= \text{Softplus}(\text{Linear}(h_l)),
\end{aligned}
\tag{50}
$$

where "Conv" is a convolutional layer , "Flatten" is a linear layer to flatten the input data into 1-dimensional space, "Linear" is a single linear layer, and "Softplus" operation is $\log(1 + \exp(\cdot))$.

The final representation vector $z$ is defined as $\mu + \sigma \odot \epsilon$ where $\epsilon \sim N(0, I)$, $\odot$ is the element-wise product, and $\mu$, $\sigma$ are the outputs of the top layer L. Similarly, for the decoding process we have:

$$
\begin{aligned}
\hat{c}_{l+1} &= \text{Unflatten}(\hat{z}_{l+1}) \\
\hat{x}_{l+1} &= \text{ConvT}(\hat{c}_{l+1}) \\
\hat{h}_{l+1} &= \text{Flatten}(\hat{x}_{l+1}) \\
\hat{\mu}_l &= \text{Linear}(\hat{h}_{l+1}) \\
\hat{\sigma}_l^2 &= \text{Softplus}(\text{Linear}(\hat{h}_{l+1})) \\
q\text{-}\mu_l &= \frac{\hat{\mu}_l + \hat{\sigma}_l^{-2} + \mu_l + \sigma_l^{-2}}{\hat{\sigma}_l^{-2} + \sigma_l^{-2}} \\
q\text{-}\sigma_l^2 &= \frac{1}{\hat{\sigma}_l^{-2} + \sigma_l^{-2}} \\
\hat{z}_l &= q\text{-}\mu_l + q\text{-}\sigma_l^2 \circ \epsilon.
\end{aligned}
\tag{51}
$$

During training, samples are passed into the encoder to estimate $\mu$ and $\sigma$ for each layer. The mean and variance can be used as priors for the corresponding decoding layer. The final embedding $z$ of the encoder's top layer is used for the joint classification task and decoding process. The distribution of the encoder's final layer is forced to be similar to different multivariate Gaussian $p_\theta^k(z) = N(z; \mu_k, I)$, where $k$ is the index of known classes and $\mu_k$ is obtained by a fully-connected layer which maps the one-hot encoding of the input's label to the latent space. Each layer of the decoder is a Gaussian probability distribution in which a prior of its mean and variance is added by the corresponding layer of encoder statistics. Putting it together, the training objective function is as follows:

$$
\begin{aligned}
\mathcal{L}_{KL} &= -\frac{1}{L}\Big[\text{KL}\big(q_\phi(z \mid x, k) || p_\theta^k(z)\big) \\
&\quad + \sum_{l=1}^{K-1} \text{KL}\big(q_\theta(\hat{x}_l|\hat{x}_{l+1}, x) || q_\theta(\hat{x}_l|\hat{x}_{l+1})\big)\Big] \\
\mathcal{L}_r &= ||x - \hat{x}||_1 \\
\mathcal{L} &= -(\mathcal{L}_r + \lambda \cdot \mathcal{L}_c + \beta \cdot \mathcal{L}_{KL}),
\end{aligned}
\tag{52}
$$

where $\mathcal{L}_c$ is the classification error and

$$
\begin{aligned}
q_\theta(\hat{x}_l|\hat{x}_{l+1}, x) &= N(\hat{x}_l; q\text{-}\mu_l, q\text{-}\sigma_l^2) \\
q_\theta(\hat{x}_l|\hat{x}_{l+1}) &= N(\hat{x}_l; \hat{\mu}_l, \hat{\sigma}_l^2).
\end{aligned}
\tag{53}
$$

At the test time, both the probability of final latent space with respect to $p_\theta^k(z)$ and reconstruction error for each test input are used for detection. If an input is preserved, the classification output is considered as the true class. As Fig. 26 shows, the latent vector $z$ is considered as a prior in which the distribution of all classes should have a low KL distance. This is similar to the role of $N(0, I)$ in the baseline VAE.

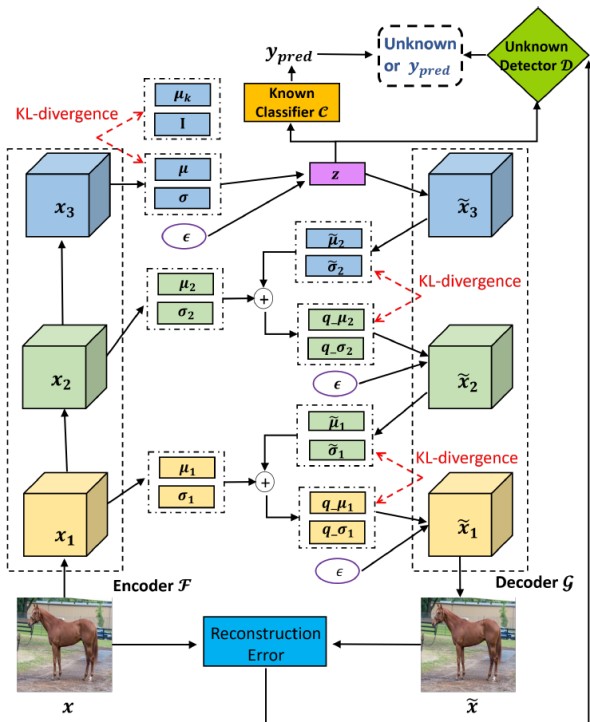

Figure 26: An overview of CGDL method. Compared to CROSR a probabilistic ladder network is used instead of a deterministic one; however, the classification task is similar. The probability distributions of the decoder network contain a prior from the corresponding layer statistics of the encoder network. Unlike, VAE in which there is a prior $N(0, I)$ on the latent space; here, the prior is learned, and each class distribution is conducted to have a low KL distance with it. The figure is taken from (162).

## 4.10 Hybrid Models for Open Set Recognition (192):

In this work, a classification network is trained in combination with a flow-based generative model. Generative models in the pixel-level space may not produce discernible results for unseen and seen samples, and they are not robust against semantically irrelevant noises. To address this issue, a flow-based model is applied to the feature representation space instead of pixel-level space (see Fig. 27). The reason for using flow-based models is their handy and comprehensive theoretical abilities. The training loss is a simple cross-entropy loss in combination with negative log-likelihood used for the training of the flow-based model. At the test time, the thresholding is applied to the likelihood of each input, and if it is preserved, the classifier's output is assigned as the in-class label.

## 4.11 Learning Open Set Network With Discriminative Reciprocal Points (RPL) (22):

Similar to Mem-AE, the idea of prototype features is used in this work. The goal is to learn a set of prototypes or reciprocal points, which can assign labels to each input based on the distance w.r.t each prototype. RPL helps the model better adjust the boundaries of different classes compared to softmax or OpenMax, and decreases the risk factor. Initially, random reciprocal points are chosen. The location of reciprocal points and the weights of a classifier network are adjusted to minimize the classification loss. This forces the network to locate features of each class near some specific reciprocal points to yield the desired class boundary using at

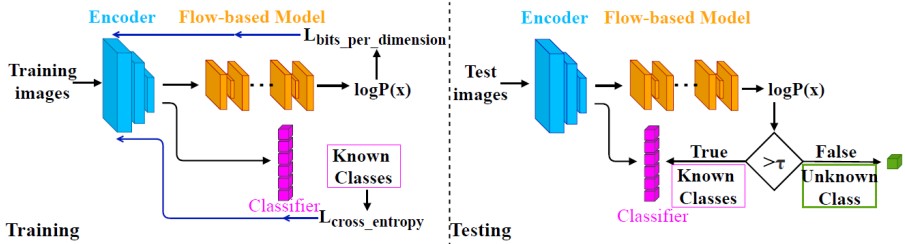

Figure 27: An overview of the hybrid models for open set recognition. As it can be seen, a classification network is trained in combination with a generative flow-based model. At the test time, the probability of the latent vector is considered as the criterion of rejection, and the classifier assigns a label if the input is not rejected. The figure is taken from (192).

least a set of points. To decrease the risk factor, samples of each class are forced to have a margin with respect to their reciprocal points, which is learned during the training process. Eq. 54 shows the mathematical formulation:

$$d(f_{\theta(x)}, P^k) = \frac{1}{M} \sum_{i=1}^{M} ||f_\theta(x) - p_i^k||_2^2$$

$$p(y = k|x, f_\theta, P) = \frac{e^{\gamma d(f_\theta(x), P^k)}}{\sum_{i=1}^{N} e^{\gamma d(f_\theta(x), P^i)}} \tag{54}$$

$$L_o = \frac{1}{M} \sum_{i=1}^{M} ||d(f_\theta(x), p_i^k) - R^k||_2^2,$$

where $P^k$ is the set of $k$th class reciprocal points, $p_i^k$ is a reciprocal point, $M$ is the number of reciprocal points for each class, $N$ is the number of classes, $R^k$ is the margin for each class, and $\gamma$ is a tunable hyper-parameter.

### 4.12 A Loss for Distance-Based Open Set Recognition (CAC) (104):

The idea of this work is similar to RPL and GOAD. For each class, CAC defines an anchor vector of dimension $N$—the number of classes. For each vector, the element corresponding to the class label is 1 and 0 otherwise. For each training sample, the learning process forces its logit scores to be in a compact ball w.r.t true class anchor vector while having a large distance from anchors of other classes. CAC can also be seen as a multi-class DSVDD. The training loss function is as follows:

$$\mathbf{d} = e(\mathbf{z}, \mathbf{C}) = (||\mathbf{z} - \mathbf{c}_1||_2, ..., ||\mathbf{z} - \mathbf{c}_N||_2)$$

$$\mathcal{L}_T(\mathbf{x}, y) = \log(1 + \sum_{j \neq y}^{N} e^{d_y - d_j}) \tag{55}$$

$$\mathcal{L}_A(\mathbf{x}, y) = d_y = ||f(x) - \mathbf{c}_y||_2$$

$$\mathcal{L}_{CAC}(\mathbf{x}, y) = \mathcal{L}_T(\mathbf{x}, y) + \lambda \cdot \mathcal{L}_A(\mathbf{x}, y),$$

where $f$ is a feature extractor projecting an input image $x$ to a vector of class logits $z = f(x)$, $N$ is the number of known classes, and $\mathbf{C}$ is a non-trainable parameter representing a set of class center points $(\mathbf{c}_1, \ldots, \mathbf{c}_N)$, one for each of the $N$ known classes.

### 4.13 Few-Shot Open-Set Recognition Using Meta-Learning (PEELER) (93):

In this work, the idea of meta-learning is combined with open set recognition. Meta-learning enable learning general features that can be easily adapted to any unseen task. Meta-learning is also called learning to learn.

Due to the ability to work in few-shot settings, meta-learning can be useful in low data regimes. At the meta iteration $i$, meta-model $h$ is initialized with the one produced by the previous meta-iteration. Let $(S_i^s, T_i^s)_{i=1}^{N^s}$ be a meta-training dataset with $N^s$ number of training problems, two steps are performed. First, an estimate $h'$ of the optimal model for the training set $S_i^s$ is produced. Then the test set $T_i^s$ is used for finding a model with a suitable loss function $L$ as the Eq. 56 shows.

$$h^* = \arg \min_h \sum_{(x_k, y_k) \in T_i^s} L[y_k, h'(x_k)] \tag{56}$$

To adopt meta-learning in OSR, a classification loss in combination with open-set loss is used. Moreover, the test set $T_i^s$ is augmented with some unseen samples. The overall loss function is defined as Eq. 57, where $L_c$ is a simple classification loss, and $L_o$ maximizes the entropy of unknown samples on the known classes.

$$h^* = \arg \min_h \Bigg\{ \sum_{(x_k, y_k) \in C_i^s \in T_i^s | y_k} L_c[y_k, h'(x_k)] \\ + \lambda \cdot \sum_{(x_k, y_k) \in T_i^s | y_k \in C_i^u} L_o[h'(x_k)] \Bigg\} \tag{57}$$

At the test time, the average features of correctly classified samples is obtained as a prototype point and used for the rejection of unseen samples.

### 4.14 Learning Placeholders for Open-Set Recognition (PROSER) (196):

This work tries to train a classifier ($\hat{f}(x)$) that can distinguish between target class and non-target classes. A dummy classifier is added to the softmax layer of the model with a shared feature extractor. Then it is forced to have the second maximum value for the correctly classified samples. When the classifier encounters novel inputs, the dummy classifier produces high values since all known classes are non-targets. Dummy classifier can be seen as the instance-dependent threshold which can well fit every known class. The loss function is defined as the Eq. 58, where $\hat{f}((x)/y$ is meant to make the most probable class zero.

$$L_1 = \sum_{(x,y) \in D_{tr}} l(\hat{f}(x), y) + \beta l(\hat{f}((x)/y, k+1) \tag{58}$$

Moreover, the mixup technique (169) is added to the loss function to boost unseen samples detection. The mixup representations are introduced to approximate the unseen sample's distribution and should be classified as the dummy class $k+1$. Finally, rejecting each test time sample is done based on the probability of the dummy class.

### 4.15 Counterfactual Zero-Shot and Open-Set Visual Recognition (187):

This work attempts to make abnormal samples in a counter-factual faithful way. As the paper mentions, most of the generative approaches such as G-OpenMax do not produce desired fake samples, the distributions of which do not resemble real distribution of unseen samples. To this end, a $\beta$-VAE (66) is used to make sample attribute variable $Z$ and the class attribute variable $Y$ independent. The $\beta$-VAE loss function is similar to simple VAE; however, the $KL$ term is induced by a coefficient $\beta$. This is shown to be highly effective in learning a disentangled sample attribute $Z$ (66). For disentangling $Y$ from $Z$, the proposed approach makes counter-factual samples by changing the variable $Y$ to have a large distance with the given input $x$ in spite of samples that are generated by changing the variable $Z$. To make counter-factual samples faithful, a Wasserstein GAN (4) loss is used for a discriminator $D(X, Y)$, which verifies the correspondence between the generated counter-factual image and the assigned label. At last, generated samples can be used to boost the performance of any OSR problem.

### 4.16 OpenGAN: Open-Set Recognition via Open Data Generation (82):

This work argues that using an easily accessible auxiliary outlier dataset can improve the open-set recognition performance as mentioned in (137) and (58). However, the auxiliary dataset might poorly generalize to diverse open-set data that are likely to be faced. Therefore, close to (189) a GAN-based approach is employed to generate fake samples that are supposed to mimic the distribution of the open-set data. Although, as mentioned in (189), GAN-based methods suffer from instability when used in the one-class training setup, in the OSR setup, labeled closed-set data is available which can be used to find the best stopping point during training. Furthermore, fake samples can be generated according to the feature space of closed-set samples passed through a pre-trained K-way classification model, which significantly reduces the dimensionality of the problem and helps stability. This work shows that a well-selected GAN-discriminator ($\mathcal{D}$) using a validation set achieves SOTA on OSR. Fig. 28 shows that at first, a closed-set classifier is trained on the given training dataset and is kept fixed afterward. Then, a GAN-based framework is trained on the embeddings of the closed-set and outlier images, which forces the generator ($\mathcal{G}$) to produce semantically fake outputs that are hard to be detected as fake for the discriminator. Eq. 59 shows the mathematical formulation of the loss function, where $\lambda_G$ controls the contribution of generated fake samples by G, and $\lambda_o$ adjusts the effect of auxiliary outlier dataset.

$$\max_{\mathcal{D}} \min_{G} \mathbb{E}_{x \sim P_{\text{closed}}} \Big[ \log(\mathcal{D}(x)) \Big] + \lambda_o \cdot \mathbb{E}_{\hat{x} \sim P_{\text{open}}} \Big[ \log(1 - \mathcal{D}(\hat{x}) \Big] + \lambda_G \cdot \mathbb{E}_{z \sim N} \Big[ \log(1 - \mathcal{D}(G(z))) \Big] \quad (59)$$

Figure 28: An overview of the OpenGAN method. Unlike pure GAN-based methods or Outlier Exposure, fake samples are generated in the latent space while a discriminator is trained to distinguish between the closed-set samples' embedding and outlier inputs. Closed-set embeddings are produced using a trained K-way closed-set classification model that is kept frozen afterward. In this way, the fake generator $G$ is pushed towards making features, that match outlier images' embeddings. The figure is taken from (82).

### 4.17 Open-set Recognition: A Good Closed-set Classifier Is All You Need? (167):

This work provides a comprehensive investigation through the correlation of closed-set accuracy and open-set performance of different deep architectures. It is shown that not only the closed-set accuracy of a model is highly correlated with its OSR performance, but also this correlation holds across a variety of loss objectives and architectures. This is surprising because stronger closed-set classifiers are more likely to overfit to the training classes and perform poorly in OSR. Fig. 29 illustrates the mentioned relationship for different architectures and within the ResNet family architectures. Another contribution of this work is providing a Semantic Shift Benchmark (SSB), which unlike common approaches of utilizing a single test set for the evaluation, makes two 'Easy' and 'Hard' sets based on the semantic similarity of the open-set categories to the training classes. The ability of a model to detect semantic novelty as opposed to the low-level distributional shift is better captured in this manner.

## 5 Out-of-distribution Detection

OOD detection aims to identify test-times samples that are semantically different from the training data categories, and therefore should not be predicted into the known classes. For instance, one could train the model on CIFAR-10 (as in-distribution data), and then evaluate on CIFAR-100 (113) as an out-of-distribution

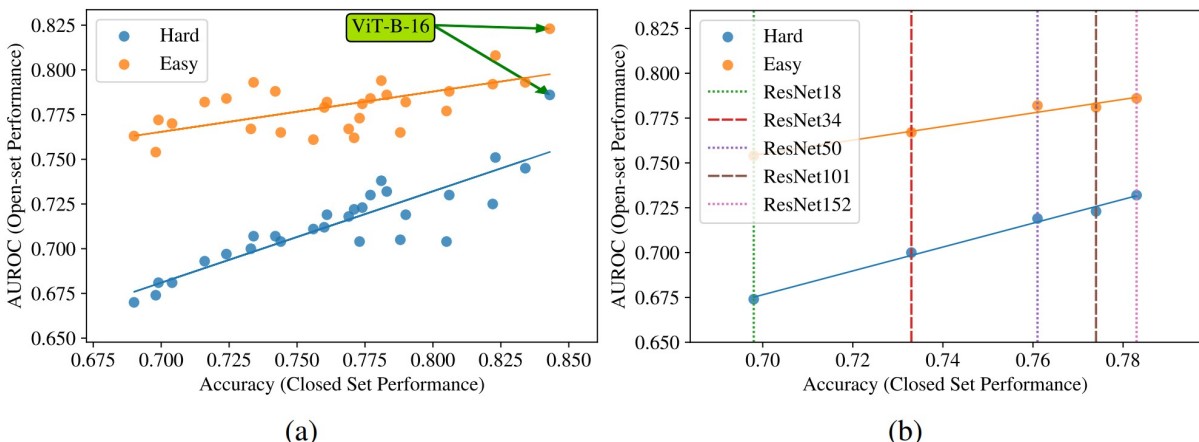

Figure 29: (a) The performance of different deep model architectures on the ImageNet dataset. The ImageNet-21K-P dataset is used to generate 'Easy' and 'Hard' OSR splits. (b) The performance of different ResNet families on the ImageNet. The figure is taken from (167).

dataset, as CIFAR-10 and CIFAR-100 have mutually exclusive classes. In the multi-class setting, the problem of OOD detection is canonical to OSR: accurately classifying samples from the known classes while detecting the unknowns. However, OOD detection encompasses a broader spectrum of learning tasks (e.g., multi-label classification) and solution space (e.g., density estimation without classification). Some approaches relax the constraints imposed by OSR and achieve strong performance. The following reviews some of recent OOD detection works and their differences.

## 5.1 A Baseline for Detecting Misclassified and Out-of-Distribution Examples in Neural Networks (56):

This work coined "out-of-distribution detection" and showed how to evaluate deep learning out-of-distribution detectors. As previous anomaly detection works for deep classifiers often had low-quality or proprietary datasets, existing datasets were re-purposed to create out-of-distribution datasets, enabling easier evaluation. This approach proposes using the maximum softmax probability (MSP) to detect out-of-distribution samples, namely $\max_k p(y = k \mid x)$. A test sample with a large MSP score is detected as an in-distribution (ID) example rather than out-of-distribution (OOD) example. This showed that a simple maximum probability score can be useful for detection in vision, natural language processing, and speech recognition settings, but there is much room for improvement. It also showed $p(y \mid x)$ models can be useful for out-of-distribution detection and that $p(x)$ models are not necessarily needed. Until now, it still serves as a general-purpose baseline that is nontrivial to surpass. Concurrent OSR work proposed additional modifications to softmax probabilities for detection (11).

## 5.2 Enhancing The Reliability of Out-of-distribution Image Detection in Neural Networks (ODIN) (92):

In this work, a technique called temperature scaling was employed. Although it has been used in other domains such as knowledge distillation (67), the main novelty of this work is showing the usefulness of this technique in the OOD domain. In temperature scaling, the softmax score is computed as in Eq. 60. OOD samples are then detected at the test time based on thresholding the maximum class probability. This simple approach, in combination with adding a small controlled noise, has shown significant improvement compared to the baseline approach MSP. ODIN further shows that adding one step gradient to the inputs in the direction of improving the maximum score has more effect on the in-class samples, and pushes them to have larger margin with the OOD samples.

$$S_i(x; T) = \frac{\exp(f_i(x)/T)}{\sum_{j=1}^{N} \exp(f_j(x)/T)} \tag{60}$$

The paper also provided mathematical explanation for the effect of temperature scaling on out-of-distribution detection. This can be seen in the Taylor approximation of the softmax score (expanded around the largest logit output $f_{\hat{y}}(x)$):

$$
\begin{aligned}
S_i(x; T) &= \frac{\exp(f_i(x)/T)}{\sum_{j=1}^{N} \exp(f_j(x)/T)} \\
&= \frac{1}{\sum_{j=1}^{N} \exp(\frac{f_j(x) - f_i(x)}{T})} \\
&\approx \frac{1}{N - \frac{1}{T} \sum_{j=1}^{N} [f_{\hat{y}}(x) - f_j(x)] + \frac{1}{2T^2} \sum_{j=1}^{N} [f_{\hat{y}}(x) - f_j(x)]^2}
\end{aligned}
\tag{61}
$$

A sufficiently large temperature $T$ has a strong smoothing effect that transforms the softmax score back to the logit space—which effectively distinguishes ID vs. OOD. In particular, the ID/OOD separability is determined by the $U_1 = \sum_{j=1, j \neq \hat{y}}^{N} [f_{\hat{y}}(x) - f_j(x)]$ and $U_2 = \sum_{j=1, j \neq \hat{y}}^{N} [f_{\hat{y}}(x) - f_j(x)]^2$. The former measures the extent to which the largest unnormalized output of the neural network deviates from the remaining outputs; while the latter measures the extent to which the remaining smaller outputs deviate from each other. For the in-class samples, $U_1$ and $E[U_2 \mid U_1]$ are higher than OOD ones. Mathematically and empirically, ODIN is not sensitive to $T$ when it is large enough to satisfy the Taylor approximation. For example, the paper shows that simply applying $T = 1000$ can yield effective performance boost without hyper-parameter tuning. In general, using temperature scaling can improve the separability more significantly than input preprocessing.

Note that ODIN differs from confidence calibration, where a much milder $T$ is employed. While calibration focuses on representing the true correctness likelihood of ID data only, the ODIN score is designed to maximize the gap between ID and OOD data and may no longer be meaningful from a predictive confidence standpoint. As seen in Fig. 30, the ODIN scores are closer to $1/N$ where $N$ is the number of classes.

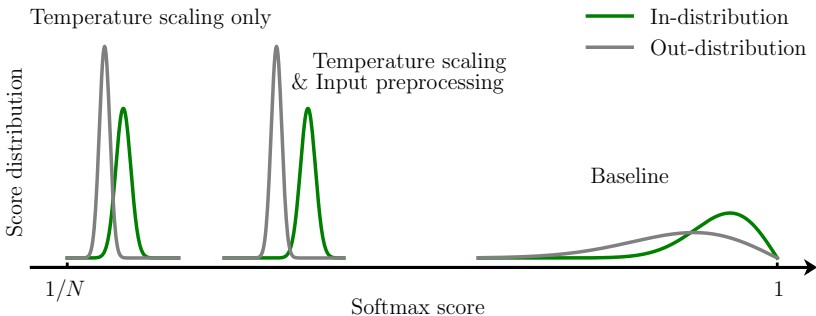

Figure 30: An overview of ODIN, a post-hoc method that uses temperature scaling and input perturbation to amplify the ID/OOD separability. The figure is taken from (92).

### 5.3 A Simple Unified Framework for Detecting Out-of-Distribution Samples and Adversarial Attacks (89):

This work was inspired from the idea of Linear Discriminant Analysis (LDA) in which $P(X = x \mid Y = y)$ is considered to be a multivariate Gaussian distribution. In order for $P(Y = y \mid X = x)$ to be similar to a softmax form, it is assumed that the feature space of the penultimate layer follows the Gaussian distribution. Therefore, a mean and variance vector is simply estimated from features of each class, and a multivariate Gaussian is fit to them. In order to check validity of the assumptions, it uses the Mahalanobis distance of the test time images to perform the classification instead of the softmax function. Surprisingly, the results are comparable or better than softmax, which supports the assumptions. It performs OOD detection using Mahalanobis distance to the closest class-conditional Gaussian distribution. Furthermore, to improve the performance, features in different layers are ensembled and a small controlled noise is added to test samples

as shown in Eq. 62, where $M(x)$ is the Mahalanobis distance with mean of the closest class-conditional Gaussian distribution. A similar idea has been discussed earlier in (134).

$$\hat{x} = x + \epsilon \cdot \text{sign}(\nabla_x M(x)) \tag{62}$$

### 5.4 Predictive Uncertainty Estimation via Prior Networks (DPN) (102):

This work discusses three different sources of uncertainty: (1) data uncertainty, (2) distributional uncertainty, and (3) model uncertainty. The importance of breaking down the final uncertainty into these terms was discussed. For instance, model uncertainty might happen because of the model's lack of capacity to approximate the given distribution well. On the other hand, data uncertainty might happen because of the intrinsic intersection of similar classes. For instance, classifying between different kinds of dogs has more data uncertainty than solving a classification problem with completely separate classes. Distributional uncertainty is related to the problem of AD, ND, OSR, and OOD detection. The goal of this work was to estimate the distributional uncertainty for each input and compare it with the data and model uncertainties. Data uncertainty $P(w_c \mid x^*, \theta)$ can be defined by the posterior distribution over class labels given a set of parameters $\theta$. Model uncertainty $P(\theta \mid D)$ is defined by the posterior distribution over the parameter given data $D$. These two types of uncertainties could be combined to give rise to the distributional uncertainty, as shown in Eq. 63.

$$P(w_c \mid x^*, D) = \int P(w_c \mid x^*, \theta) P(\theta \mid D) \, d\theta \tag{63}$$

As computing the integral is not tractable, the above formula is usually converted into Eq. 64, where $q(\theta)$ is an approximation of $P(\theta \mid D)$.

$$P(w_c \mid x^*, D) \approx \frac{1}{M} \sum_{i=1}^{M} P(w_c \mid x^*, \theta^i), \theta^i \sim q(\theta) \tag{64}$$

Each $P(w_c \mid x^*, D)$ can then be seen as a categorical distribution located on a simplex, as shown in Fig. 31.

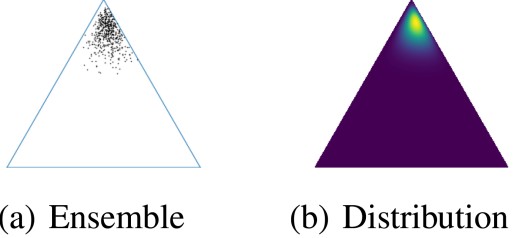

(a) Ensemble      (b) Distribution

Figure 31: The figure shows a distribution over the categorical distributions for modeling uncertainty using both model uncertainty and data uncertainty. The figure is taken from (102).

Now to extract distributional uncertainty from the model uncertainty, Eq. 64 is decomposed into the following Eq. 65.

$$P(w_c \mid x^*, D) = \iint P(w_c \mid \mu) P(\mu \mid x^*, \theta) P(\theta \mid D) \, d\mu \, d\theta, \tag{65}$$

where $P(w_c \mid \mu)$ is a categorical distribution given a realization from a Dirichlet distribution, $P(\mu \mid x^*, \theta)$ is the Dirichlet distribution given the input and model parameters $\theta$, and $P(\theta \mid D)$ is the distribution over model parameters given the dataset $D$. For simplicity, in this work $P(\theta \mid D)$ is equal to $\delta(\theta - \hat{\theta})$ and is produced by the output of a deep network. Therefore $P(\mu \mid x^*, \theta) = P(\mu \mid x^*, \hat{\theta}) = \text{Dir}(\mu \mid \alpha)$, and $\alpha = f(x^*, \hat{\theta})$, where $f(.,.)$ is represented by a deep neural network.

At the training time, the Dirichlet Prior Network (DPN) is expected to yield a flat distribution over the simplex for OOD samples, indicating large uncertainty in the mapping from $x$ to $y$. Some out-of-distribution data is used to minimize the KL distance of $\mathrm{Dir}(\mu \mid \alpha)$ and the flat Dirichlet distribution. For the in-class samples, the KL divergence between $\mathrm{Dir}(\mu \mid \alpha)$ and a sharp, sparse Dirichlet distribution is minimized. The objective Dirichlet distributions are obtained by pre-setting their parameters during training process. During test time, different criterion such as max probability, last layer's entropy ($H(.)$), and distributional uncertainty as in Eq. 66 are used for OOD detection.

$$
\begin{aligned}
I(y, \mu \mid x^*, D) = {} & H[E_{P(\mu \mid x^*, D)}[P(y \mid \mu)]] \\
& - E_{P(\mu \mid x^*, D)}[H[P(y \mid \mu)]]
\end{aligned}
\tag{66}
$$

### 5.5 Confidence-calibrated Classifiers for Detecting Out-of-distribution Samples (88):

This work attempted to maximize the entropy of confidence scores for OOD samples, similar to (33). Similar to (111), it generates OOD samples by jointly training a GAN and a classifier. As shown in Eq. 67, the first term solves a classification task on in-class samples, the second term uses KL divergence to make the confidence score distribution of generated OOD samples uniform. The remainder terms train the GAN on the in-class samples. Note that, the GAN is forced to generate high-quality OOD samples that produce high uncertainty when they are passed to the classifier. Therefore, the generated samples are located on the boundaries of in-class and outlier distributions. The paper also shows that leveraging on-boundary in-class samples significantly improves its confidence calibration.

$$
\begin{aligned}
& \min_G \max_D \min_\theta E_{P_{\mathrm{in}}(\hat{x}, \hat{y})}[-\log P_\theta(y = \hat{y} \mid \hat{x})] \\
& + \beta E_{P_G(x)}[\mathrm{KL}(U(y) \| P_\theta(y \mid x))] + E_{P_{\mathrm{in}}(\hat{x})}[\log D(x)] \\
& + E_{P_G(x)}[\log(1 - D(x))]
\end{aligned}
\tag{67}
$$

$P_{\mathrm{in}}$ denotes the in-class distribution and $P_\theta(y \mid x)$ is a classifier trained on the dataset drawn from $P_{\mathrm{in}}(x, y)$. Test time OOD detection is done based on thresholding of the maximum softmax value.

### 5.6 Deep Anomaly Detection with Outlier Exposure (OE) (58):

This work introduced Outlier Exposure (OE) and reported extensive experiments on its usefulness for various settings. When applied to classifiers, the Outlier Exposure loss encourages models to output a uniform softmax distribution on outliers, following (88). More generally, the Outlier Exposure objective is

$$
\mathbb{E}_{(x,y) \sim \mathcal{D}_{\mathrm{in}}}[\mathcal{L}(f(x), y) + \lambda \cdot \mathbb{E}_{x' \sim \mathcal{D}_{\mathrm{out}}^{\mathrm{OE}}}[\mathcal{L}_{\mathrm{OE}}(f(x'), f(x), y)]],
$$

assuming a model $f$, the original learning objective $\mathcal{L}$; when labeled data is not available, $y$ can be ignored. Models trained with this objective can have their maximum softmax probabilities (56) better separate in- and out-of-distribution examples. To create $\mathcal{D}_{\mathrm{out}}^{\mathrm{OE}}$, data unlike the training data will need to be scraped or curated or downloaded. Samples from $\mathcal{D}_{\mathrm{out}}^{\mathrm{OE}}$ are gathered from already existing and available datasets that might not be directly related to the task-specific objective function; however, they can significantly improve the performance because they contain many diverse variations. Concurrent work (33) explores a similar intuition for small-scale image classification, while Outlier Exposure shows how to improve OOD detection for density estimation models, natural language processing models, and small- and large-scale image classifiers.

### 5.7 Using Self-Supervised Learning Can Improve Model Robustness and Uncertainty (60):

This work investigated the benefits of training supervised learning tasks in combination with SSL methods in improving robustness of classifiers against simple distributional shifts and OOD detection tasks. To do so, an auxiliary rotation prediction was added to a simple supervised classification. The work measures robustness against simple corruptions such as Gaussian noise, shot noise, blurring, zooming, fogging etc. It has been observed that although auxiliary SSL tasks do not improve the classification accuracy, the model's robustness and detection abilities are significantly improved. Additionally, when the total loss function is trained in an

adversarially robust way, the robust accuracy is improved. Finally, the method is tested in the ND setting using rotation prediction, and horizontal and vertical translation prediction similar but simpler than GT or GOAD. They also test in the setting of multiclass classification setting and find that auxiliary self-supervised learning objectives improves the maximum softmax probability detector (56). In addition, similar to (33) and (88), they attempted to make distribution of the confidence layer for some background or outlier samples uniform. Outliers are selected from other accessible datasets, as in Outlier Exposure (58).

### 5.8 Unsupervised Out-of-Distribution Detection by Maximum Classifier Discrepancy (185):

This work relies on a surprising fact—two classifiers trained with different random initializations can act differently on unseen test time samples at their confidence layers. Motivated by this, the work attempts to increase the discrepancy on unseen samples and reduce the discrepancy on seen ones. The discrepancy loss is the difference between the first classifier's last layer entropy and that of the second one. This forces the classifiers to have the same confidence scores for in-class inputs, yet increases their discrepancy for the others. Fig. 32 shows the overall architecture. First, two classifiers are trained on the in-class samples and are encouraged to produce the same confidence scores. Second, an unlabeled dataset containing both OOD and in-class data is employed to maximize their discrepancy on outliers while preserving their consistency on inliers.

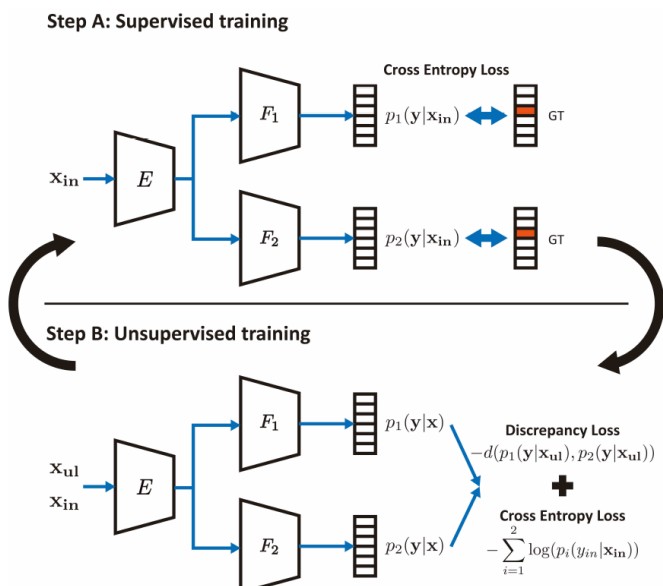

Figure 32: The overall architecture of (185). As it can be seen, at the first step, both of the classifiers are trained. Then, at the next step, an auxiliary discrepancy loss is added to the supervised classification to adjust the boundaries of in-class and OOD samples. The figure is taken from (185).

### 5.9 Why ReLU Networks Yield High-Confidence Predictions Far Away From the Training Data (54):

This work proved that ReLU networks produce piece-wise affine functions; therefore, they can be written as $f(x) = V^l x + a^l$ on the polytope $Q(x)$ as follows:

$$
\begin{aligned}
\Gamma_{l,i} &= \{z \in R^d \mid \Delta^l(x)(V_i^l z + a_i^l \geq 0) \\
Q(x) &= \cap_{l=1,\ldots,L} \cap_{i=1,\ldots,n_l} \Gamma_{l,i},
\end{aligned}
\tag{68}
$$

where $\Delta^{(l)}$ is a diagonal matrice defined elementwise as:

$$
\Delta^{(l)}(x)_{ij} = \begin{cases} \operatorname{sign}\left(f_i^{(l)}(x)\right) & \text{if } i = j, \\ 0 & \text{else.} \end{cases}
\tag{69}
$$

$n_l$ and $L$ are the number of hidden units in the $l$th layer and the total number of layers, respectively. The following theorem proves the deficiency of ReLU networks.

*Theorem. 1* Let $R^d = \cup_{l=1}^R Q_l$ and $f(x) = V^l x + a^l$ be the piecewise affine representation of the output of a ReLU network on $Q_l$. Suppose that $V^l$ does not contain identical rows for all $l = 1, ..., R$ Then for almost any $x \in R^d$ and $\epsilon \geq 0$ there exists an $\alpha$ and a class $k \in \{1, ..., K\}$ such that for $z = \alpha x$ it holds

$$\frac{e^{f_k(z)}}{\sum_{r=1}^K e^{f_r(z)}} \geq 1 - \epsilon \tag{70}$$

The equation goes to 1 if $\alpha \to \infty$. From this, we can imply that for ReLU networks there exist infinitely many inputs which yield high confidence predictions. Note that arbitrarily high confidence prediction can not be obtained due to the bounded domain of inputs. In order to relieve this problem, a technique that is called confidence enhancing data augmentation is used, as the Eq. 71 shows.

$$\frac{1}{N} \sum_{i=1}^N L_{\text{CE}}(y_i, f(x_i)) + \lambda \cdot E[L_{p_{\text{out}}}(f, Z)]$$

$$L_{p_{\text{out}}} = \max_{l=1,...,K} \log\left(\frac{e^{f_l(z)}}{\sum_{k=1}^K e^{f_k(z)}}\right) \tag{71}$$

where $p_{\text{out}}$ and $p_{\text{in}}$ are in-class and out-of-distribution distributions respectively, which we are certain that the set of the intersection of their supports has zero or close to zero probability mass. An example of such an out-distribution pout would be the uniform distribution on $[0, 1]^{w \times h}$ or other noise distributions.

The above training objective needs many samples to enforce low confidence on the entire out-distribution, an alternative technique called adversarial confidence enhancing training (ACET) is used. ACET uses the idea of adversarial robust training to not only minimize the objective function at each point but also the worst case in a neighborhood of the point as Eq. 72 shows.

$$\frac{1}{N} \sum_{i=1}^N L_{\text{CE}}(y_i, f(x_i)) + \lambda \cdot E[\max_{||u-Z||_p \leq \epsilon} L_{p_{\text{out}}}(f, u)] \tag{72}$$

At the test time, a thresholded confidence score is used to distinguish between inliers and outliers.

### 5.10 Do Deep Generative Models Know What They Don't Know? (110):

This work shows that generative models surprisingly assign higher likelihood scores to outliers. This holds for VAEs, auto-regressive models, and different kinds of flow-based methods. In the generative modeling, usually, a parametric model on the data distribution is assumed as $p_\theta(x)$. Then, it finds the best $\theta$ that minimizes the KL distance between the true but unknown distribution $p^*$ and $p$, which is equal to maximizing the likelihood of $p_\theta(x)$ on the input distribution, as the Eq. 73 shows.

$$\text{KL}[p^* || p_\theta(x)] = \int p^*(x) \log \frac{p^*(x)}{p_\theta(x)} \, dx \approx -\frac{1}{N} \log p_\theta(X) - H[p^*] \tag{73}$$

Also, assuming the existence of a latent space $Z$ the integrals can be written as Eq. 74 where $Z$ and $X$ are hidden and input variables, respectively. Let $f$ be a diffeomorphism from the data space to a latent space, which commonly happens in flow-based modelings, where $|\frac{\partial f}{\partial x}|$ is known as the volume element.

$$\int_z p_z(z)\, dz = \int_x p_z(f(x)) \left| \frac{\partial f}{\partial x} \right| dx = \int_x p_x(x)\, dx \tag{74}$$

The parameters of $p$ can be decomposed as $\theta = \{\phi, \psi\}$ with $\phi$ being the diffeomorphism's parameters, i.e. $f(x; \phi)$, and $\psi$ being the auxiliary distribution's parameters, i.e. $p(z; \psi)$. Then the objective function can be written as Eq. 75.

$$\theta^* = \arg\max_\theta \log p_\theta(X) =$$
$$\arg\max_{\phi, \psi} \sum_{i=1}^{N} \log p_z(f(x_n, \phi), \psi) + \log \left| \frac{\partial f_\phi}{\partial x_n} \right| \tag{75}$$

An interesting aspect of the objective function above is that it encourages the function $f$ to have high sensitivity to small changes of input samples $x_n$. It is mentioned in the paper that if we plot the effect of each term separately, the first term shows the desired behavior for inliers and outliers, but the second term causes the problem. Changing $f$ to constant-volume (CV) transformations (34) can alleviate the problem, but not entirely. Finally, using the second-order expansion of the log-likelihood around an interior point $x_0$, it has been shown that the assigned likelihoods have a direct relation to the model curvature and data's second moment. Therefore, the problem of generative models might be fundamental. Eq. 76 shows the details of expansion where $Tr$ means trace operation.

$$0 \leq E_q[\log p(x, \theta)] - E_{p^*}[\log p(x, \theta)] \approx$$
$$\nabla_{x_0} \log p(x_0, \theta)^T (E_q[x] - E_{p^*}[x])$$
$$+ \frac{1}{2} \mathrm{Tr}\{\nabla_{x_0}^2 \log p(x_0, \theta)(\Sigma_q - \Sigma_{p^*})\} \tag{76}$$

## 5.11 Likelihood Ratios for Out-of-Distribution Detection (132):

This paper employs likelihood ratio to alleviate the problem of OOD detection in generative models. The key idea is to model the background ($X_B$) and foreground ($X_F$) information separately. Intuitively, background information is assumed to be less harmed than foreground information when semantically irrelevant information are added to the input distribution. Therefore, two autoregressive models are trained on noisy and original input distribution, and their likelihood ratio is defined as Eq. 77.

$$\mathrm{LLR}(x) = \log \frac{p_\theta(X)}{p_{\theta_0}(X)} = \log \frac{p_\theta(X_B) p_\theta(X_F)}{p_{\theta_0}(X_B) p_{\theta_0}(X_F)}$$
$$\log \frac{p_\theta(X_F)}{p_{\theta_0}(X_F)} = \log p_\theta(X_F) - \log p_{\theta_0}(X_F) \tag{77}$$

$p_\theta(.)$ is the model trained on the in-distribution data, and $p_{\theta_0}(.)$ is the background model trying to estimate the general background statistics.

At the test time, a thresholding method is used on the likelihood ratio score.

## 5.12 Generalized ODIN (68):

As an extension to ODIN (92), this work proposes a specialized network to learn temperature scaling and a strategy to choose perturbation magnitude. G-ODIN defines an explicit binary domain variable $d \in \{d_{in}, d_{out}\}$, representing whether the input $x$ is inlier (i.e, $x \sim p_{in}$). The posterior distribution can be decomposed into

$p(y \mid d_{\text{in}}, x) = \frac{p(y, d_{\text{in}}|x)}{p(d_{\text{in}}|x)}$. Note that in this formulation, the reason for assigning overconfident scores to outliers seems more obvious, since the small values of $p(y, d_{\text{in}} \mid x)$ and $p(d_{\text{in}} \mid x)$ produce the high value of $p(y \mid d_{\text{in}}, x)$. Therefore, they are decomposed and estimated using different heads of a shared feature extractor network as $h_i(x)$ and $g(x)$ for $p(y \mid d_{\text{in}}, x)$ and $p(d_{\text{in}} \mid x)$ respectively. This structure is called dividend/divisor and the logit $f_i(x)$ for class $i$ can be written as $\frac{h_i(x)}{g(x)}$. The objective loss function is a simple cross entropy, similar to previous approaches. Note that the loss can be minimized by increasing $h_i(x)$ or decreasing $g(x)$. For instance, when the data is not in the high-density areas of in-distribution, $h_i(x)$ might be small; therefore, $g(x)$ is forced to be small to minimize the objective function. In the other case, $g(x)$ are encouraged to have larger values. Therefore, they approximate the role of aforementioned distributions $p(y \mid d_{\text{in}}, x)$ and $p(d_{\text{in}} \mid x)$. At the test time $\max_i h_i(x)$ or $g(x)$ are used. Fig. 33 shows an overview of the method.

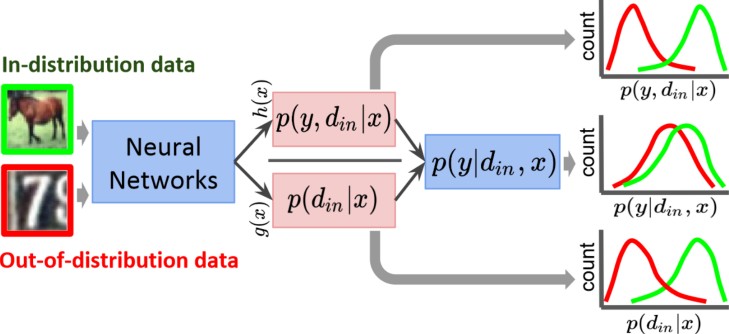

Figure 33: The overall architecture of Generalized ODIN. As evident, different heads are applied on the penultimate layer to model the mentioned distributions. The figure is taken from (68).

### 5.13  Background Data Resampling for Outlier-Aware Classification (91):

As mentioned before, in AD, ND, OSR, and OOD detection, some methods use a background or outlier dataset to boost their performances. However, to avoid different kinds of biases, the size of auxiliary datasets becomes important. In this work, a re-sampling technique is proposed to select an optimal number of training samples from the outlier dataset such that on-boundary samples play a more influential role in the optimization task. The work first provided an interesting probabilistic interpretation of outlier exposure technique. The loss function can be written as Eq. 80, where $L_{\text{cls}}$ and $L_{\text{uni}}$ are shown in Eq. 78 and Eq. 79 respectively.

$$L_{\text{cls}}(f(x; \theta), y) = -\log f_y(x; \theta) \tag{78}$$

$$L_{\text{uni}}(f(x; \theta)) = -\frac{1}{K} \sum_{k=1}^{K} \log f_k(x; \theta) - \log K \tag{79}$$

$$L(\theta; p, q) = E_{X,Y \sim p(.,.)}[L_{\text{cls}}(f(X; \theta); Y)] \\ + \alpha E_{X \sim q(\cdot)}[L_{\text{uni}}(f(X; \theta))]. \tag{80}$$

By expanding Eq. 80 and taking its gradient w.r.t. classifier $(f(x))$ logits, the optimal classifier is obtained as Eq. 81 shows.

$$f_k^*(x) = c(x)p_{Y|X}(k \mid x) + \frac{1 - c(x)}{K} \\ c(x) = \frac{p_X(x)}{p_X(x) + \alpha q_X(x)}, \tag{81}$$

where $c(x)$ can be seen as the relative probability of a sample $x$ being in-class distribution $p$ or background distribution $q$ with the ratio $\alpha$. Suppose $D'_b$ is the re-sampled dataset and $D_b$ is the given background one, OOD detection loss after reweighting can be written as:

$$L_{\text{out}}(\theta; w) = \frac{1}{|D'_b|} \sum_{(x,y) \in D'_b} L_{\text{uni}}(f(x; \theta))$$
$$= \frac{1}{\sum_{i=1}^{|D_b|}} w_i L_{\text{uni}}(f(x; \theta)) \tag{82}$$

The optimal parameters $\theta^*$ are learned as follows:

$$\theta^*(w) = \arg\ \min_\theta L(\theta; D, w)$$
$$= \arg\ \min_\theta L_{\text{in}}(\theta; D) + \alpha L_{\text{out}}(\theta; w) \tag{83}$$

Therefore, an iterating optimization is solved between finding $\theta^t$ and $w^t$ at each step by fixing one and solving the optimization problem for the other. At last, the largest values of weights can be selected with regard to the dataset's desired size.

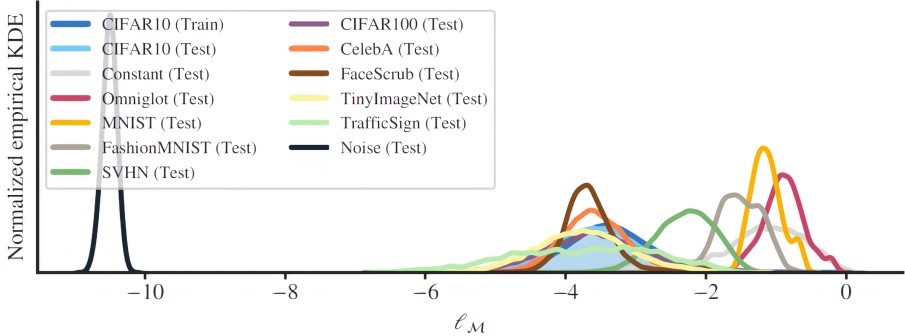

Figure 34: The assigned likelihood values of a simple generative model to different datasets when trained on CIFAR10. As evident, simpler datasets achieve higher values. The figure is taken from (152).

### 5.14 Input Complexity and Out-of-Distribution Detection With Likelihood-Based Generative Models (152):

This work further investigated the problem of generative models assigning higher likelihood values to OOD samples. In particular, this work finds a strong tie between the OOD samples' complexity and likelihood values. Simpler input can lead to higher likelihood value. Fig. 35 shows this phenomenon. Furthermore, another experiment to support the claim is designed that starts from a random noise, on which a mean average pooling is applied at each step. To preserve the dimension, an upscaling is done after average poolings. Surprisingly, simpler images on which more average poolings are applied to achieve higher likelihoods. Motivated by this, the work proposed to detect OOD samples by accounting for input complexity in combination with likelihood values. Since it is hard to compute the input complexity, the paper instead calculates an upper bound using a lossless compression algorithm (28). Given a set of inputs $\mathbf{x}$ coded with the same bit depth, the normalized size of their compressed versions, $L(x)$ (in bits per dimension), is used as the complexity measurement. Finally, the OOD score is defined as Eq. 84:

$$S(x) = -l_M(x) - L(x), \tag{84}$$

where $l_M(x)$ is the log-likelihood of an input $x$ given a model $M$.

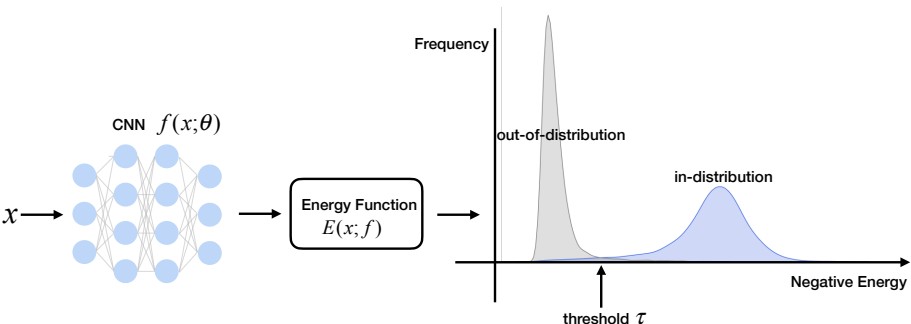

Figure 35: The energy-based OOD detection framework. The energy function maps the logit outputs to a scalar through a convenient `logsumexp` operator. Test samples with lower energy are considered ID and vice versa. The figure is taken from (95).

Intuitively, considering $M_0$ as a universal compressor, then $p(x \mid M_0) = 2^{-L(x)}$ and Consequently, the OOD score can be defined as follows :

$$S(x) = -\log_2 p(x \mid M) + \log_2 p(x \mid M_0) = \log_2 \frac{p(x \mid M_0)}{p(x \mid M)} \tag{85}$$

In the cases where a simple OOD sample is fed into the model, the universal compressor $M_0$ assigns a high probability to it and effectively corrects the high likelihood, wrongly given by the learned model $M$. Similar interpretation holds for the complex OOD samples too.

### 5.15 Energy-based Out-of-distribution Detection (95):

This work proposes using the energy score derived from the logit outputs for OOD detection, and demonstrated superiority over softmax score. Energy-based models map each input $\mathbf{x}$ to a single deterministic point that is called energy (86). A set of energy values $E(\mathbf{x}, y)$ could be turned into a density function $p(x)$ through the Gibbs distribution:

$$p(y \mid x) = \frac{e^{-E(\mathbf{x},y)/T}}{\int_{y'} e^{-E(\mathbf{x},y')/T}} = \frac{e^{-E(\mathbf{x},y)/T}}{e^{-E(\mathbf{x})/T}}, \tag{86}$$

where $T$ is the temperature parameter, $\mathbf{x}$ is input data, and $E(\mathbf{x})$ is called *Helmholtz free energy* and is equal to:

$$E(\mathbf{x}) = -T \cdot \log \int_{y'} e^{-E(\mathbf{x},y')/T}. \tag{87}$$

In deep networks, by considering $E(\mathbf{x}, y) = -f_y(\mathbf{x})$, one can express the free energy function in terms of the denominator of the softmax activation:

$$E(\mathbf{x}; f) = -T \cdot \log \sum_i^K e^{f_i(\mathbf{x})/T}. \tag{88}$$

The paper also shows that cross-entropy loss facilitates pushing down the energy for in-distribution data during the training process. Moreover, softmax scores can be analyzed through an energy-based perspective, as Eq. 89 shows. During the optimization, $E(\mathbf{x}; f)$ is forced to be small for in-distribution samples while being shifted by $f^{\max}(\mathbf{x})$ to satisfy the maximum function. Consequently, this results in a biased scoring

function that is not suitable for OOD detection.

$$\max_y p(y \mid \mathbf{x}) = \max_y \frac{e^{f_y(\mathbf{x})}}{\sum_i e^{f_i(\mathbf{x})}} = \frac{e^{f^{\max}(\mathbf{x})}}{\sum_i e^{f_i(\mathbf{x})}}$$
$$= \frac{1}{\sum_i e^{f_i(\mathbf{x}) - f^{\max}(\mathbf{x})}} \tag{89}$$
$$\implies \log \max_y p(y \mid \mathbf{x}) = E(\mathbf{x}; f(\mathbf{x}) - f^{\max}(\mathbf{x}))$$
$$= E(\mathbf{x}; f) + f^{\max}(\mathbf{x}).$$

The energy score is hyperparameter-free, easy to compute, and achieves strong performance compared to the softmax score. Beyond post hoc OOD detection, the paper further demonstrated that energy score can be utilized for model regularization. Different from outlier exposure which forces the uniform softmax distribution for outlier training samples, energy-based regularization directly optimizes the energy gap between ID and OOD:

$$\min_\theta E_{(\mathbf{x},y) \sim D_{\text{in}}^{\text{train}}} [-\log F_y(\mathbf{x})] + \lambda \cdot L_{\text{energy}}$$
$$L_{\text{energy}} = E_{(\mathbf{x}_{\text{in}}, y) \sim D_{\text{in}}^{\text{train}}} [\max(0, E(\mathbf{x}_{\text{in}}) - m_{\text{in}})^2] \tag{90}$$
$$+ E_{\mathbf{x}_{\text{out}} \sim D_{\text{out}}^{\text{train}}} [\max(0, m_{\text{out}} - E(\mathbf{x}_{\text{out}}))^2],$$

where $m$ is the margin hyper-parameter, and $F(\mathbf{x})$ is the softmax output of the classification model. $D_{\text{out}}^{\text{train}}$ is the unlabeled auxiliary OOD training data, and $D_{\text{in}}^{\text{train}}$ is the ID training data. The optimization results in a stronger performance than OE. At the test time, OOD samples are detected based on a threshold on $-E(\mathbf{x}; f)$.

### 5.16 Likelihood Regret: An Out-of-Distribution Detection Score for Variational Autoencoder (178):

Previous works showed that VAEs can reconstruct OOD samples perfectly, resulting in the difficulty in detecting OOD samples. The average test likelihoods of VAE across different datasets have a much smaller range than PixelCNN (116) or Glow(80), showing that distinguishing between OOD and inlier samples is much harder in VAE. The reason might be because of the different ways they model the input distribution. Auto-regressive and flow-based methods model their input at pixel level, while the bottleneck structure in VAE forces the model to ignore some information. To address this issue, a criterion called *likelihood regret* is proposed. It measures the discrepancy between a model trained to maximize the average likelihood of a training dataset, for instance, a simple VAE, and a model maximizing the likelihood of a single input image. The latter is called an ideal model for each sample. Intuitively, the likelihood difference between the trained model and the ideal one might not be high; however, this does not hold for OOD inputs. Suppose the following optimization is performed to train a simple VAE :

$$(\phi^*, \theta^*) \approx \arg \max_{\phi, \theta} \frac{1}{n} \sum_{i=1}^n L(x_i; \theta, \tau(x_i, \phi)), \tag{91}$$

where $\phi$ and $\theta$ are the parameters of encoder and decoder respectively, $\tau(x_i, \phi)$ denotes the sufficient statistics $(\mu_x, \sigma_x)$ of $q_\phi(z \mid x)$, and $L(x_i; \theta, \tau(x_i, \phi))$ expresses the ELBO. For obtaining the ideal model, the decoder can be fixed, and an optimization problem solved on $\tau(x_i, \phi)$ such that its individual ELBO is maximized:

$$\hat{\tau} = \arg \max_\tau L(x; \theta^*, \tau) \tag{92}$$

Finally, the likelihood regret is defined as follows:

$$\text{LR}(x) = L(x; \theta^*, \hat{\tau}(x)) - L(x; \theta^*, \phi^*) \tag{93}$$

The optimization of Eq. 92 could be started from the initial point that the encoder produces for each input, and a threshold is used on LR values at the test time.

### 5.17 Understanding Anomaly Detection With Deep Invertible Networks Through Hierarchies of Distributions and Features (146):

This work studied the problem of flow-based generative models for OOD detection. It was noted that local feature such as smooth local patches can dominate the likelihood. Consequently, smoother datasets such as SVHN achieve higher likelihoods than a less smooth one like CIFAR-10, irrespective of the training dataset. Another exciting experiment shows the better performance of a fully connected network than a convolutional Glow network in detecting OOD samples using likelihood value. This again supports the existence of a relationship between local statistics like continuity and likelihood values. Fig. 36 shows the similarity of different dataset local statistics that are computed based on the difference of a pixel value to the mean of its $3 \times 3$ neighboring pixels.

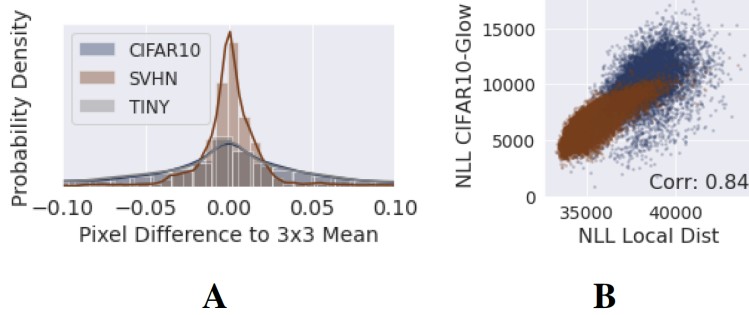

**A**      **B**

Figure 36: (A) shows the distribution of local pixel value differences. As it can be seen, distributions are highly overlapped. (B) Likelihoods extracted from the local pixel value differences correlate with CIFAR10-Glow likelihoods. The figure is taken from (146).

Furthermore, let us call the summation of the likelihood of pixel value differences computed in a local $3 \times 3$ neighboring pixels using a histogram with 100 equally distanced bins with pseudo-likelihoods. A strong Spearman correlation is found between the pseudo-likelihoods and the exact values of likelihoods, supporting the above assumptions. To address this problem, the following three steps are used:

- Train a generative network on a general image distribution like 80 Million Tiny Images

- Train another generative network on images drawn from the in-distribution, e.g., CIFAR-10

- Use their likelihood ratio for OOD detection

Also, to improve the efficacy, the following outlier loss, which uses OOD samples, can be added to the maximum likelihood objective function:

$$L_o = -\lambda. \log \big(\sigma \big(\frac{\log(p_g(x)) - \log(p_{in}(x))}{T}\big)\big), \tag{94}$$

where $\sigma$ is sigmoid function, $T$ is the temperature parameter, $p_g$ is the general-distribution-network likelihood, and $p_{in}$ is the specific in-distribution network likelihood.

### 5.18 Self-Supervised Learning for Generalizable Out-of-Distribution Detection (107):

In this work, a self-supervised learning method is employed to use the information of an unlabeled outlier dataset such that the OOD detection utility of an in-distribution classifier is improved. To do so, at first, the classifier is trained on in-class training samples until the desired performance is achieved. Then, additional outputs (a set of $k$ reject classes) are added to the last layer. Each training batch consists of ID data and some outlier samples. The following loss function is used:

$$\min(E_{P_{\text{in}(\hat{x},\hat{y})}}[-\log(P_\theta(y = \hat{y} \mid \hat{x}))]) \\ + \lambda E_{P_{\text{out}(x,\text{target})}}[-\log(P_\theta(y = \text{target} \mid x))], \tag{95}$$

where $\lambda$ is a learning coefficient for OOD features, and the target is selected based on the following rules:

$$
\begin{aligned}
&\text{if} \quad \arg\max(P_\theta(x)) \in k \quad \text{then} \\
&\qquad \text{target} \leftarrow \arg\max(P_\theta(x)) \\
&\text{else} \\
&\qquad \text{target} \leftarrow \text{random}(k)
\end{aligned}
\tag{96}
$$

This is similar to unsupervised deep k-means in (20). If an unlabeled sample in the outlier dataset resembles in-class samples, it is assigned a random reject class each time; otherwise, it is assigned to a specific reject class. This helps unlabeled inliers to be separated from outliers. At the test time, the sum of the softmax output of the OOD classes is used as the detection score.

### 5.19 SSD: A Unified Framework for Self-Supervised Outlier Detection (150):

This work has a very similar idea to GDFR (*c.f.* Section 4.8). It incorporates SSL methods so to alleviate the need for labeling in-class samples. This is different from several aforementioned methods that require solving a classification task. As a result, SSD can be flexibly used in different settings such as ND, OSR, and OOD detection. The main idea is to employ the contrastive learning as in (25), and learn semantically meaningful features. After representation learning, a k-means clustering is applied to estimate class centers with mean and covariance $(\mu_m, \Sigma_m)$. Then for each test time sample, the Mahalanobis distance to the closest class centroid is used as the OOD detection score:

$$
s_x = \min_m (z_x - \mu_m)^T \Sigma_m^{-1} (z_x - \mu_m)
\tag{97}
$$

where $z_x$ represents the feature for input $x$. The contrastive learning objective function is simple. Using image transformations, it first creates two views of each image, commonly referred to as positives. Next, it optimizes to pull each instance close to its positive instances while pushing away from other images, commonly referred to as negatives:

$$
L_{\text{batch}} = \frac{1}{2N} \sum_{i=1}^{2N} -\log \frac{e^{u_i^T u_j / \tau}}{\sum_{k=1}^{2N} I(k \neq i) e^{u_i^T u_j / \tau}},
\tag{98}
$$

where $u_i = \frac{h(f(x_i))}{||h(f(x_i))||_2}$ is the normalized feature vector, $(x_i, x_j)$ are positive pairs for the $i$th image from a batch of $N$ images, and $h(\cdot)$ is the projection head. Moreover, when a few OOD samples are available, the following scoring function can be used:

$$
\begin{aligned}
s_x = {} & (z_x - \mu_{\text{in}})^T \Sigma_{\text{in}}^{-1} (z_x - \mu_{\text{in}}) \\
& - (z_x - \mu_{\text{ood}})^T \Sigma_{\text{ood}}^{-1} (z_x - \mu_{\text{ood}})
\end{aligned}
\tag{99}
$$

This framework can be extended to the supervised settings when the labels of in-class distribution are available. SSD employs the supervised contrastive learning objective proposed in (79):

$$
L_{\text{batch}} = \frac{1}{2N} \sum_{i=1}^{2N} -\log \frac{\frac{1}{2N_{y_i} - 1} \sum_{k=1}^{2N} I(y_k = y_i) e^{u_i^T u_j / \tau}}{\sum_{k=1}^{2N} I(k \neq i) e^{u_i^T u_j / \tau}},
\tag{100}
$$

where $N_{y_i}$ is the number of images with label $y_i$ in the batch.

### 5.20 MOS: Towards Scaling Out-of-distribution Detection for Large Semantic Space (69):

MOS first revealed that the performance of OOD detection can degrade significantly when the number of in-distribution classes increases. For example, analysis reveals that a common baseline MSP's average false positive rate (at 95% true positive rate) would rise from 17.34% to 76.94% as the number of classes increases from 50 to 1,000 on ImageNet1k. To overcome the challenge, the key idea of MOS is to decompose the large

semantic space into smaller groups with similar concepts, which allows simplifying the decision boundaries between known vs. unknown data. Specifically, MOS divides the total number of $C$ categories into $K$ groups, $G_1, G_2, ..., G_K$. Grouping is done based on the taxonomy of the label space if it is known, applying k-means using the features extracted from the last layer of a pre-trained network or random grouping. Then the standard groupwise softmax for each group $G_k$ is defined as follows:

$$p_c^k(x) = \frac{e^{f_c^k(x)}}{\sum_{c' \in G_k} e^{f_{c'}^k(x)}}, c \in G_k \tag{101}$$

where $f_c^k(x)$ and $p_c^k(x)$ denote output logits and the softmax probability for class $c$ in group $G_k$, respectively. Fig. 37 shows the overall architecture. The training loss function is defined as below:

$$L_{\text{GS}} = -\frac{1}{N} \sum_{n=1}^{N} \sum_{k=1}^{K} \sum_{c \in G_k} y_c^k \log(p_c^k(x)) \tag{102}$$

where $y_k^c$ and $p_k^c$ represent the label and the softmax probability of category $c$ in $G_k$, and $N$ is the total number of training samples.

A key component in MOS is to utilize a category `others` in each group, which measures the probabilistic score for an image to be unknown with respect to the group. The proposed OOD scoring function, Minimum Others Score (MOS), exploits the information carried by the `others` category. MOS is higher for OOD inputs as they will be mapped to `others` with high confidence in all groups, and is lower for in-distribution inputs. Finally, test time detection is done based on the following score:

$$S_{\text{MOS}}(x) = -\min_{1 \le k \le K} (p_{\text{others}}^k(x)) \tag{103}$$

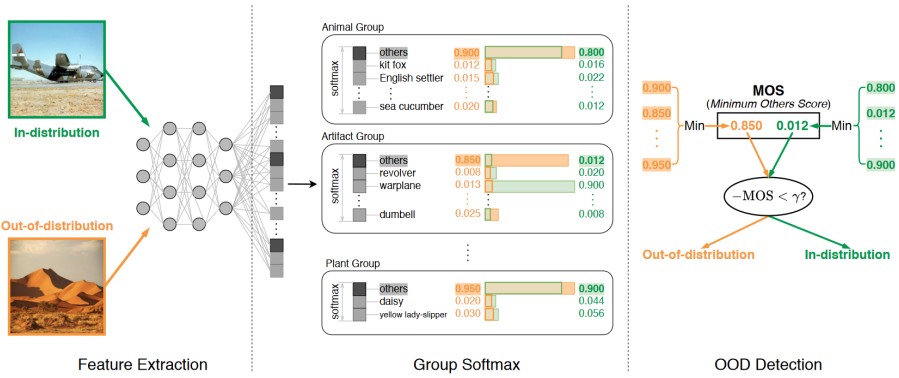

Figure 37: The overall architecture of MOS. As the figure shows, each sample is labeled as "others" for groups to which it does not belong, except the correct group. At the test time, detection is done based on the minimum of "others" class scores. The figure is taken from (69).

### 5.21 Can Multi-Label Classification Networks Know What They Don't Know? (171):

In this work, the ability of OOD detectors in the multi-label classification setting was investigated. In the multi-label classification setting, each input sample might have one or more corresponding labels, which make the problem harder since the joint distribution across labels can be intractable to model. This work proposes the *JointEnergy* criterion as a simple and effective method, which estimates the OOD indicator scores by aggregating label-wise energy scores from multiple labels. Also, they show that JointEnergy can be mathematically interpreted from a joint likelihood perspective. Similar to what has been discussed in

(95) $P(y_i = 1 \mid x)$ can be written as $\frac{e^{-E(x,y_i)}}{e^{-E(x)}}$, then by defining *label-wise free energy* that is a special case of $K$-class free energy with $K = 2$:

$$E_{y_i}(x) = -\log(1 + e^{f_{y_i}(x)}) \tag{104}$$

JointEnergy can be defined as:

$$E_{\text{Joint}}(x) = \sum_{i=1}^{K} -E_{y_i}(x) \tag{105}$$

Through derivations, the JointEnergy can be decomposed into three terms:

$$\begin{aligned} E_{\text{joint}}(x) &= \log p(x \mid y_1 = 1, \dots, y_K = 1) \\ &+ (K-1) \cdot \log p(x) + Z \end{aligned} \tag{106}$$

where the first term takes into account joint likehood across labels; and the second term reflects the underlying data density, which is supposed to be higher for in-distribution data $x$. The summation overall results in stronger separability between ID and OOD. The architecture overview is provided in Fig. 38.

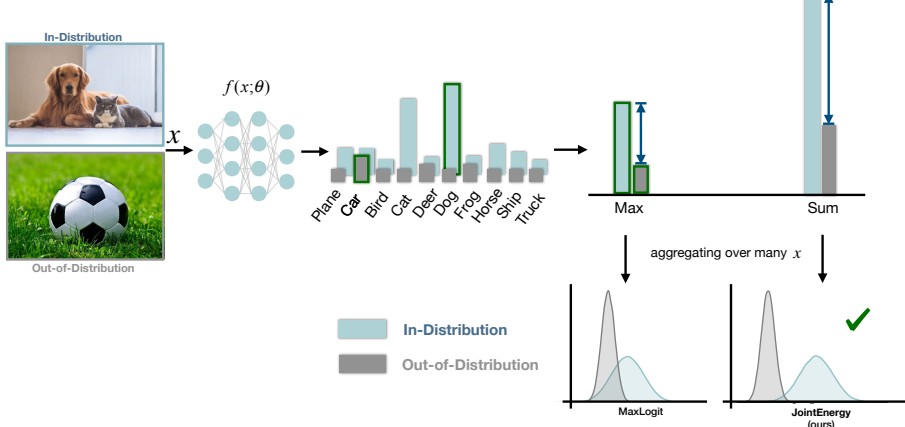

Figure 38: An overview of JointEnergy for OOD detection in multi-label classification networks. During inference time, input $x$ is passed through a classifier, and label-wise scores are computed for each label. OOD indicator scores are either the maximum-valued score (denoted by green outlines) or the sum of all scores. Taking the sum results in a larger difference in scores and more separation between in-distribution and OOD inputs (denoted by red lines), resulting in better OOD detection. Plots in the bottom right depict the probability densities of MaxLogit versus JointEnergy. The figure is taken from (171).

### 5.22 On the Importance of Gradients for Detecting Distributional Shifts in the Wild (70) :

This work proposes a simple post hoc OOD detction method GradNorm, which utilizes the vector norm of gradients with respect to weights, backpropagated from the KL divergence between the softmax output and a uniform probability distribution. The GradNorm is generally higher for in distribution (ID) data than that for OOD data. Therefore, it can be used for OOD detection. Specifically, the KL divergence is defined as follows:

$$D_{\text{KL}}(\mathbf{u} \| \text{Softmax}(f(x))) = -\frac{1}{C} \sum_{c=1}^{C} \log \left( \frac{e^{f_c(x)/T}}{\sum_{j=1}^{C} e^{f_j(x)/T}} \right), \tag{107}$$

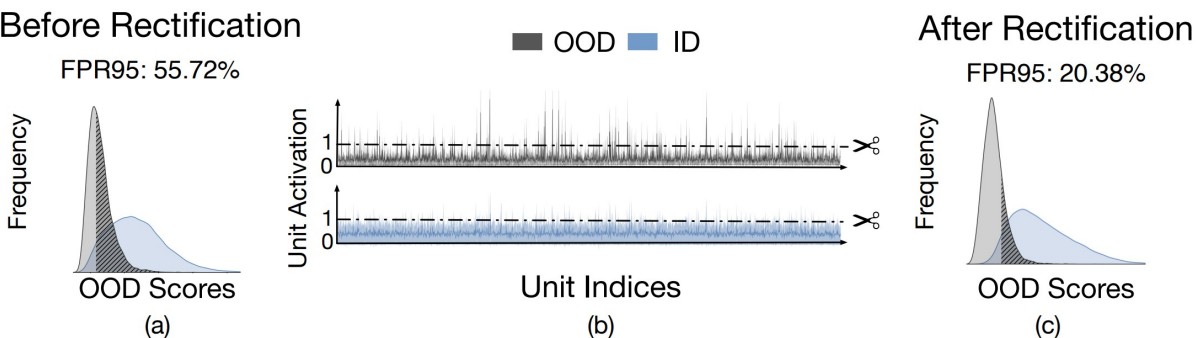

Figure 39: a) The distribution of ID and OOD uncertainty scores before truncation when ImageNet and iNaturalist are used as in-class and out-of-distribution samples, b) the distribution of per-unit activations in the penultimate layer for ID and OOD data, and c) the distribution of OOD scores after rectification for the ID and OOD data are depicted in the plots above. When ReAct is used, it significantly enhances the separation of the ID and OOD information. The figure is taken from (163).

where $\mathbf{u}$ is the uniform distribution, $T$ is the temperature, $f_c(x)$ denotes the $c$-th element of $f(x)$ corresponding to the label $c$, and $C$ is the number of in-distribution classes. Then, the OOD score is defined as $S(x) = ||\frac{\partial D_{\mathrm{KL}}}{\partial W}||_p$ where $W$ can be (1) a block of parameters, (2) all parameters, and (3) last layer parameters. It was mentioned that the third approach is better than others and achieves significant results.

### 5.23   ReAct: Out-of-distribution Detection With Rectified Activations (163):

The internal activations of neural networks are examined in depth in this study, which as discovered have considerably different signature patterns for OOD distributions and thus enable OOD detection. The distribution of activations in the penultimate layer of ResNet-50, which is trained on the ImageNet, is depicted in Fig. 39. Each point on the horizontal axis represents a single unit of measurement. The mean and standard deviation are depicted by the solid line and shaded area, respectively. The mean activation for the ID data (blue) is well-behaved, with a mean and standard deviation that are both nearly constant. Contrary to this, when looking at the activation for OOD data (gray), the mean activations display substantially greater variations across units and are biased towards having sharp positive values. The unintended consequence of having such a high unit activation is that it can present itself in the output of the model, resulting in overconfident predictions on the OOD data. Other OOD datasets exhibit a similar distributional property, which is consistent with previous findings. Therefore, a technique called Rectified Activations (also known as ReAct) is presented as a solution to this problem, in which the outsized activation of a few selected hidden units can be attenuated by rectifying the activations at an upper limit $c > 0$. It is demonstrated that ReAct can generalize effectively to a variety of network designs and that it is compatible with a variety of output-based OOD detection methods, including MSP (56), ODIN (92), and energy score (95), among others.

### 5.24   VOS: LEARNING WHAT YOU DON'T KNOW BY VIRTUAL OUTLIER SYNTHESIS (38):

In this paper, the key idea is to synthesize virtual outliers to enhance the model's decision boundary. The work differs from prior works (106; 189; 127) in two aspects. Firstly, it was one of the first works addressing the object-level out-of-distribution, where every input instance is an object instead of an entire image. Object-level OOD detection is useful when a complex scene contains objects of both ID and OOD categories. Secondly, instead of generating samples in the image space, this work proposes generating virtual outliers in the feature space, which is shown to be more tractable than synthesizing images in the high-dimensional pixel space.

The method overview is shown in Figure 40. To generate outlier features, VOS first models the feature space of the in-distribution classifier as a class-conditional multivariate Gaussian distribution:

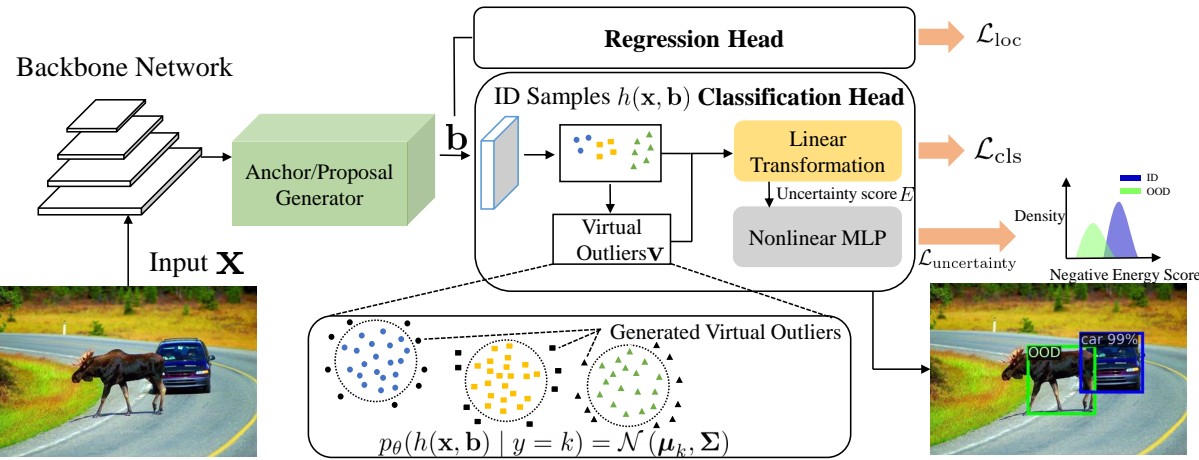

Figure 40: Overview of virtual outlier synthesis (VOS) for model regularization and OOD detection. The figure is taken from (38).

$$p_\theta(h(x, \mathbf{b}|y = k)) = N(\boldsymbol{\mu}_k, \Sigma) \tag{108}$$

where $\boldsymbol{\mu}_k$ is the mean of class $k \in \{1, ..., K\}$, $\Sigma$ is the tied covariance matrix, and $b \in R^4$ be the bounding box coordinates associated with object instances in the image, and $h(\mathbf{x}, \mathbf{b})$ is the latent representation of an object instance $(\mathbf{x}, \mathbf{b})$. The covariance $(\widehat{\boldsymbol{\Sigma}})$ and mean $(\widehat{\boldsymbol{\mu}}_k)$ are estimated empirically based on passing the training samples $(\{(\mathbf{x}_i, \mathbf{b}_i, y_i)\}_{i=1}^N)$ to the trained network. To be specific:

$$\begin{aligned}
\widehat{\boldsymbol{\mu}}_k &= \frac{1}{N_k} \sum_{i:y_i=k} h(\mathbf{x}_i, \mathbf{b}_i) \\
\widehat{\boldsymbol{\Sigma}} &= \frac{1}{N} \sum_k \sum_{i:y_i=k} (h(\mathbf{x}_i, \mathbf{b}_i) - \widehat{\boldsymbol{\mu}}_k)(h(\mathbf{x}_i, \mathbf{b}_i) - \widehat{\boldsymbol{\mu}}_k)^\top.
\end{aligned} \tag{109}$$

Then, virtual outliers are synthesized based on the estimated class-conditional distribution, with likelihoods smaller than a threshold $\epsilon$:

$$\mathcal{V}_k = \left\{ \mathbf{v}_k \mid \frac{1}{(2\pi)^{m/2}|\widehat{\boldsymbol{\Sigma}}|^{1/2}} \exp\left(-\frac{1}{2}(\mathbf{v}_k - \widehat{\boldsymbol{\mu}}_k)^\top \widehat{\boldsymbol{\Sigma}}^{-1}(\mathbf{v}_k - \widehat{\boldsymbol{\mu}}_k)\right) < \epsilon \right\}, \tag{110}$$

where $\mathbf{v}_k \sim \mathcal{N}\left(\widehat{\boldsymbol{\mu}}_k, \widehat{\boldsymbol{\Sigma}}\right)$ denotes the sampled virtual outliers for class $k$.

Having virtual outliers generated, the method regularizes the decision boundary via the energy score (95), where ID objects have negative energy values and the synthesized outliers have positive energy:

$$\mathcal{L}_{\text{uncertainty}} = E_{\mathbf{v}\sim\mathcal{V}}\left[-\log \frac{1}{1 + e^{-\phi(E(\mathbf{v};\theta))}}\right] + E_{\mathbf{x}\sim\mathcal{D}}\left[-\log \frac{e^{-\phi(E(\mathbf{x};\theta))}}{1 + e^{-\phi(E(\mathbf{x};\theta))}}\right], \tag{111}$$

where $\phi(\cdot)$ is a nonlinear MLP function. OOD detection can be done by replacing image-level energy with object-level energy. For ID object $(\mathbf{x}, \mathbf{b})$, the energy is defined as:

$$E(\mathbf{x}, \mathbf{b}; \theta) = -\log \sum_{k=1}^K w_k \cdot \exp^{f_k((\mathbf{x},\mathbf{b});\theta)}, \tag{112}$$

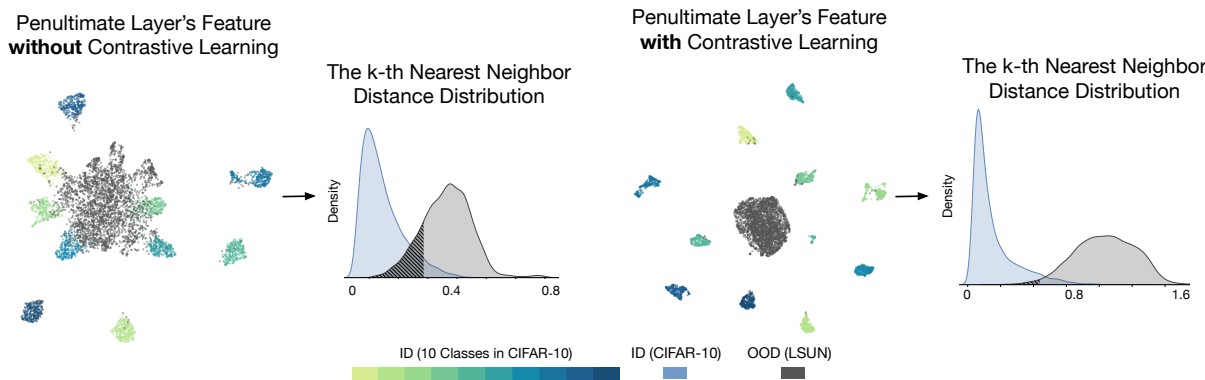

Figure 41: Overview of the deep KNN approach for leveraging embedding space to detect OOD samples. The figure is taken from (164).

where $f_k((\mathbf{x}, \mathbf{b}); \theta) = W_{\text{cls}}^{\top} h(\mathbf{x}, \mathbf{b})$ is the logit output for class $k$ in the classification branch and $W_{\text{cls}} \in \mathbb{R}^{m \times K}$ is the weight of the last fully connected layer. A training objective for object detection combines a standard loss and an uncertainty regularization loss:

$$\min_{\theta} \mathbb{E}_{(\mathbf{x}, \mathbf{b}, y) \sim \mathcal{D}} \left[ \mathcal{L}_{\text{cls}} + \mathcal{L}_{\text{loc}} \right] + \beta \cdot \mathcal{L}_{\text{uncertainty}} \tag{113}$$

where $\beta$ is the weight of the uncertainty regularization. $\mathcal{L}_{\text{cls}}$ and $\mathcal{L}_{\text{loc}}$ are losses for classification and bounding box regression, respectively. OOD detection is performed by using the logistic regression uncertainty branch during testing. Specifically, given an input $\mathbf{x}^*$, the object detector produces a bounding box prediction $\mathbf{b}^*$. The OOD uncertainty score for the predicted object $(\mathbf{x}^*, \mathbf{b}^*)$ is given by:

$$p_{\theta}\left(g \mid \mathbf{x}^*, \mathbf{b}^*\right) = \frac{\exp^{-\phi(E(\mathbf{x}^*, \mathbf{b}^*))}}{1 + \exp^{-\phi(E(\mathbf{x}^*, \mathbf{b}^*))}}. \tag{114}$$

A thresholding mechanism can be used to differentiate between ID and OOD objects for OOD detection:

$$G\left(\mathbf{x}^*, \mathbf{b}^*\right) = \begin{cases} 1 & \text{if } p_0\left(g \mid \mathbf{x}^*, \mathbf{b}^*\right) \geq \gamma \\ 0 & \text{if } p_{\theta}\left(g \mid \mathbf{x}^*, \mathbf{b}^*\right) < \gamma \end{cases} \tag{115}$$

The threshold $\gamma$ is typically chosen so that a high fraction of ID data (e.g., 95%) is correctly classified.

### 5.25 Out-of-distribution Detection with Deep Nearest Neighbors (164):

A non-parametric nearest-neighbor distance is explored for detecting OODs in this study. The distance-based methods assume that the test OOD samples are relatively far away from the ID data, and rely on feature embeddings derived from the model. Prior parametric distance methods (e.g. maximum Mahalanobis distance) impose distributional assumptions about the underlying feature space, and this study suggests that these assumptions may not always hold. The method uses a threshold-based criterion to determine whether the input is OOD by computing the $k$-th nearest neighbor (KNN) distance between the embeddings of the test input and the embeddings of the training set.

Specifically, OOD detection is performed by using the normalized penultimate feature $\mathbf{z} = \phi(\mathbf{x})/\|\phi(\mathbf{x})\|_2$, where $\phi : \mathcal{X} \mapsto \mathbb{R}^m$ is a feature encoder. Denote the embedding set of training data as $\mathbb{Z}_n = (\mathbf{z}_1, \mathbf{z}_2, \dots, \mathbf{z}_n)$. In testing, the normalized feature vector $\mathbf{z}^*$ is obtained for a test input $\mathbf{x}^*$, and calculate the Euclidean distances $\|\mathbf{z}_i - \mathbf{z}^*\|_2$ with respect to embedding vectors $\mathbf{z}_i \in \mathbb{Z}_n$. $\mathbb{Z}_n$ is reordered according to the increasing distance $\|z_i - z^*\|_2$. The reordered data sequence is represented by $\mathbb{Z}'_n = (\mathbf{z}_{(1)}, \mathbf{z}_{(2)}, \dots, \mathbf{z}_{(n)})$. OOD detection is based on the following decision function:

$$G\left(\mathbf{z}^*; k\right) = \mathbf{1}\left\{-r_k\left(\mathbf{z}^*\right) \geq \lambda\right\},$$

---
**Algorithm 1** OOD Detection with Deep Nearest Neighbors

---
**Input:** Training dataset $\mathbb{D}_{in}$, pre-trained neural network encoder $\phi$, test sample $\mathbf{x}^*$, threshold $\lambda$
For $\mathbf{x}_i$ in the training data $\mathbb{D}_{in}$, collect feature vectors $\mathbb{Z}_n = (\mathbf{z}_1, \mathbf{z}_2, ..., \mathbf{z}_n)$
**Testing Stage:**
Given a test sample, we calculate feature vector $\mathbf{z}^* = \phi(\mathbf{x}^*)/\|\phi(\mathbf{x}^*)\|_2$
Reorder $\mathbb{Z}_n$ according to the increasing value of $\|\mathbf{z}_i - \mathbf{z}^*\|_2$ as $\mathbb{Z}'_n = (\mathbf{z}_{(1)}, \mathbf{z}_{(2)}, ..., \mathbf{z}_{(n)})$
**Output:** OOD detection decision $1\{-\|\mathbf{z}^* - \mathbf{z}_{(k)}\|_2 \geq \lambda\}$

---

Figure 42: Algorithm 1 describes the KNN-based OOD detection in detail. The figure is taken from (164).

where $r_k(\mathbf{z}^*) = \left\|\mathbf{z}^* - \mathbf{z}_{(k)}\right\|_2$ is the distance to the $k$-th nearest neighbor, and $\mathbf{1}\{\cdot\}$ is the indicator function.

As the distance threshold is only estimated based on ID data, the method is OOD-agnostic since unknown data is not required in the testing procedure. The method is also model-agnostic since it applies to a variety of loss functions (e.g., cross-entropy and supervised contrastive loss), and model architectures (e.g., CNN and ViT). Moreover, the KNN method is easy to use thanks to modern implementations, such as Faiss (75), a library that allows running this in milliseconds, even when the database contains billions of images. In contrast, some prior methods cause numerical instability, such as Mahalanobis distance, which requires the inverse computation of the covariance matrix.

# 6 A Summary of the Shared Core Ideas

As previously mentioned, methods and formulations of out-of-distribution detection, open-set recognition, anomaly detection, and novelty detection overlap extensively. Here, we present a selection of notions that have been proved to be advantageous across a variety of areas, albeit under different titles. All of these concepts have been discussed in prior sections, which can be referred to for more information.

**Making a compressed representation for normal samples:** The concept is founded mostly on the projection of normal samples into a feature space so that normal training samples are located in close proximity to one another. In this manner, the feature extractor is compelled to concentrate on the most shared features among all normal inputs, which implies that the normal and abnormal feature spaces will have less in common; as a result, abnormal inputs will be projected far from normal inputs and can be easily identified.

This intuition has been followed in anomaly detection by DSVDD (136) and DSAD (137), by finding the most compact hyper-sphere in the feature space that includes normal training samples. GOAD (12) takes a similar approach, determining the most compact hyper-sphere for each set of linear transformations applied to the normal inputs. In open-set recognition, (33) defines a so called "objectoshpere" loss that minimizes the $l_2$ norm of normal training samples, which is equal to compressing them in a hyper-sphere centered in (0, 0). CAC (104) follows the same strategy as GOAD, but rather than relying on transformations, it takes advantage of the labels that are made available during open-set recognition and makes use of a variety of normal samples in order to generate some distinct hyper-spheres.

**Analyzing normal gradient:** In this approach, instead of directly using normal features, their gradients are utilized. (172) demonstrated that self-supervised techniques such as GT (48) can perform effectively in the face of a small number of outliers or anomalous samples in the training dataset. This occurs because normal samples alter the gradients substantially more than abnormal samples; thus, the notwork can still mostly extract normal features. Similarly, in out-of-distribution detection, (70) shows that the vector norm of gradients with respect to weights, backpropagated from the KL divergence between the softmax output and a uniform probability distribution is greater for normal samples than abnormal inputs.

**Outlier exposure technique:** Outlier-exposure-based methods usually assume that there are some abnormal samples that can be easily used to improve performance. For example, (58) uses online training datasets that are free to use during the training process to make better normal boundaries, which improve the out-of-distribution detection method. Similarly, DSAD (137) uses datasets that are already on the internet to learn the distribution of abnormal training samples. The same idea is continued by (91; 33), which was explained in more detail earlier.

**Generating outliers:** This idea is mainly based on generating fake outliers by the premature training of generative models on the normal distribution. Then, the out of distribution model is trained with the combination of the fake and existing normal training samples. The approach is followed by DSAD (137), G2D (127), and Old-is-Gold (189) in anomaly detection domain. In the same way, methods like G-OpenMax (46), OpenGAN (35) take the advantages of this approach to improve the performance of open-set recognition and out-of-distribution detection.

**Prototype-based methods:** Here, the objective is to determine the least number of prototypes are needed to adequately cover the normal feature space. The normality score is then calculated by cofeatures of each input to the prototypes learned during training. In anomaly detection and open-set recognition, Mem-AE (49) and RPL (22) implement this concept, respectively.

**Generative models do not make abnormal inputs.** It is assumed that generative models trained on normal distributions struggle to create or reconstruct abnormal inputs as well as normal inputs. As a result, measurements such as reconstruction error in AE-based approaches or discriminator output in GAN-based structures can be utilized to differentiate normal from abnormal. In anomaly detection, methods such as (139; 189; 1; 122; 159) use this approach. In open-set recognition and out-of-distribution detection, C2AE (117), CROSR (183) and CGDL (162) follow the same methodology.

**Leveraging the knowledge of pre-trained models.** Instead of using the raw training datasets available on the internet, why not utilize the models that have been trained on these datasets? Pre-trained features intuitively contain information that can be conveyed to downstream tasks more effectively than the original training samples. Several approaches have been developed in anomaly detection and open-set recognition based on this notion, such as MKD (144), which distills the knowledge of a pre-trained model for normal training samples into another network to improve anomaly detection performance. Similarly, in open-set recognition, DTL (121) employs the filters of a pre-trained model that are more stimulated when the model is presented with normal inputs as an additional detection criterion to filter open-set samples during the evaluation.

**Self-supervised learning-based methods:** It has been shown by (167) that learning better representations has a direct correlation with the performance of open-set recognition. Therefore, self-supervised learning methods as a well-known unsupervised representation learning technique seems to be a good fit for anomaly detection, novelty detection, open-set recognition, and out-of-distribution detection. Following this approach, several methods such as CSI (165), DA-Contrastive (158), CutPaste (90) have been proposed, which show a strong performance in detecting semantic and pixel-level anomaly detection. In out-of-distribution detection, (60) investigates the effect of self-supervised learning on detecting outliers extensively. A similar approach was also followed by (150; 107), which was explained in detail in previous sections.

## 7 Dataset

### 7.1 Semantic-Level Datasets

Below we summarize datasets that can be used to detect semantic anomalies. Semantic anomalies are those kinds of anomalies that the variation of pixels leads to the change of semantic content. Datasets such as MNIST, Fashion-MNIST, SVHN, and COIL-100 are considered as toy datasets. CIFAR-10, CIFAR-100, LSUN, and TinyImageNet are hard datasets with more variations in color, illumination, and background.

Finally, Flowers and Birds are fine-grained semantic datasets, which makes the problem even harder.

**MNIST (85):** This dataset includes $28 \times 28$ grayscale handwritten digits from 0-9 and consists of 60k training images and 10k testing ones.

**Fashion MNIST (177):** This dataset comprises of $28 \times 28$ grayscale images of 70k clothing items from 10 categories. The training set has 60k images and the test set has 10k images.

**CIFAR-10 (83):** The CIFAR-10 has 60k natural images. It consists of 32x32 RGB images of 10 classes, There are 50k training images and 10k test images.

**CIFAR-100 (83):** This dataset is very similar to CIFAR-10, but it has 100 classes containing 600 images each.The 100 classes are grouped into 20 super classes. Each class has 500 training images and 100 testing images.

**TinyImageNet (31):** The Tiny ImageNet dataset consists of a subset of ImageNet images . It contains 10,000 test images from 200 different classes. Also, two more datasets, TinyImageNet (crop) and TinyImageNet (resize) can be constructed, by either randomly cropping image patches of size $32 \times 32$ or downsampling each image to size $32 \times 32$.

**LSUN (184):** The Large-scale Scene UNderstanding dataset (LSUN) has a testing set of 10,000 images of 10 different scene categories such as bedroom, kitchen room, living room, etc. Similar to TinyImageNet, two more datasets, LSUN (crop) and LSUN (resize), can be reconstructed by randomly cropping and downsampling the LSUN testing set, respectively.

**COIL-100 (112):** COIL-100 is a dataset of colorful images of 100 objects. It comprises 7200 $128 \times 128$ images. Images are captured from objects placed on a motorized turntable against a black background, and there are 72 images of each object in different poses.

**SVHN (113):** SVHN can be seen as similar in flavor to MNIST (e.g., the images are of small cropped digits), but incorporates an order of magnitude more labeled data (over 600,000 digit images) and comes from a significantly harder, unsolved, real-world problems (recognizing digits and numbers in natural scene images). SVHN is obtained from house numbers in Google Street View images.

**Flowers (115):** Flowers is a 102 category dataset, consisting of 102 flower categories. The flowers chosen to be flowers commonly occurring in the United Kingdom. Each class consists of between 40 and 258 images. The images have large scale, pose and light variations. In addition, there are categories that have large variations within the category and several very similar categories.

**Birds (175):** CUB-200-2011 is a bird classification task with 11,788 images across 200 wild bird species. There is roughly equal amounts of train and test data. It is generally considered one of the most challenging datasets since each species has only 30 images for training.

## 7.2 Pixel-Level Datasets

In these datasets unseen samples, outliers or anomalies do not have semantic difference with inliers. This means an area of the original image is defected; however, the original meaning is still reachable, yet has been harmed.

**MVTec AD ((13)):**This dataset is an industrial dataset, it provides 5354 high-resolution images divided into ten objects and five texture categories and contains 3629 training images. The test set contains 467 normal images and 1258 abnormal images having various kinds of defects.

**PCB ((71)):** PCB dataset containing 1386 images with 6 kinds of defects for the use of detection, classification and registration tasks. Images are captured in high-resolution.

**LaceAD ((179)):** LaceAD contains 9,176 images from the top 10 lace fabric manufacturing companies worldwide, where the images are captured in the real production environment by a high-resolution DSLR

camera, They are categorized into 17 subsets based on their patterns. Each image has the size of $512 \times 512$ and has been labeled by professional workers.

**Retinal-OCT ((77)):** This consists of 84,495 X-Ray images in 4 categories CNV, DME, DRUSEN, and NORMAL, each of which has subtle differences with respect to others.

**CAMELYON16 ((9)):** Detecting metastases of lymph nodes is an extremely important variable in the diagnosis of breast cancers. Tissue with metastasis may differ from a healthy one only in texture, spatial structure, or distribution of nuclei, and can be easily confused with normal tissue. The training dataset of Camelyon16 consists of 110 whole-slide images (WSIs) of tumors, and 160 non-tumor cases, and testing dataset with 80 regular slides and 50 slides containing metastases.

**Chest X-Rays ((173; 73; 129)):** Chest X-Ray datasets are medical imaging datasets which comprise a large number of frontal-view X-ray images of many unique patients (collected from the year of 1992 to 2015). The datasets include eight to fourteen common disease labels, mined from the text radiological reports via NLP techniques. Images are not registered and captured in different pose and contrast, which makes the detection task challenging.

**Species ((62)):** This dataset consists of organisms that fall outside ImageNet-21K. Consequently, ImageNet-21K models can treat these images as anomalous.

**ImageNet-O ((65)):** ImageNet-O is a dataset of adversarially filtered examples for ImageNet out-of-distribution detectors. To create this dataset, at first, ImageNet-22K is downloaded, and shared examples from ImageNet-1K are deleted. With the remaining ImageNet-22K examples that do not belong to ImageNet-1K classes, examples that are classified by a ResNet-50 as an ImageNet-1K class with a high confidence are kept. Finally, visually clear images are selected. This creates a dataset of OOD examples that are hard for a ResNet-50. These examples are challenging for other models to detect, including Vision Transformers.

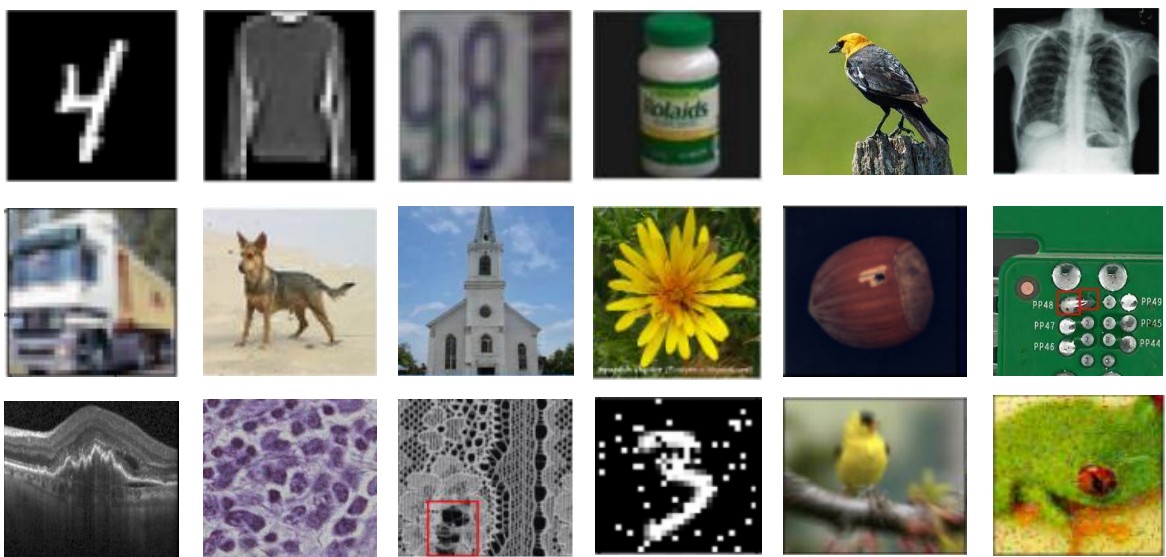

Figure 43: Sample visualization of MNIST, Fashion-MNIST, SVHN, COIL-100, Birds, Chest X-Rays, Cifar-10, TinyImageNet, LSUN, Flowers, MVTecAD, PCB, Retinal-OCT, CAMELYON16, LaceAD, MNIST-C, ImageNet-C, and ImageNet-P.

## 7.3 Synthetic Datasets

These datasets are usually made using semantic-level datasets; however, the amount of pixel-variations is under control such that unseen, novel, or abnormal samples are designed to test different aspects of trained

models while preserving semantic information. For instance, MNIST-c includes MNIST samples augmented with different kinds of noises such as shot noise and impulse noise, which are random corruptions that may occur during the imaging process. These datasets could be used to not only test the robustness of the proposed models, but also for training models in the AD setting instead of novelty detection or open-set recognition. Due to the lack of comprehensive research in the field of anomaly detection, these datasets can be very beneficial.

**MNIST-C (109):** MNIST-C dataset is a comprehensive suite of 15 corruptions applied to the MNIST test set for benchmarking out-of-distribution robustness in computer vision. Through several experiments and visualizations, it is shown that the corruptions significantly degrade the performance of state-of-the-art computer vision models while preserving the semantic content of the test images.

**ImageNet-C, ImageNet-P (55):** This can be seen as the ImageNet version of MNIST-C. For the ImageNet-C, a set of 75 common visual corruptions are applied on each image, and for the ImageNet-P, a set of perturbed or subtly differing ImageNet images are introduced. It is shown that although these perturbations are not chosen by an adversary, currently existing networks exhibit surprising instability on common perturbations.

An overall visualization of the mentioned datasets can be found in Fig. 43.

# 8 Evaluation Protocols

## 8.1 AUC-ROC

Receiver Operating Characteristics (ROC) is a well-known criterion. Given a test dataset including positive and negative (or seen and unseen ) samples, it characterizes the relation between the false positive rate (FPR) and the true positive rate (TPR) at different detection thresholds. AUC-ROC is the area under the ROC curve, which is a threshold-independent metric. The highest value of ROC is 1, and 0.5 indicates that the model assigns the positive label with random guessing.

In the literature, AD and ND are usually tested in one-vs-all setting that considers one class as normal and the rest of the classes as anomaly, unseen or unknown. For OOD detection, the in-distribution data is considered as positive and OOD data is considered as negative. For instance, one can train a model on CIFAR-10 and consider MNIST as outlier at the test time. This means training and testing datasets have a large contrast with each other. Sometimes instead of MNIST, uniform noise or Gaussian noise can be used.

## 8.2 FPR@TPR

Although AUC-ROC is a common metric, in practice, models have to select a specific threshold to perform detection. To address this issue, an operating point on the ROC which is desired with respect to applications is selected. A commonly-used metric is FPR@TPR$x$, which measures the FPR when the TPR is $x = 0.95$.

## 8.3 AUPR

AUPR is the Area under the Precision-Recall curve, which is another threshold independent metric. The PR curve depicts the precision$=\frac{TP}{TP+FP}$ and recall$=\frac{TP}{TP+FN}$ under different thresholds. In some literature, the metrics AUPR-In and AUPR-Out denote the area under the precision-recall curve where in-distribution and out-of-distribution images are specified as positives, respectively. This metric is usually used in OOD detection setting. FP is the number of false positives, TN is the number of true negatives, FN is the number of false negatives, and TP is the number of true positives.

## 8.4 Accuracy

This metric is usually used in OSR, which is a common choice for evaluating classifiers under the closed-set assumption, as follows:

$$A = \frac{\sum_{i=1}^{C}(TP_i + TN_i)}{\sum_{i=1}^{C}(TP_i + TN_i + FP_i + FN_i)} \tag{116}$$

This can be easily extended to open-world assumption in which UUCs must be classified correctly:

$$A_O = \frac{\sum_{i=1}^{C}(TP_i + TN_i) + TU}{\sum_{i=1}^{C}(TP_i + TN_i + FP_i + FN_i) + (TU + FU)}, \tag{117}$$

where $TU$ specifies the number of correctly predicted UUC samples.

Although accuracy is a common metric; however, it is very sensitive to imbalanced number of samples, which is not the case in metrics such as AUC-ROC. To cope with this issue, a normalized accuracy (NA), which weights the accuracy for KKCs (AKS) and the accuracy for UUCs (AUS) is defined as follows:

$$\text{NA} = \lambda_r \text{AKS} + (1 - \lambda_r)\text{AUS} \tag{118}$$

where

$$\text{AKS} = \frac{\sum_{i=1}^{C}(TP_i + TN_i)}{\sum_{i=1}^{C}(TP_i + TN_i + FP_i + FN_i)}$$
$$\text{AUS} = \frac{TU}{TU + FU} \tag{119}$$

and $0 \leq \lambda_r \leq 1$ is a regularization constant.

## 8.5 F-measure

The F-measure or F-score is the harmonic mean of precision $P$ and recall $R$:

$$F = 2 \times \frac{P \times R}{P + R} \tag{120}$$

Note that $F$ must be computed in the same way as the multi-class closed set scenario. This is because the correct classifications of UUCs would be considered as true positive classifications; however, it makes no sense since there is no UUC sample in the training process. Therefore, the computations of Precision and Recall only for KKCs are modified to give a relatively reasonable F-measure for OSR. The new measures are called macro-F-measure and micro-F-measure respectively as follows:

$$P_{\text{ma}} = \frac{1}{C}\sum_{i=1}^{C}\frac{TP_i}{TP_i + FP_i}, \quad R_{\text{ma}} = \frac{1}{C}\sum_{i=1}^{C}\frac{TP_i}{TP_i + FN_i}$$
$$P_{\text{mi}} = \frac{\sum_{i=1}^{C}TP_i}{\sum_{i=1}^{C}(TP_i + FP_i)}, \quad R_{\text{mi}} = \frac{\sum_{i=1}^{C}TP_i}{\sum_{i=1}^{C}(TP_i + FN_i)} \tag{121}$$

Note that although the precision and recall only consider the KKCs; however, by computing $FN_i$ and $FP_i$ false UUCs and false KKCs are taken into account (76).

### 8.6 Comparing Evaluating Procedures of Each Domain

Each domain assessment process establishes the practical restrictions that must be met before the claimed performance of proposed methods within that domain can be trusted. Furthermore, in some domains, such as OSR and OOD detection, highlighting the differences in evaluation protocols utilized in each domain might help to gain a better expectation about the reported performances when applied in practical applications. As a result, given an arbitrary dataset such as CIFAR-10 (83), the following training/testing setup is employed at each domain :

**AD/ND :** (1) Select a random class out of 10 given classes as normal distribution. (2) Train a one-class feature extractor according to the discussed approaches. (3) Test the trained feature extractor on the entire test-set, which implies the implicit assumption of 10% likelihood for the occurrence of normal events and 90% for the abnormal ones.

**OOD Detection :** (1) Use the whole training dataset to train a multi-class feature extractor. (2) As the out-of-distribution dataset, choose another dataset, such as one of the datasets listed above. (3) Consider the entire test set to be normal and select some random samples from the out-of-distribution dataset so that the chance of abnormal occurrences is lower than normal when all the samples are combined. As can be observed, this assessment methodology makes an implicit assumption about the rarity of anomalies, which OSR does not.

**OSR :** (1) As the normal distribution, pick $K$ random classes. (2) Train a multi-class feature extractor according to the discussed approaches. (3) Test the trained feature extractor on the entire test-set. As previously stated, $K$ is decided based on the **openness score**. This suggests that abnormal events may have a higher probability than normal events, because anomaly means not being a member of the normal sets. For instance, in the self-driving car application, a model might be only responsible for detecting people and cars as two small sets of many definable sets existing in the agent environment. This is a key distinction between the OOD detection and OSR viewpoints.

## 9 Core Challenges

Here we examine the core and ongoing challenges in the fields. This allows the community to employ existing knowledge and craft future solutions effectively.

### 9.1 Designing Appropriate Self-Supervised Learning Task

As aforementioned, some methods such as (48; 12; 55) use SSL tasks to identify outliers; however, it has been recently investigated (6; 182) that augmentation techniques used as SSL tasks are not always beneficial for ND, OSR, and, OOD.

(182) shows that self-supervision acts as a yet-another model hyper-parameter, and should be carefully chosen in light of the types of real abnormalities in the data. In other words, the alignment between the augmentation and underlying anomaly generating process is essential for the success of SSL-based anomaly detection approaches. SSL can even worsen detection performance in the absence of such alignment. Similarly, (6) shows that Maximum Softmax Probability (MSP), as the simplest baseline for OSR, applied on Vision Transformers (ViTs) trained with carefully selected augmentations can surprisingly outperform many recent methods. As a result, the community needs to research more on the SSL tasks particularly designed to fulfill the goal of outlier detection.

### 9.2 Data Augmentation

One source of uncertainty in classifying known or normal training samples could be the lack of generalization performance. For instance, if one rotates a bird image, its content is not harmed and must be distinguished as the bird again. Some of the mentioned works try to embed this ability into models by designing different SSL objective functions. However, there is also another way to do that, using data augmentations. Data

augmentation is a common technique to enrich the training dataset. Some approaches (32; 194; 188; 61; 143) improve generalization performance using different data augmentation techniques.

From another point of view, (127) attempts to generate unseen abnormal samples and use them to convert a one-class learning problem into a simple two-class classification task. Similarly, however in the OSR setting, (82) and (35) follow the same idea. All these studies can also be seen as working on a training dataset to make it richer for the further detection task. From what has been said, it is obvious that working on data instead of models could achieve very effective results and must be investigated more in the sense of different trade-offs in the future.

Recently, (106) has shown the effectiveness of generating fake samples by utilizing SDE-based models (161) instead of GANs. The key innovation in this study is a new strategy for creating synthetic training outliers using SDE to train OE-based ND algorithms. By optimizing previously trained SOTA models for the task of ND, the paper shows the effectiveness of the obtained samples. Following the conventional OE pipeline, it minimizes the binary cross entropy loss between the normal training data and artificial outliers.

### 9.3 Small Sample Size and Few-Shot Learning

Learning with a small sample size is always challenging, but desirable. One way of approaching the problem could be to exploit meta-learning algorithms (41; 114), and to learn generalizable features that can be easily adapted to AD, ND, OSR, or OOD detection using a few training samples (100). One challenge in meta-learning is handling distribution shift between training and adaptation phase, which might result in producing one-class meta-learning algorithms such as (43). Also, other approaches explored generating a synthetic OOD dataset to improve few-shot classification of in-class samples (74). Although, the combination of meta-learning and AD, ND, OOD detection, and OSR has gained significant attention recently, some important aspects—such as generalizing to detect UUCs using only a few KUC and the convergence of meta-learning algorithms in one-class setting—remain underexplored.

Recently, some efforts have been made to manipulate the knowledge of pre-trained vision-language models to tackle the problem of zero-shot and few-shot AD and OOD. For instance, (99) uses CLIP (128) and textual queries to define the normal class without utilizing any visual inputs. Similarly, yet in the OOD evaluation setting, (42) provides an extensive number of experiments to show the capabilities of vision transformers at detecting near-out-of-distribution samples. They also test zero-shot and multi-modal settings using vison-language based models.

## 10 Future Challenges

Here we provide plausible future directions, which might be of interest to both practitioners and academic researchers.

### 10.1 Evaluating Baselines and the Evaluation Protocols of OOD Detection

The evaluation protocols for OOD detection has room for improvement. For instance, (2) trains a mixture of three Gaussians on the CIFAR-10 dataset (as ID), and evaluated against OOD datasets including TinyImagenet (crop), TinyImagenet (resize), LSUN, LSUN(resize) and iSUN. The model is trained channel-wise at a pixel-level. Tab. 2 shows the detection results on different datasets. The results are comparable with SOTA despite the simplicity. In particular, LSUN performs poorly since the majority of them have uniform color and texture, with little variation and structure. Similar to what has been observed in likelihood-based methods, LSUN "sits inside" CIFAR-10 with a similar mean but lower variance, and ends up being more likely under the wider distribution.

We also provide a better insight into the performance of OOD detection baselines, evaluated on both near and far out-of-distribution datasets. For a model that trained on CIFAR-10, we have considered the CIFAR-100 as the near OOD dataset. The results are presented in Tables 3, 4, and 6. As it is shown, none of the methods are good at detecting both near and far OOD samples, except OE approaches that use an extra auxiliary dataset. Using Mahalanobis distance can improve the performance of most of the methods at

detecting far OOD samples, while degrading the performance of near OOD detection. Moreover, it can also have poor performance at detecting even some of far OOD samples due to inaccurate Gaussian density estimation. Furthermore, its performance varies significantly when OOD dataset is resized or cropped, showing its dependency on low-level statistics. For instance, notice the SVHN column of Table 6. This is in correspondence with what has been shown recently by (133) on the deficiencies of Mahalanobis distance as well.

One solution to resolve the issue might be applying input pre-processing techniques such as ODIN to alleviate the effect of first and second-order statistics in assigning OOD scores; however, it increases the execution speed by the sum of an extra forward and backward pass during testing. Additionally, techniques such as ensembling or MC-Dropout (44) might be slightly better than others on some OOD datasets; yet, they need multiple forward passes, increasing the execution time significantly. For example, the reported MC-Dropout is 40 times slower than a simple MSP. In summary, future works are encouraged to evaluate OOD detection on both near and far OOD datasets.

Table 2: The performance of a simple method using only low-level features on different datasets (2).

| OOD Dataset | Average Precision |
|---|---|
| TinyImageNet(crop) | 96.8 |
| TinyImageNet(resize) | 99.0 |
| LSUN | 58.0 |
| LSUN(resize) | 99.7 |
| iSUN | 99.2 |

## 10.2 AD Needs to Be Explored More

As mentioned earlier, AD and ND are not completely the same, both historically and fundamentally. An essential and practical category of problems in the real-world application are those that can not be cleaned easily, and consequently, contain different kinds of noises such as, **label noise** or **data noise**. This is the case in complex and hazardous systems such as modern nuclear power plants, military aircraft carriers, air traffic control, and other high-risk systems (64). Recently proposed methods in ND need to be evaluated in AD settings using the proposed synthetic datasets, and new solutions need to be proposed. As the openness score is usually high for AD detectors, having a high recall while providing a low false alarm rate is necessary for their practicality (29).

Furthermore, almost all the AD or ND methods are evaluated in the one-vs-all setting. This results in having a normal class with a few distribution modes, which is not a good approximation of real-world scenarios. Therefore, evaluating AD or ND methods in multi-class settings similar to OSR domain while having no access to labels could give a more clear perspective on the practicality of SOTA methods. Table 7 reports the performance of the most notable works using the standard evaluation protocols explained before. All the results are reported from the main papers.

## 10.3 OSR Methods for Pixel-Level Datasets

Almost all the methods existing in OSR are evaluated on semantic datasets. As class boundaries in such datasets are usually far from each other, discriminative or generative methods can model their differences effectively. However, in many applications such as Chest X-ray datasets, variations are subtle. Existing methods can result in poor performance on such tasks. For instance, a model may be trained on 14 known chest diseases. A new disease, for example, COVID-19, may emerge as unknowns. In this case, the proposed model must detect it as a new illness instead of classifying it into pre-existing disease categories. Also, in many clinical applications where medical datasets are gathered, usually, disease images are more accessible than healthy ones; thus, OSR problems must be learned on sickness as normal images and detect healthy ones as abnormal inputs.

Table 5 shows the performance of a simple MSP baseline on MVTecAD dataset when some frequent faults are considered as normal classes. In such scenarios, the goal is to detect and classify well-known faults

while distinguishing rare ones as outliers that need to be treated differently. Although this is a common and practical industrial setting, the baseline does not perform better than random, casting doubt on their generality for safety-critical applications.

Recently, (26) has shown the effectiveness of using a prior Gaussian distribution on the penultimate layer of classifier networks, similar to what has been done in several before-mentioned works, in tasks in which the distribution of seen classes are very similar to each other, for instance, Flowers or Birds datasets that were introduced in the previous sections. However, more research should be done in this setting since it is more practical and quite harder than the traditional ones. Table 8 reports the performance of the most notable works using the standard evaluation protocols explained before. All the results are reported from the main papers.

### 10.4  Adversarial Robustness

Carefully designed imperceptible perturbations fooling deep learning-based models to make incorrect predictions are called adversarial attacks (186). Up to this time, it has been shown that classifiers are susceptible to adversarial attacks, such that their performance degrades significantly at test time. As in OOD detection, OSR, AD and ND, being robust against adversarial attacks is crucial. Recent works in OSR (154; 135), ND (143; 142), and OOD detection (103; 24; 149; 23) have investigated the effects of adversarial attacks on models; however, more is needed. For instance, as anomalies in AD or UUCs in OSR are inaccessible at the training time, therefore, achieving a robust model on attacked anomalies or UUCs would not be trivial.

The relation of different defense approaches against adversarial attacks with novelty detection can also reveal some important insights about the internal mechanisms of the proposed models. For instance, membership attack (156) attempts to infer whether an input sample has been used during the training process or not, which can be seen as designing novelty detectors without having any generalization to UKC samples. Also, (37) investigates the relation of detecting poisoning attacks and novelty detectors. Poisoning examples that are intentionally added by attackers to achieve backdoor attacks could be treated as one type of "outliers" in the training dataset. It is claimed that differential privacy not only improves outlier detection and novelty detection, but also backdoor attack detection of ND models.

From an entirely different point of view, as it is mentioned in (72), adversarial robust training can be employed to boost learned feature space in a semantic way. This path has been followed in ARAE (143) and Puzzle-AE (142) to improve the performance of AEs in detecting unseen test time samples. Similar intention is followed in the one-class learning method (50) that shows robustness is beneficial for detecting novel samples. This path also needs to be explored more, for instance despite standard adversarial attacks in classification tasks (101), attacks do not need to be imperceptible anymore in AD or ND and sometimes perceptible ones improve detection performance more.

### 10.5  Fairness and Biases of Models

Research on fairness has witnessed a significant growth recently (190; 18; 39):. It has been shown that models become biased towards some sensitive variables during their training process. For instance, (174) show that for attribute classification task in the CelebA (97) dataset, attribute presence is correlated with the gender of people in the image, which is obviously not desirable. Attributes such as gender in the mentioned example are called protected variables. In OOD detection literature, a recent work (105) systematically investigates how spurious correlation in the training set impacts OOD detection. The results suggest that the OOD detection performance is severely worsened when the correlation between spurious features and labels is increased in the training set. For example, a model that exploits the spurious correlation between the water background and label `waterbird` for prediction. Consequently, a model that relies on spurious features can produce a high-confidence prediction for an OOD input with the same background (i.e., water) but a different semantic label (e.g., boat).

Fairness and AD or ND seem to have fundamental contrast with each other. In fairness, unbiased models are required in which equality constraints between minorities and majorities hold while the goal of AD models

Table 3: OOD example detection for the maximum softmax probability (MSP) baseline detector, maximum logit value, the MSP detector after fine-tuning with Outlier Exposure (OE), the maximum logit value after fine-tuning with Outlier Exposure (OE), ensemble of 3 models, Mahalanobis distance, and Monte Carlo dropout. Inlier distribution is considered as CIFAR-10. All results are based on our evaluations and are average percentages of 10 runs. Missing values (-) indicate that we are proposing this setting, and it has no specific reference.

| Method | References | Criterion | Gaussian | Rademacher | Blob | TinyImageNet(crop) | TinyImageNet(resize) | LSUN(crop) | LSUN(resize) | iSUN | SVHN | CIFAR-100 |
|---|---|---|---|---|---|---|---|---|---|---|---|---|
| MSP | (56) | FPR95 | 14.53 | 94.78 | 70.50 | 17.06 | 40.10 | 12.65 | 29.23 | 36.22 | 28.37 | 43.27 |
| | | AUROC | 94.78 | 79.85 | 94.63 | 94.64 | 88.30 | 96.45 | 91.40 | 90.00 | 91.94 | 87.77 |
| | | AUPR | 70.50 | 32.21 | 74.23 | 75.09 | 58.15 | 83.16 | 65.36 | 62.46 | 67.10 | 55.68 |
| MLV | - | FPR95 | 52.60 | 73.27 | 11.67 | 9.59 | 47.67 | 4.93 | 27.28 | 36.42 | 43.54 | 56.52 |
| | | AUROC | 75.48 | 70.08 | 96.85 | 97.84 | 89.16 | 98.93 | 94.05 | 93.38 | 91.11 | 87.13 |
| | | AUPR | 27.07 | 25.47 | 83.56 | 90.31 | 65.65 | 95.35 | 78.18 | 73.99 | 72.08 | 61.47 |
| MSP-OE | (58) | FPR95 | 0.71 | 0.50 | 0.58 | 6.61 | 13.00 | 1.32 | 5.16 | 5.64 | 4.77 | 28.36 |
| | | AUROC | 99.60 | 99.78 | 99.84 | 98.77 | 97.27 | 99.70 | 98.95 | 98.87 | 98.42 | 93.29 |
| | | AUPR | 94.25 | 97.36 | 98.94 | 95.06 | 88.08 | 98.56 | 94.56 | 94.20 | 89.33 | 76.19 |
| MLV-OE | - | FPR95 | 0.69 | 0.43 | 0.49 | 4.98 | 11.17 | 1.11 | 4.10 | 4.52 | 4.08 | 30.38 |
| | | AUROC | 99.62 | 99.79 | 99.86 | 98.96 | 97.58 | 99.74 | 99.11 | 99.02 | 98.61 | 93.10 |
| | | AUPR | 94.30 | 97.46 | 99.07 | 95.72 | 89.10 | 98.71 | 95.15 | 94.68 | 90.11 | 76.36 |
| Ensemble | - | FPR95 | 6.84 | 16.71 | 16.71 | 15.99 | 100 | 12.34 | 25.04 | 100.00 | 16.71 | 100.00 |
| | | AUROC | 97.37 | 86.94 | 91.20 | 93.18 | 85.69 | 95.23 | 90.21 | 88.00 | 92.05 | 83.90 |
| | | AUPR | 82.32 | 41.71 | 64.52 | 71.49 | 56.32 | 78.07 | 64.99 | 61.03 | 67.29 | 53.00 |
| Mahalanobis | (150) | FPR95 | 1.35 | 2.01 | 7.38 | 35.82 | 48.38 | 28.61 | 27.98 | 39.02 | 24.79 | 48.40 |
| | | AUROC | 99.57 | 99.60 | 98.21 | 87.78 | 87.75 | 87.10 | 92.25 | 90.40 | 90.86 | 86.71 |
| | | AUPR | 96.49 | 97.95 | 90.63 | 46.79 | 55.33 | 41.59 | 65.14 | 62.17 | 53.36 | 54.06 |
| MC-Dropout | (44) | FPR95 | 15.31 | 33.58 | 16.54 | 20.75 | 38.77 | 16.81 | 28.44 | 34.62 | 28.73 | 37.48 |
| | | AUROC | 93.89 | 83.41 | 94.73 | 93.55 | 88.52 | 95.09 | 91.36 | 89.73 | 91.07 | 88.43 |
| | | AUPR | 63.52 | 35.40 | 74.91 | 71.65 | 58.19 | 77.26 | 65.34 | 61.76 | 62.41 | 56.84 |
| ODIN | (92) | FPR95 | 0.00 | 0.00 | 99.4 | 04.30 | 07.50 | 04.80 | 03.80 | 06.10 | 51.00 | 51.40 |
| | | AUROC | 100.00 | 99.90 | 42.50 | 99.10 | 98.50 | 99.00 | 99.20 | 98.80 | 89.90 | 88.3 |
| | | AUPR | 63.52 | 35.40 | 74.91 | 71.65 | 58.19 | 77.26 | 65.34 | 61.76 | 62.41 | 56.84 |

is to assign higher anomaly scores to rarely happening events. To address this issue, (155; 193) proposed fairness-aware ADs while using the label of protected variables as an extra supervision in the training process.

From a different point of view (181), introduces a significantly important bias in semi-supervised anomaly detection methods such as DSAD (137). Suppose DSAD has been implemented in law enforcement agencies to spot suspicious individuals using surveillance cameras. As a few number of training samples has been used as abnormal samples during the process, the trained model might have been biased towards detecting specific types of anomalies more than others. For instance, if the auxiliary abnormal training dataset includes more men than women, boundaries of detecting abnormal events as men might be looser than women at the test time. This could also happen in the classification settings such as OOD detection or OSR. (153) reports the existence of unfair biases toward some unrelated protected variables in detecting chest diseases for a classifier trained on Chest X-Ray datasets. From what has been said, it seems fairness and AD, ND, OSR, and OOD detection are strongly correlated because of some critical applications in which they are used and further research on their correlation is necessary for having practical models.

## 10.6  Multi-Modal Datasets

In many situation, training dataset consists of multi-modal training samples, for instance in Chest-X-Ray datasets, labels of images are found automatically by applying NLP methods on the prescribes of radiologists. In these situations, joint training of different modes could help models to learn better semantic features. However, in this way, models need to be robust in different modes too. For example, in visual Question Answering tasks, we expect our model not to produce any answer for out-of-distribution input texts or images. Note that the correlation between different modes must be attended here, and independent training of AD, ND, OOD detection, or OSR models for different modes results in sticking in local minimas. To cope with this issue, (87) has explored the performance of VQA models in detecting unseen test time samples. However, there are many more challenges need to be investigated in this way. For instance, the robustness and fairness of VQA OOD models is significantly more challenging compared to single mode datasets, besides due to heavy training process of these models, inventing few-shot methods might be demanding in the fields.

Table 4: OOD example detection for the maximum softmax probability (MSP) baseline detector, maximum logit value, the MSP detector after fine-tuning with Outlier Exposure (OE), the maximum logit value after fine-tuning with Outlier Exposure (OE), ensemble of 3 models, Mahalanobis distance, and Monte Carlo dropout. Inlier distribution is considered as CIFAR-100. All results are based on our evaluations and are average percentages of 10 runs. Missing values (-) indicate that we are proposing this setting, and it has no specific reference.

| Method | References | Criterion | Gaussian | Rademacher | Blob | TinyImageNet(crop) | TinyImageNet(resize) | LSUN(crop) | LSUN(resize) | iSUN | SVHN | CIFAR-10 |
|---|---|---|---|---|---|---|---|---|---|---|---|---|
| MSP | (56) | FPR95 | 54.32 | 39.08 | 57.11 | 43.34 | 65.88 | 47.32 | 62.98 | 63.34 | 69.12 | 65.14 |
| | | AUROC | 64.66 | 79.27 | 75.61 | 86.34 | 74.56 | 85.56 | 75.59 | 75.73 | 71.43 | 75.12 |
| | | AUPR | 19.69 | 30.05 | 29.99 | 56.98 | 33.71 | 56.49 | 34.11 | 33.88 | 30.44 | 33.92 |
| ODIN | (92) | FPR95 | 01.20 | 13.90 | 13.70 | 09.20 | 37.60 | 07.20 | 32.30 | 36.40 | 37.00 | 76.4 |
| | | AUROC | 99.50 | 92.60 | 95.90 | 97.90 | 90.80 | 98.30 | 91.90 | 90.50 | 89.00 | 73.20 |
| | | AUPR | 98.70 | 83.70 | 94.50 | 97.70 | 89.90 | 98.20 | 90.90 | 87.80 | 86.30 | 70.60 |
| MLV | | FPR95 | 71.89 | 72.35 | 81.09 | 22.51 | 66.17 | 22.20 | 61.30 | 60.86 | 67.01 | 64.41 |
| | | AUROC | 44.24 | 46.22 | 53.62 | 94.72 | 77.72 | 95.09 | 79.54 | 79.19 | 74.03 | 77.55 |
| | | AUPR | 13.82 | 14.20 | 15.98 | 79.10 | 38.00 | 81.11 | 39.14 | 37.27 | 31.99 | 37.30 |
| MSP-OE | (58) | FPR95 | 12.41 | 16.89 | 12.04 | 22.02 | 69.42 | 13.27 | 60.89 | 62.42 | 43.10 | 62.57 |
| | | AUROC | 95.69 | 93.01 | 97.11 | 95.69 | 76.04 | 97.55 | 80.94 | 79.96 | 86.86 | 75.41 |
| | | AUPR | 71.13 | 56.81 | 85.91 | 85.34 | 39.57 | 90.99 | 48.52 | 45.86 | 53.27 | 32.28 |
| MLV-OE | - | FPR95 | 10.71 | 16.66 | 8.09 | 17.34 | 73.95 | 08.50 | 56.02 | 60.73 | 32.59 | 64.91 |
| | | AUROC | 96.12 | 91.86 | 97.94 | 96.38 | 75.84 | 98.31 | 83.33 | 81.89 | 88.91 | 73.74 |
| | | AUPR | 72.81 | 52.03 | 88.60 | 86.55 | 39.72 | 92.90 | 51.06 | 48.21 | 54.72 | 30.48 |
| Ensemble | - | FPR95 | 22.72 | 43.51 | 48.07 | 44.68 | 100.00 | 47.26 | 100.00 | 91.44 | 57.18 | 57.18 |
| | | AUROC | 89.15 | 68.64 | 79.24 | 82.90 | 70.47 | 82.39 | 70.66 | 71.08 | 73.61 | 75.13 |
| | | AUPR | 46.45 | 21.61 | 35.80 | 44.31 | 28.98 | 44.12 | 28.75 | 28.66 | 31.76 | 32.62 |
| Mahalanobis | (150) | FPR95 | 0.82 | 0.10 | 2.70 | 73.79 | 43.40 | 76.42 | 37.94 | 42.07 | 32.05 | 70.80 |
| | | AUROC | 99.78 | 99.98 | 99.48 | 57.77 | 87.62 | 54.35 | 90.12 | 88.91 | 92.76 | 70.99 |
| | | AUPR | 98.63 | 99.88 | 97.64 | 17.27 | 60.18 | 16.11 | 65.97 | 62.82 | 75.87 | 27.12 |
| MC-Dropout | (44) | FPR95 | 54.45 | 41.41 | 46.64 | 47.32 | 68.05 | 55.38 | 63.53 | 65.03 | 75.98 | 63.33 |
| | | AUROC | 62.35 | 76.51 | 80.59 | 85.23 | 73.66 | 82.24 | 74.93 | 74.70 | 68.89 | 76.87 |
| | | AUPR | 18.74 | 27.14 | 35.08 | 54.12 | 32.57 | 49.08 | 33.04 | 32.26 | 28.63 | 36.31 |

## 10.7 Explainablity Challenge

Explainable AI (XAI) has found a seriously important role in the recently proposed deep network architectures, especially when they are used in safety-critical applications (5). In AD, OSR, ND and OOD detection due to some of their critical applications, we should be able to explain the reason of decisions our models make (63; 57). For instance, if a person is distinguished as suspicious in the surveillance cameras, there must be good reasons for that by which the model has made its decision.

The challenges of explainability can be defined into two different approaches. First, we should explain why a sample is normal, known or in-distribution. Secondly, we should explain why a sample is abnormal, unknown or out-of-distribution. There are a lot of different techniques to explain the decisions of models in the literature such as Grad-cam (151), Smoothfgrad (157), which have been used in Multi-KD (144), CutPaste (90) and (168). However, they only have been used to explain normal, seen or in-distribution samples and their results are not as accurate as enough for unseen or abnormal inputs. To cope with the issue, (96) proposed a VAE based method which can provide the reasons of abnormality of input samples while performing accurately on explaining normal samples as well. However, it does not work well on complex training datasets such as CIFAR-10, which shows the need of conducting more research toward mitigating the problem.

Another important challenge of explainability can be seen in one-class classification or ND approaches, in which there is only access to one-label at the training time. Therefore, Gradcam or Smoothgrad which use the availability of fine-grain labels can not be used anymore. To address this issue, (98) proposed a fully convolutional architecture with a heatmap upsampling algorithm that is called receptive field upsampling, which starts from a sample latent vector and reverse the effect of applied convolution operators to find important regions in the given input sample. However, explainable OCC models are still largely unexplored and more investigations in this direction are still necessary.

Table 5: OOD example detection for the maximum softmax probability (MSP) baseline detector. Inlier distribution is a set of faults in MVTecAD dataset, and outliers are rare faults. All results are average percentages of 10 runs.

| Class Name | Normal Sets | AUC | FPR | AUPR |
|---|---|---|---|---|
| Cable | good,combined, missing cable, poke insulation | 51.70 | 1 | 24.20 |
| Capsule | good, poke, faulty imprint, crack | 56.40 | 1 | 15.80 |
| Wood | good, color, scratch, liquid | 53.30 | 1 | 86.40 |
| Carpet | good, metal, contamination, hole | 50.20 | 1 | 14.70 |

## 10.8   Reliability of ODD Detection Methods

While OOD detection research has advanced considerably, SOTA performance has become saturated under the existing testing frameworks. Naturally, this raises the question of whether SOTA OOD detection methods are similarly effective in practice. Novel OOD detection research directions have recently emerged that expose weaknesses in existing SOTA methods that prevent their use in the real world. (78) introduces a novel evaluation framework for OOD detection. They introduced new test datasets with semantically preserved and realistic distribution shifts, as well as a novel metric to evaluate method generalization of OOD detection methods to real-world scenarios. Due to semantic consistency, SOTA OOD detection methods are expected to have high performance on realistic distribution shifts. Under realistic distribution shifts, SOTA methods are shown to have significant performance drops, demonstrating a serious concern when it comes to OOD detection in the real world. Similarly, (59) aims to make multi-class OOD detection in real-world settings more feasible by introducing new benchmark datasets consisting of high-resolution images over thousands of classes. As a result of their research, the authors demonstrate the need for new benchmarks to redirect research toward real-world applications. Additionally, (7) examines the adversarial safety of OOD detection methods and proposes a method that improves robustness. Defenses against adversarial attacks are also paramount for safely deploying OOD detection into the real world. The saturation of OOD detection performance does not signal the end of this research. Continuing to advance this field requires new standardized evaluation frameworks that more accurately quantify a method's practical effectiveness.

## 10.9   Multi-Label OOD Detection and Large-Scale Datasets

While OOD detection for multi-class classification has been extensively studied, the problem for multi-label networks remain underexplored (171). This means each input has more than one true label by which it must be recognized. This is more challenging since multi-label classification tasks have more complex class boundaries, and unseen behaviors could happen in a subset of input sample labels. Another challenge of multi-label datasets can be explored in anomalous segmentation tasks. Different from classification in which one can report an entire image as an abnormal input, the specific anomalous part must be specified here.

Current methods have been primarily evaluated on small data sets such as CIFAR. It's been shown that approaches developed on the CIFAR benchmark might not translate effectively into ImageNet benchmark with a large semantic space, highlighting the need to evaluate OOD detection in a large-scale real-world setting. Therefore, future research is encouraged to evaluate on ImageNet-based OOD detection benchmark (69), and test the limits of the method developed.

## 10.10   Open-World Recognition

Although detecting novel, unknown or out-of-distribution samples is enough in controlled lab environments, novel categories must be continuously detected and then added to the recognition function in real-world operational systems. This becomes even more challenging when considering the fact that such systems require minimal downtime, even to learn (10). While there has been much research on incremental (131) and life-long learning (120) for addressing the problem of adding new knowledge to a pre-existing one, open-world

Table 6: OOD example detection for the maximum softmax probability (MSP) baseline detector, maximum logit value (MLV), the MSP detector after fine-tuning with Outlier Exposure (OE), the maximum logit value after fine-tuning with Outlier Exposure (OE), ensemble of 3 models, Mahalanobis distance, and Monte Carlo dropout. Inlier distribution is considered as TinyImageNet. All results are based on our evaluations and are average percentages of 10 runs. Missing values (-) indicate that we are proposing this setting, and it has no specific reference.

| Method | References | Criterion | Gaussian | Rademacher | Blob | LSUN(crop) | LSUN(resize) | iSUN | SVHN |
|--------|-----------|-----------|----------|------------|------|------------|--------------|------|------|
| MSP | (56) | FPR95 | 72.34 | 47.60 | 90.31 | 29.33 | 44.37 | 45.68 | 44.75 |
| | | AUROC | 33.36 | 70.52 | 22.79 | 93.66 | 86.16 | 85.94 | 89.05 |
| | | AUPR | 12.27 | 22.76 | 10.55 | 77.91 | 50.79 | 51.16 | 67.21 |
| ODIN | (92) | FPR95 | 43.70 | 59.40 | 74.60 | 14.80 | 38.90 | 38.90 | 23.70 |
| | | AUROC | 70.00 | 50.80 | 46.20 | 96.80 | 87.10 | 87.60 | 93.90 |
| | | AUPR | 56.60 | 45.10 | 43.10 | 96.60 | 83.10 | 87.60 | 92.80 |
| MLV | - | FPR95 | 67.38 | 21.56 | 97.58 | 10.96 | 28.53 | 27.80 | 27.51 |
| | | AUROC | 45.34 | 90.24 | 15.96 | 97.75 | 91.71 | 91.98 | 94.09 |
| | | AUPR | 14.15 | 49.31 | 9.77 | 91.22 | 64.00 | 66.12 | 79.28 |
| MSP-OE | (58) | FPR95 | 45.32 | 49.53 | 0.05 | 0.53 | 0.12 | 0.12 | 0.39 |
| | | AUROC | 76.30 | 65.11 | 99.99 | 99.76 | 99.97 | 99.97 | 99.83 |
| | | AUPR | 28.32 | 19.97 | 99.93 | 98.37 | 99.82 | 99.79 | 98.16 |
| MLV-OE | - | FPR95 | 11.21 | 46.46 | 0.05 | 0.52 | 0.11 | 0.12 | 0.38 |
| | | AUROC | 95.45 | 68.30 | 99.99 | 99.81 | 99.97 | 99.97 | 99.83 |
| | | AUPR | 66.66 | 21.45 | 99.93 | 98.48 | 99.81 | 99.79 | 98.16 |
| Ensemble | - | FPR95 | 71.09 | 45.96 | 78.16 | 46.55 | 57.62 | 58.94 | 54.60 |
| | | AUROC | 35.75 | 74.09 | 51.86 | 83.72 | 76.54 | 75.70 | 77.09 |
| | | AUPR | 12.61 | 25.61 | 15.70 | 50.66 | 34.67 | 33.48 | 34.89 |
| Mahalanobis | (150) | FPR95 | 66.87 | 48.15 | 22.23 | 98.46 | 72.04 | 79.93 | 96.83 |
| | | AUROC | 48.74 | 70.28 | 92.41 | 13.33 | 71.11 | 66.65 | 27.59 |
| | | AUPR | 14.78 | 22.60 | 62.45 | 9.48 | 28.17 | 25.19 | 10.81 |
| MC-Dropout | (44) | FPR95 | 76.09 | 56.14 | 91.36 | 30.44 | 43.25 | 47.22 | 47.67 |
| | | AUROC | 30.38 | 58.92 | 21.31 | 93.13 | 85.68 | 84.45 | 87.44 |
| | | AUPR | 11.82 | 17.54 | 10.39 | 76.53 | 48.49 | 46.56 | 62.24 |

recognition needs a few more steps. This means novel classes must be found continuously, and the system must be updated to include these new classes in its multi-class open-set recognition algorithm.

The mentioned process poses many different challenges, from the scalability of current open-set recognition algorithms to designing new learning algorithms to avoid problems such as catastrophic forgetting (131) of OSR classifiers. Furthermore, all the previously mentioned future works can be reformulated in open-world recognition problems again, which means by considering a few existing works in these subjects, it needs to be explored more.

## 10.11  Vision Transformers in OOD Detection and OSR

Vision Transformers (ViTs) (36) have recently been proposed to replace CNNs and have shown a great performance in different applications such as object detection (19), medical image segmentation (166), and visual tracking (22). Similarly, some methods have recently reported the benefits of ViTs in OOD detection (81; 42) and have shown their capabilities at detecting near OOD samples. For instance, (42) has reported the significant superiority of ViTs compared to previous works when they are trained on CIFAR-10 and tested on CIFAR-100 as inlier and outlier datasets, respectively. However, as ViTs usually get pre-trained on extra-large datasets such as ImageNet-22K that has a large intersection with training and testing datasets, the integrity of the train-test mismatch does not hold anymore, and the problem would be converted to "how much does it remember from pretraining". This means ViTs should be evaluated on datasets that have no intersection with the pre-trained knowledge.

To address this issue, we have evaluated ViT-B16(36) on SVHN and MNIST when six randomly selected classes are considered as normal and the remaining ones as outliers or unseens. MSP is considered to detect unknown samples. As Table 9 shows, ViT-B16 that is pre-trained on ImageNet-22K is not comparatively as good as other baselines that are trained from scratch. As all the experiments are evaluated in a near

Table 7: AUROC results of the most notable works for anomaly/novelty detection. The performance is averaged for each dataset in the one-vs-all setting. All the results in the tables are reported from the reference papers.

(a)

| Dataset | Method | References | 0 | 1 | 2 | 3 | 4 | 5 | 6 | 7 | 8 | 9 | Mean |
|---|---|---|---|---|---|---|---|---|---|---|---|---|---|
| MNIST | OC-GAN | (122) | 99.80 | 99.90 | 94.20 | 96.30 | 97.50 | 98.00 | 99.10 | 98.10 | 93.90 | 98.10 | 97.50 |
| | LSA | (1) | 99.30 | 99.90 | 95.90 | 96.60 | 95.60 | 96.40 | 99.40 | 98.00 | 95.30 | 98.10 | 97.50 |
| | AnoGan | (147) | 96.60 | 99.20 | 85.00 | 88.70 | 89.40 | 88.3 | 94.70 | 93.50 | 84.90 | 92.40 | 91.30 |
| | OC-SVM | (148) | 99.50 | 99.90 | 92.60 | 93.60 | 96.70 | 95.50 | 98.70 | 96.60 | 90.30 | 96.20 | 96.00 |
| | DeepSVDD | (136) | 98.00 | 99.70 | 91.70 | 91.90 | 94.90 | 88.50 | 98.30 | 94.60 | 93.90 | 96.50 | 94.80 |
| | U-std | (14) | 99.90 | 99.90 | 99.00 | 99.30 | 99.20 | 99.30 | 99.70 | 99.50 | 98.60 | 99.10 | 99.35 |
| | Multiresolution | (144) | 99.82 | 99.82 | 97.79 | 98.75 | 98.43 | 98.16 | 99.43 | 98.38 | 98.41 | 98.10 | 98.71 |

(b)

| Dataset | Method | References | Plane | Car | Bird | Cat | Deer | Dog | Frog | Horse | Ship | Truck | Mean |
|---|---|---|---|---|---|---|---|---|---|---|---|---|---|
| CIFAR-10 | OC-GAN | (122) | 75.70 | 53.10 | 64.00 | 62.00 | 72.30 | 62.0 | 72.30 | 57.50 | 82.00 | 55.40 | 65.66 |
| | LSA | (1) | 73.50 | 58.00 | 69.00 | 54.20 | 76.10 | 54.60 | 75.10 | 53.50 | 71.70 | 54.80 | 64.10 |
| | AnoGan | (147) | 96.60 | 99.20 | 85.00 | 88.70 | 89.40 | 88.3 | 94.70 | 93.50 | 84.90 | 92.40 | 91.30 |
| | OC-SVM | (148) | 63.00 | 44.00 | 64.90 | 48.70 | 73.50 | 50.00 | 72.50 | 53.30 | 64.90 | 50.80 | 58.56 |
| | DeepSVDD | (136) | 98.00 | 99.70 | 91.70 | 91.90 | 94.90 | 88.50 | 98.30 | 94.60 | 93.90 | 96.50 | 94.8 |
| | GT | (48) | 76.20 | 84.80 | 77.10 | 73.20 | 82.80 | 84.80 | 82.00 | 88.70 | 89.50 | 83.40 | 82.30 |
| | CSI | (165) | 89.90 | 99.10 | 93.10 | 86.40 | 93.90 | 93.20 | 95.10 | 98.70 | 97.90 | 95.50 | 94.30 |
| | U-std | (14) | 78.90 | 84.90 | 73.40 | 74.80 | 85.10 | 79.30 | 89.20 | 83.00 | 86.20 | 84.80 | 81.96 |
| | Multiresolution | (144) | 90.53 | 90.35 | 79.66 | 77.02 | 86.71 | 91.40 | 88.98 | 86.78 | 91.45 | 88.91 | 87.18 |

(c)

| Dataset | Method | References | Bottle | Cable | Capsule | Carpet | Grid | Hazelnut | Leather | Metal-nut | Pill | Screw | Tile | Toothbrush | Transistor | Wood | Zipper | Mean |
|---|---|---|---|---|---|---|---|---|---|---|---|---|---|---|---|---|---|---|
| MVTecAD | LSA | (1) | 86.00 | 80.00 | 71.00 | 67.00 | 70.00 | 85.00 | 75.00 | 74.00 | 70.00 | 54.00 | 61.00 | 50.00 | 89.00 | 75.00 | 88.00 | 73.00 |
| | AnoGan | (147) | 69.00 | 50.00 | 58.00 | 50.00 | 52.00 | 62.00 | 68.00 | 49.00 | 51.00 | 51.00 | 53.00 | 67.00 | 57.00 | 35.00 | 59.00 | 55.00 |
| | DeepSVDD | (136) | 86.00 | 71.00 | 69.00 | 75.00 | 73.00 | 77.00 | 87.00 | 54.00 | 81.00 | 59.00 | 71.00 | 65.00 | 70.00 | 64.00 | 74.00 | 72.00 |
| | GT | (48) | 74.29 | 33.32 | 67.79 | 82.37 | 82.51 | 65.16 | 48.24 | 45.90 | 53.86 | 61.91 | 84.70 | 79.79 | 94.00 | 44.58 | 87.44 | 67.06 |
| | U-std | (14) | 93.10 | 81.80 | 96.80 | 87.90 | 95.20 | 96.50 | 94.50 | 94.20 | 96.10 | 94.20 | 94.60 | 93.30 | 66.60 | 91.10 | 95.10 | 91.40 |
| | Multiresolution | (144) | 99.39 | 98.37 | 80.46 | 73.58 | 95.05 | 82.70 | 94.29 | 79.25 | 91.57 | 78.01 | 89.19 | 85.55 | 92.17 | 83.31 | 93.24 | 87.74 |

Table 8: AUROC results of the most notable works for OSR. The performance is averaged across each dataset using the explained evaluation procedures for 10 random trials.(MLS stands for (167)).

| Method | References | MNIST | SVHN | CIFAR-10 | CIFAR + 10 | CIFAR + 50 | TinyImageNet |
|--------|-----------|-------|------|----------|-----------|-----------|-------------|
| OpenMax | (11) | 98.10 | 89.40 | 81.10 | 81.70 | 79.60 | 81.10 |
| G-OpenMax | (46) | 98.40 | 89.60 | 67.60 | 82.70 | 81.90 | 58.00 |
| OSRCI | (33) | 98.80 | 91.00 | 69.90 | 83.80 | 82.70 | 58.60 |
| C2AE | (117) | 98.90 | 92.20 | 89.50 | 95.50 | 93.70 | 74.80 |
| CROSR | (183) | 99.20 | 89.90 | 88.30 | - | - | 58.90 |
| GDFR | (123) | - | 93.50 | 80.70 | 92.80 | 92.60 | 60.80 |
| RPL | (22) | 99.60 | 96.80 | 90.10 | 97.60 | 96.80 | 80.90 |
| OpenGan | (82) | 99.90 | 98.80 | 97.30 | - | - | 90.70 |
| MLS | (167) | 99.30 | 97.10 | 93.60 | 97.90 | 96.50 | 83.00 |

Table 9: OSR AUROC results of baselines such as the maximum softmax probability (MSP), Openmax, and CROSR compared to ViT-B16. Known distribution is considered a random selection of 6 classes of the respective datasets and unknown ones are the rest. All results are percentages and the average of 5 runs.

| Method | SVHN | MNIST |
|--------|------|-------|
| Softmax | 88.60 | 97.80 |
| Openmax | 89.40 | 98.10 |
| CROSR | 89.90 | 99.10 |
| ViT-B16 | 82.18 | 94.89 |

OOD detection setting, they support the before-mentioned deficiency of ViTs. From what has been said, a future line of research could be evaluating ViTs in a more controlled situation such that their real benefits would be more precise. Indeed, the recent Species dataset collects examples that do not fall under any of the ImageNet-22K classes and is a first step to rectify this problem (62).

### 10.12 Identifiablity Problems

Providing a method whereby unfamiliar samples are detected is not always trivial. (45) investigates the theoretical foundations of the problem, demonstrating that determining the accuracy is just as difficult as identifying the ideal predictor, and so the success of any method is dependent on the implicit assumptions about the nature of the shift, which is characterized by mismatches between the source (training) and target (test) distributions. It is demonstrated that no approach of measuring accuracy will work in general without assumptions on the source classifier or the nature of the shift. Average Thresholded Confidence (ATC) is a simple approach for predicting model performance using softmax probability. It learns a threshold for model confidence on the validation source data and predicts the target domain accuracy as the proportion of unlabeled target points with a score greater than the threshold. This work advances the positive answer to the question of a feasible technique for selecting a threshold that enables prediction accuracy with the thresholded model confidence.

## 11 Conclusion

In many applications, it is not feasible to model all kinds of classes occurring during testing; thus, scenarios existing in domains such as OOD detection, OSR, ND (one-class learning), and AD become ubiquitous. Up to this time, these domains, despite having the same intention and a large intersection, have been followed independently by researchers.

To address the need, this paper gives a comprehensive review on existing techniques, datasets, evaluation criteria, and future challenges. More importantly, limitations of the mentioned approaches are discussed, and

certain promising research directions are pointed out. We hope this helps the research community build a broader and cross-domain perspective.

## Acknowledgements

We would like to thank Yuki M. Asano for the extremely useful discussions and for reviewing the paper prior to submission.

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

# A   Appendix

You may include other additional sections here.

