# OpenReview forum: "A Unified Survey on Anomaly, Novelty, Open-Set, and Out of-Distribution Detection: Solutions and Future Challenges"
_TMLR — Accepted by TMLR_

### Review · Reviewer_8zi1 · 2022-08-09

**Summary Of Contributions:**

The paper aims to comprehensively review relevant works in out-of-distribution, open-set, and anomaly detection. The paper highlights an essential concern with other survey papers in the field, i.e., the existing survey papers only focus on a specific domain without examining the relationship between different domains. This paper provides a cross-domain and comprehensive review of the literature across these areas and at the same time identifies common ideas.

**Requested Changes:**

- (Critical to securing your recommendation for acceptance) The paper discusses different methods in different subsections for each of the survey topics. While Section 2 is added for taxonomy, currently, it seems that the discussion of each method is fairly disjoint without a clear connection between different methods. I encourage authors to provide summary unifying ideas across similar methods and more broadly across different areas.

-  (Critical to securing your recommendation for acceptance) Several things in Figure 2 are unclear. The phrase 'Unfamiliar Data Detection Approaches' looks unfamiliar to me, so it is a bit hard to parse the remaining hierarchy. It is also clear why 'Pre-trained Feature
Adaptation ' is branched separately from the discriminative branch.


**Strengths And Weaknesses:**

**Strengths**

- Survey paper is clearly written and easy to follow. The exposition is clear and situated nicely and is improved from its last version. Methods are described both mathematically and visually. This survey paper can help newcomers reduce their barriers to entry.

- Table 1 and Figure 2 are illustrative and nice additions to the latest version. I have some minor comments, please read below.

- Explanation for each method covered in the paper is clear. To the best of my knowledge, the survey does a good job of covering relevant papers with diverse ideas in each field.

- Authors have also made some efforts to include relatively old approaches.

**Weaknesses**

- Figure 2 can be improved a bit (see comments below).
- While authors have included Section 2 to discuss the taxonomy, relation between different approaches is still left a bit unclear
- Section 3 looks a bit out of the place.
- Typos. Section 3.4 title.

---

> ### Author Response · Authors · 2022-08-16
> **Requested Changes:**
>
> Thank You for your comments.
> As you mentioned, section 2 tries to give readers a unified perspective on the relationships of different tasks as the main message of this paper. Afterward, we reviewed the critical/baselines methods of anomaly detection, open set recognition, and novelty detection. We have explained each technique/method and clarified how these methods are connected to others. (For examples, see the explanation of "Redefining the Adversarially Learned One-Class Classifier Training Paradigm (Old-is-Gold) Zaheer
> et al. (2020):" paper. We carefully discussed how this method is related to the ALLOC method.)
> We agree with the reviewer that summarizing those papers with respect to the core ideas is also very beneficial, but in this way, we have to ignore the details of the methods (due to limited space).  On the other side, the main difference between these methods is their evaluation setting.  We thoroughly discuss that in section 7.
> We are also modifying Fig 2 to make it more clear. We will upload a new version of the paper very soon.

---

> ### Author Response · Authors · 2022-08-27
> **The new version has been uploaded.**
>
> The authors kindly ask you to check out the new version. For the first concern, please refer to our last response.
>
> For the last concern, the authors agree with the reviewer and changed "Unfamiliar Data Detection" to "The Taxonomy Of the Explained Methods". This especially makes the paper clear for inexperienced and newcomers in the field. Also, please note that methods that are based on pre-trained feature adaptation are not trained from scratch and somehow employ adaptation techniques that might be discriminative, generative, or even a hybrid of both. As reviewer "pimz" also pointed out, separating pre-trained adaptation methods from others is fair since they have privileged access to out-of-domain knowledge.
>
> Finally, the authors hope that they have solved all your concerns.

---

### Review · Reviewer_GscL · 2022-08-20

**Summary Of Contributions:**

The paper provides a review the related work in Anomaly Detection (AD), Novelty Detection (ND), out of distribution detection (OOD) and Open Set Recognition (OSR), aiming to provide a unified view of these problems in machine learning. The paper provides an extensive review of different state-of-the-art methods in these areas as well as an empirical evaluation on different synthetic and real datasets.

**Broader Impact Concerns:**

Many figures and algorithms have been taken from the original papers, but the authors in this paper do not include any explicit reference to the original source. I don’t think that the authors have done this maliciously, but this can be a source of problems in terms of copyright infringement and plagiarism.

**Requested Changes:**

I have read the comments and requested changes in the previous version of the paper and briefly checked the changes in the paper with respect to the previous version. I fully agree with the changes suggested by the action editors and I think that the authors have not really addressed them for this revised version of the manuscript.
In this sense, my main concern is that the paper mostly focuses on the short description (in some cases insufficient) of a large number of papers rather than providing that unified view promised in the introduction. For instance, the authors mention differences between novelty and anomaly detection tasks (depending on the degree of supervision). But then, in Section 3 both anomaly and novelty detection methods are treated a bit arbitrarily. For example, OC-SVM is defined as a novelty detection method, but the authors include it in a section for anomaly detection methods.
The authors have done a good job on gathering a very good number of relevant methods for the different tasks, but the survey needs a significant amount of work in relating them to provide that unified view, which is the objective of the survey.
The description of some of the methods needs to be revised and improved: in some cases, these are too brief and includes equations or algorithms that are not really explained.
Section 9 seems too disconnected. Some of the ideas can be of use, but the discussions sometimes are shallow and the different aspects considered in the different subsections are not well connected.
As mentioned by other reviewers and the action editors in the previous version of the manuscript, the writing quality of the paper can be significantly improved and the paper should be proofread carefully before submission. In this sense, I don’t see significant improvements with respect to the previous version.
There is no relevant final discussion or conclusion in the paper. Section 10 is too short. I think this can also be significantly improved.


**Strengths And Weaknesses:**

Strengths:
+  The idea of providing a unified survey including AD, ND, OOD, and OSR can be of interest and useful to a broad audience in the machine learning community.
+ The authors included an extensive set of techniques across the different problems.
+ The paper includes an empirical evaluation comparing the methods explained in the survey.

Weaknesses:
- The paper does not really deliver the unified view of the different learning tasks considered, as stated in the introduction. The paper mostly focuses on the brief description of an extensive number of methods in the related work.
- Some of the descriptions of the algorithms included in the survey are very brief and includes equations that are not really explained (e.g. equations (1) and (2)). It gives the impression that the paper prioritizes quantity (of the methods included in the survey) than quality, which really limits the usefulness of the paper.
- The different subsections in Section 9 looks rather disconnected and the storyline is not clear.
- The writing quality of the paper can be significantly improved (typos, grammar, etc.).
- The paper includes many figures that are not original from the authors, but taken from other papers without making the reference explicit. This can create problems related to copyright infringement and plagiarism.

---

> ### Author Response · Authors · 2022-08-21
> **unified perspective, AD+ND, missing citations**
>
> Thank you for the comments.
> About a unified perspective: Please be advised the unified review here means that how these tasks, i.e., OSR, ND, AD, and ODD are related/connected (As main our contribution? Please let us know if another paper addresses this crucial issue in the field which comprehensively discuss this). And what is the main difference between these tasks. We do not tend to provide a unifying perspective on the previously proposed methods (There are a lot of survey papers about this). In this way, we explained and showed how these tasks are connected. Please see Fig 1 and 2, Tab 1,  and the introduction and evaluation protocol section. We have just explained the notable previous methods for each task to make the paper more beneficial for the readers. All in all, we can emphasize that the message of this paper is providing a unified overlooking of these correlated tasks( not the methods). We suggest the reviewer again read the paper completely (without skipping any section) to get the paper's central message.
>
> About ND and AD:   I guess you miss some parts of the papers. We have clarified /explained that ND and AD are two different approaches. However, the community considers both the same in most papers. We follow the community when we have explained methods, while we have provided more explanations about this in other sections.
>
> A survey aims not to deliver the technical novelty of each of the previous methods; the duty is to provide a complete review of the most notable previous methods where you can see we have done that.
>
> About the reference to the original papers of the photos:  We do not agree with you; in each part where we explain a method (paper), we cite the original paper, which shows that all materials from that part are borrowed from that paper. Please read the text carefully, if you do not change your mind, we can cite the original paper in the caption of each figure.

---

> ### Author Response · Authors · 2022-08-21
> **Insufficient Description of Equations and The Explanations of Section 9**
>
> Insufficient Description of Equations  : Please note that the methods should be explained based on their practicality according to the current advances in the field; otherwise, the paper could end up being very long, which from the author's point of view, is not useful for anyone. OC-SVM is fairly old and mainly based on SVM, a famous method in the field. Therefore, we think there is no need to re-explain the algorithm; instead, we have explained the philosophy behind manipulating SVM for one-class learning applications and addressed how it was further improved. Furthermore, as other reviewers also mentioned, I would refer you to the explanations of deep-learning based methods that are carefully selected and discussed sufficiently to satisfy the needs of either newcomers or experts in the field.
>
> The Explanations of Section 9 : The title of this section is "Future works", and future works, by definition, could be elaborated even independently. We have done our best to not only give intuitions about the future areas but also provide empirical evidence to support our claims as you also mentioned in the strengths. Besides, we also made some cross-bridges between the future works wherever it was possible; for instance, we shed light on the abilities of adversarial attacks (with a large perturbation noise) as a data augmentation technique to generate fake abnormal samples to improve the performance of AD methods, which is totally different from their common functionality in the field. Also, I would refer you to "open-world recognition", which discusses the practical challenges of implementing out-of-distribution detection, anomaly detection, and open-set recognition methods.
>
> Finally, we addressed the commonalities of different approaches in the corresponding explanations of each method. For instance, take a look at the description of (4.13), which discusses the similarity of GOAD with GT and DSVDD. Also, it is explained that several works that are considered different techniques in AD and OOD, implicitly have introduced the same technique, "outlier exposure", such as 6.6 and 4.10. There are also many other similarities that are addressed in the explanations.

---

> ### Author Response · Authors · 2022-08-27
> **The new version has been uploaded.**
>
> The authors kindly ask you to check out the new version. For your first concern, please refer to our previous responses.
>
> For the last concern, we have added Section 8 and tried our best to make the explanation of the methods self-contained. Also, as Reviewer "8zi1" mentioned, we tried our best to polish the paper and make it clear; however, we applied one further polishing to remove probable few typos.

---

### Review · Reviewer_pimz · 2022-08-22

**Summary Of Contributions:**

This paper provides a unified review of anomaly detection, open-set recognition, and out-of-distribution detection. While each of these directions has been studied extensively, a compressive review was most certainly needed. This paper effectively fills this gap. It provides an exhaustive summary of a wide range of works in each direction and discussed future challenges for all three domains.

**Broader Impact Concerns:**

None.

**Requested Changes:**

Already discussed with suggestions in the section above.

**Strengths And Weaknesses:**

This paper provides a very exhaustive review of three major domains in open-world machine learning. Each of these domains has its own reviews published previously, so the challenge of writing a joint review is indeed paramount. The paper meets the bar in discussing a wide range of papers from each domain.

However, it still lacks a borad categorization of the paper discussed (even within each domain). I discuss some of the key limitations and suggested changes below.

*Disentangling core challenges and future works*: Authors provide an extensive list of future challenges for the community. I recommend splitting this list into two parts. 1) Establish the core challenges that the community has already extensively been working on. E.g. self-supervised outlier detection (similar setting in anomaly detection), and few-shot detection are some of the techniques that have been recently explored. 2) Identify the challenges that the community is still missing as a recommendation or future work (e.g., use of transformers or explainability of detection models)

*Extensive quantitative benchmarking missing for AD/ND tasks*: While the paper compared most works in OOD detection quantitatively (Table 3,4,5,6) it lacks a similar rigorous benchmarking for AD and ND tasks (table 7, 8). I suggest incorporating something similar to cifar-10 per-class anomaly detection benchmark ([1] and follow-up works)

*Missing categorization of works* The current version of the paper also lacks a concrete categorization of existing works. The existing categorization is done (table 1) based on the domain (OOD detection, AD, OSR). This is just the global categorization, and I believe that each work should be categorized in a more fine-grained manner. For example, they can be categorized into.
* Use of labels: supervised, unsupervised, or semi-supervised
* Paradigm: deep-learning vs non-deep learning
* Use of Pretraining only (feature-space detection) vs training from scratch

While some of these details are available in the detailed description provided for each method, it is necessary to elevate them to broad categorization (e.g., in a table). In existing table-1, I suggest authors add the publication venue details of each method, as it might highlight if the research in three communities (OOD, OSR, AD) are clustered across venues.

*Fig.1*: The left-half of the figure (concentric circles) makes it hard to parse the definition of each domain, and in fact can lead to the wrong conclusion. For example, if I follow the color combination of detectors on the right, e.g., yellow for OOD detection, then it gives the impression that the yellow color band on the left is actually the region of OOD images. But that is not the case, as indicated by the definition (a U b) vs (c U d). One way to avoid this confusion would be to highlight these definitions (at least increase their font size). Overall I encourage the authors to improve the presentation of this figure.

Related work (section 9.5) Sehwag et al. (2019) initiated the work on the robustness of OOD detection under adversarial conditions and effective defenses, which is followed up by Meinke et al. (2021) and Chen et al.(2021). I suggest adding this related work.

Attribution: In each figure caption, I recommend citing the paper from which the figure is borrowed.

A very minor issue in fig.1 is that while the caption highlight four classes (car, dog, cat, and airplane), one of the image is a bus (red London bus). This gives a wrong depiction of the task. Though easy fix.

* Tack, Jihoon, et al. "Csi: Novelty detection via contrastive learning on distributionally shifted instances." Advances in neural information processing systems 33 (2020): 11839-11852.
* Sehwag, Vikash, et al. "Analyzing the robustness of open-world machine learning." Proceedings of the 12th ACM Workshop on Artificial Intelligence and Security. 2019.
* Meinke, Alexander, Julian Bitterwolf, and Matthias Hein. "Provably Robust Detection of Out-of-distribution Data (almost) for free." arXiv preprint arXiv:2106.04260 (2021).
* Chen, Jiefeng, et al. "Atom: Robustifying out-of-distribution detection using outlier mining." Joint European Conference on Machine Learning and Knowledge Discovery in Databases. Springer, Cham, 2021.

---

> ### Author Response · Authors · 2022-08-22
> **Requested Changes**
>
> We appreciate your comprehensive comment. We will try to address all your concerns in the next version to make a high-quality paper for the community.
>
> About your first comment, We also agree and try to disentangle the core challenges from the future works in the next version.
>
> For the second comment, we will also provide per-class results for cifar-10. We think this could give more intuition to the field since the performance of models is not equal on all the classes of cifar-10, and reporting it might reveal more information about the behavior of methods.
>
> For the last one, please note that we have already provided a categorization in Fig.2, mostly similar to what has been suggested.
> We kindly ask the reviewer to retake a look at this figure, in which we categorize methods based on the "use of labels" and "Use of Pretraining only". As we mentioned in section 2, while some surveys report fine-grained categorizations, they don't seem very precise; for instance, they make reconstruction-based methods independent of density estimation-based approaches, which might not be accurate. It can be shown that reconstruction-based techniques such as VAEs approximate the input distribution by some independent complex Gaussians.   Therefore, we have tried to make some keywords (table 1) and very general categorizations to avoid imprecise and vague categorizations.
>
> We will also apply other minor suggestion and add the references in the next version. Finally, the authors hope that they have addressed all your concerns thoroughly and reasonably.

---

> ### Author Response · Authors · 2022-08-27
> **The new version has been uploaded.**
>
> The authors kindly ask you to check out the new version. As suggested by the reviewer, we have disentangled the core challenges and future works. In the core challenges, topics such as self-supervised learning for outlier detection, few-shot learning, and data augmentation are discussed.  The detailed explanation can be found in section 8.
>
> To address your second concern, we have expanded Table 7 and reported per-class results for the most commonly used datasets, MNIST, CIFAR-10, and MVTecAD. Furthermore, please note that we have created a GitHub repository that includes OOD baselines and all the evaluation metrics discussed in the paper, which can benchmark SOTA pre-trained models on different datasets. This will help the community have a consistent evaluation code to compare and benchmark different methods.
>
> For the third concern, as we explained in the last response, a categorization almost the same as what has been suggested exists in Fig.2. Please check out our last response.
>
> We agree with the reviewer about Fig 1 and applied the suggestions to clarify the figure. The suggested related works have been added. Also, references to the works from which the figures are taken have been specified.
>
> Finally, the authors hope that they have solved all your concerns.

---

### Author Response · Authors · 2022-09-07
**Discussion**

Dear Reviewers
Thank you for your tremendous and constructive comments. As we previously mentioned, we did our best to address your concerns and improve paper quality. We eagerly look forward to hearing from you about our responses and the new version of the paper. Discussing between reviewers and us will lead to avoiding misconceptions/misunderstandings and also help us to improve the quality of the paper more and more.